# Stochastic Anderson Mixing for Nonconvex Stochastic Optimization

**Fuchao Wei[1], Chenglong Bao[3,4*], Yang Liu[1,2]**
[1]Department of Computer Science and Technology, Tsinghua University
[2]Institute for AI Industry Research, Tsinghua University
[3]Yau Mathematical Sciences Center, Tsinghua University
[4]Yanqi Lake Beijing Institute of Mathematical Sciences and Applications

## Abstract

Anderson mixing (AM) is an acceleration method for fixed-point iterations. Despite its success and wide usage in scientific computing, the convergence theory of AM remains unclear, and its applications to machine learning problems are not well explored. In this paper, by introducing damped projection and adaptive regularization to the classical AM, we propose a Stochastic Anderson Mixing (SAM) scheme to solve nonconvex stochastic optimization problems. Under mild assumptions, we establish the convergence theory of SAM, including the almost sure convergence to stationary points and the worst-case iteration complexity. Moreover, the complexity bound can be improved when randomly choosing an iterate as the output. To further accelerate the convergence, we incorporate a variance reduction technique into the proposed SAM. We also propose a preconditioned mixing strategy for SAM which can empirically achieve faster convergence or better generalization ability. Finally, we apply the SAM method to train various neural networks including the vanilla CNN, ResNets, WideResNet, ResNeXt, DenseNet and LSTM. Experimental results on image classification and language model demonstrate the advantages of our method.

## 1 Introduction

Stochastic optimization is important in various areas such as statistics [15], machine learning [4, 54] and power systems [25]. In this paper, we consider the following stochastic optimization problem:

$$\min_{x \in \mathbb{R}^d} f(x) = \mathbb{E}_\xi \left[ F(x; \xi) \right], \tag{1}$$

where $F : \mathbb{R}^d \times \Xi \to \mathbb{R}$ is continuously differentiable and possibly nonconvex and the random variable $\xi \in \Xi$ may follow an unknown probability distribution. It is assumed that only noisy information about the gradient of $f$ is available through calls to some *stochastic first-order oracle* ($\mathcal{SFO}$). One special case of (1) is the *empirical risk minimization* problem:

$$\min_{x \in \mathbb{R}^d} f(x) \stackrel{\text{def}}{=} \frac{1}{T} \sum_{i=1}^T f_{\xi_i}(x), \tag{2}$$

where $f_{\xi_i} : \mathbb{R}^d \to \mathbb{R}$ is the loss function corresponding to the $i$-th data sample and $T$ denotes the number of data samples. $T$ can be extremely large such that it prohibits the computation of the full gradient $\nabla f$. Thus designing efficient and effective numerical algorithms for solving problem (1) or (2) with rigorous theoretical analysis is a challenging task.

---

*Corresponding author. clbao@mail.tsinghua.edu.cn.

35th Conference on Neural Information Processing Systems (NeurIPS 2021).

One classical approach for solving (1) is the stochastic gradient descent (SGD) method [45]. It mimics the gradient descent (GD) method with noisy gradients and exhibits optimal convergence rate for some strongly convex stochastic problems [10, 48]. Some early related works of SGD in convex optimization can be found in [38, 39, 40]. For nonconvex cases, Ghadimi and Lan [17] proposed a randomized stochastic gradient (RSG) method that randomly selects a solution $\bar{x}$ from previous iterates. To ensure $\bar{x}$ satisfying $\mathbb{E}\left[\|\nabla f(\bar{x})\|_2^2\right] \leq \epsilon$, the total number of $\mathcal{SFO}$-calls[2] needed by RSG is $O\left(\epsilon^{-2}\right)$. For training neural networks, adaptive learning rate methods were proposed to accelerate SGD, e.g. Adagrad [12], RMSprop [55] and Adam [29], though their theoretical properties in nonconvex optimization were unknown until recently [9]. Regarding to second-order optimization methods [49, 36, 37], one notable work is the framework of stochastic quasi-Newton (SQN) methods proposed by Wang et al. [60], which covers a class of SQN methods and has theoretical guarantees in nonconvex stochastic optimization. However, these second-order methods usually demand more gradient evaluations in every iteration and less noisy gradient information to achieve actual acceleration [4]. There are also some recent works inspired from the dynamics perspective, such as the Langevin dynamics based algorithms [62, 33, 11], but many of them focus on the theory and their applications in deep learning are still under-explored.

In practice, the choices of optimizers can vary for different applications [63] and their practical performances are still unsatisfactory in terms of convergence rate or generalization ability [66]. For these reasons, we develop a novel second-order method based on Anderson mixing (AM) [2]. Our analysis and experiments show the proposed method is competent from both theoretical and practical perspectives and performs well in training various neural networks in different tasks.

AM is a sequence acceleration method in scientific computing [5]. It is widely used to accelerate the slow convergence of nonlinear fixed-point iterations arising in computational physics and quantum chemistry, e.g. the self-consistent field iteration in electronic structure calculations [16, 8], where the function evaluation is costly. It turns out that AM is closely related to multisecant quasi-Newton methods in nonlinear problems [14, 5] or the GMRES method [46] in linear problems [59, 41].

Inspired by the great success of AM in accelerating fixed-point iterations, it is natural to ask whether AM can be applied to accelerate nonlinear optimization since the gradient descent with constant stepsize is a fixed-point iteration. This idea has been explored in [50, 51], but the proposed Regularized Nonlinear Acceleration (RNA) method is built on the minimal polynomial extrapolation (MPE) approach [5], a sequence transformation method that has subtle difference from AM. Also, their methods rely heavily on the contraction assumption of the fixed-point map and the strong convexity. For AM, although current research has proved local linear convergence of AM for fixed-point iterations under some conditions [56, 57, 3], there exists no version of AM that guarantees convergence for nonconvex optimization, let alone stochastic optimization.

In this paper, we develop a stochastic version of AM. Due to the nonconvexity and noise inside the problems and lack of line search or trust-region, a straightforward migration of AM to nonconvex stochastic optimization is infeasible. As a result, we make several fundamental modifications to AM. We highlight the main contributions of our works as follows:

1. We develop a stochastic version of AM, namely Stochastic Anderson Mixing (SAM), by introducing *damped projection* and *adaptive regularization*. We prove its almost sure convergence to a stationary point and analyze its work complexity. When a randomly chosen iterate $x_R$ is returned as the output of SAM, we prove that the worst-case $\mathcal{SFO}$-calls complexity to guarantee $\mathbb{E}\left[\|\nabla f(x_R)\|_2^2\right] \leq \epsilon$ is $O\left(\epsilon^{-2}\right)$. (See Theorems 1-4.)

2. We give a variance reduced extension of SAM by borrowing the stochastic variance reduced gradient (SVRG) [28] technique and analyze its $\mathcal{SFO}$-calls complexity. (See Theorem 5.) We also propose a preconditioned mixing strategy for AM and obtain the preconditioned SAM method which can empirically converge faster or generalize better. (See Section 2.3.)

3. Extensive experiments on training Convolutional Neural Network (CNN), ResNet, WideResNet, ResNeXt, DenseNet, and LSTM on different tasks and datasets show the faster convergence or better generalization ability of our method compared with the state-of-the-art methods. (See Section 4.)

---

[2]One call to the $\mathcal{SFO}$ in (1) or one evaluation of $\nabla f_{\xi_i}(x)$ in (2) is counted as one $\mathcal{SFO}$-call.

## 2 Methodology

### 2.1 Anderson Mixing

AM is proposed for acceleration of fixed-point iterations. We assume the fixed-point iteration is $x_{k+1} = g(x_k) \stackrel{\text{def}}{=} x_k + r_k$, where $r_k \stackrel{\text{def}}{=} -\nabla f(x_k)$. Then $g(x_k) - x_k = r_k$. Here, we adopt the description of AM in [14, 59]. Let $\Delta$ denote the forward difference operator, say, $\Delta x_k = x_{k+1} - x_k$. Let $X_k$ and $R_k$ record the most recent $m(m \leq k)$ iterations:

$$X_k = [\Delta x_{k-m}, \Delta x_{k-m+1}, \cdots, \Delta x_{k-1}], R_k = [\Delta r_{k-m}, \Delta r_{k-m+1}, \cdots, \Delta r_{k-1}]. \quad (3)$$

AM can be decoupled into two steps. We call them *the projection step* and *the mixing step*:

$$\bar{x}_k = x_k - X_k \Gamma_k, \quad \text{(Projection step)} \quad (4a)$$

$$x_{k+1} = \bar{x}_k + \beta_k \bar{r}_k, \quad \text{(Mixing step)} \quad (4b)$$

where $\beta_k$ is the mixing parameter, and $\bar{r}_k \stackrel{\text{def}}{=} r_k - R_k \Gamma_k$ is reminiscent of *extragradient* [30]. $\Gamma_k$ is determined by solving

$$\Gamma_k = \underset{\Gamma \in \mathbb{R}^m}{\arg\min} \|r_k - R_k \Gamma\|_2. \quad (5)$$

Combining (4a) and (4b), we obtain the full form of AM [14, 59, 5, 42]:

$$x_{k+1} = x_k + \beta_k r_k - (X_k + \beta_k R_k)\Gamma_k. \quad (6)$$

**Remark 1.** *To see the rationality of AM, we assume $f$ is twice continuously differentiable. Then a quadratic approximation of $f$ implies $\nabla^2 f(x_k)(x_j - x_{j-1}) \approx \nabla f(x_j) - \nabla f(x_{j-1})$ in a local small region around $x_k$, so it is reasonable to assume $R_k \approx -\nabla^2 f(x_k) X_k$. Thus we see $\|r_k - R_k\Gamma\|_2 \approx \|r_k + \nabla^2 f(x_k) X_k \Gamma\|_2$. Hence, we can recognize (5) as solving $\nabla^2 f(x_k) p_k = \nabla f(x_k)$ in a least-squares sense, where $p_k = X_k\Gamma_k$. When the quadratic approximation is exact, solving (5) is a minimal residual procedure, thus being verified as a* residual projection *method [47]. Moreover, let $H_k$ be the solution to the constrained optimization problem [14]:*

$$\min_{H_k} \|H_k - \beta_k I\|_F \text{ subject to } H_k R_k = -X_k, \quad (7)$$

*then iterate (6) is $x_{k+1} = x_k + H_k r_k$, which is indeed a multisecant quasi-Newton method. Note that a key simplification in AM is using differences of historical gradients $R_k$ to approximate $-\nabla^2 f(x_k) X_k$, which reduces the heavy cost to compute Hessian-vector products [7, 19, 26].*

### 2.2 Stochastic Anderson Mixing

We describe our method Stochastic Anderson Mixing (SAM) in this section. At the $k$-th iteration, let $S_k \subseteq [T] \stackrel{\text{def}}{=} \{1, 2, \ldots, T\}$ be the sampled mini-batch and the corresponding objective function value is $f_{S_k}(x_k) = \frac{1}{|S_k|} \sum_{i \in S_k} f_{\xi_i}(x_k)$. Then $r_k \stackrel{\text{def}}{=} -\nabla f_{S_k}(x_k)$ and the noisy $R_k$ is defined correspondingly (cf. (3)). Due to the instability and inaccurate estimation of $R_k$, we stabilize the projection step by proposing *damped projection* and *adaptive regularization* techniques. Algorithm 1 is a sketch of our method. Now, we elaborate the mechanism of this algorithm.

**Damped projection.** From Remark 1, we see the determination of $\Gamma_k$ in (5) relies on the local quadratic approximation of (2), which can be rather inexact in general nonlinear optimization. To improve the stability, we propose a *damped projection* method for (4a). Let $\alpha_k$ be the damping parameter, we obtain $\bar{x}_k$ via

$$\bar{x}_k = (1 - \alpha_k) x_k + \alpha_k (x_k - X_k \Gamma_k) = x_k - \alpha_k X_k \Gamma_k. \quad (8)$$

Combining (8) and (4b) and noting that $\bar{r}_k = r_k - \alpha_k R_k \Gamma_k$, the new iterate $x_{k+1}$ is given by

$$x_{k+1} = x_k + \beta_k r_k - \alpha_k (X_k + \beta_k R_k)\Gamma_k. \quad (9)$$

It is worth noting that $\beta_k$ and $\alpha_k$ in (9) behave like stepsize or learning rate in SGD, and the extra term $(\alpha_k X_k + \alpha_k \beta_k R_k)\Gamma_k$ can be viewed as a generalized momentum term.

**Algorithm 1** Stochastic Anderson Mixing (SAM)
___
**Input**: $x_0 \in \mathbb{R}^d, m = 10, \alpha_k = 1, \beta_k = 1, \mu \in (0,1), max\_iter > 0$.
**Output**: $x \in \mathbb{R}^d$
 1: **for** $k = 0, 1, \ldots, max\_iter$ **do**
 2:    $r_k = -\nabla f_{S_k}(x_k)$
 3:    **if** $k = 0$ **then**
 4:       $x_{k+1} = x_k + \beta_k r_k$
 5:    **else**
 6:       $m_k = \min\{m, k\}$
 7:       $X_k = [\Delta x_{k-m_k}, \Delta x_{k-m_k+1}, \cdots, \Delta x_{k-1}]$
 8:       $R_k = [\Delta r_{k-m_k}, \Delta r_{k-m_k+1}, \cdots, \Delta r_{k-1}]$
 9:       Check Condition (14) and use smaller $\alpha_k$ if (14) is violated.
10:       $x_{k+1} = x_k + \beta_k r_k - (\alpha_k X_k + \alpha_k \beta_k R_k) \left( R_k^{\mathrm{T}} R_k + \delta_k X_k^{\mathrm{T}} X_k \right)^{\dagger} R_k^{\mathrm{T}} r_k$
11:    **end if**
12:    Apply learning rate schedule of $\alpha_k, \beta_k$
13: **end for**
14: **return** $x_k$
___

**Adaptive regularization.** Since $R_k$ may be rank deficient and no safeguard method is used in AM, the least squares problem (5) can be unstable. A remedy is to add *regularization* [6] to (5). One well-known choice is the constant regularization introduced in [50, 53], which can be viewed as forcing $\|\Gamma_k\|_2$ to be small [53], leading to a penalty term to (5):

$$\Gamma_k = \underset{\Gamma \in \mathbb{R}^m}{\arg\min} \|r_k - R_k \Gamma\|_2^2 + \delta \|\Gamma\|_2^2, \tag{10}$$

where $\delta \geq 0$ is the penalty constant. The solution of (10) is $\Gamma_k = \left( R_k^{\mathrm{T}} R_k + \delta I \right)^{\dagger} R_k^{\mathrm{T}} r_k$, where "$\dagger$" denotes the Penrose-Moore inverse. This regularized variant of AM is named as RAM and serves as a baseline in the experiments.

Here, we propose a new regularization, namely *adaptive regularization*, to better suit the stochastic optimization. Since $-X_k \Gamma_k = \bar{x}_k - x_k$ denotes the update from $x_k$ to $\bar{x}_k$, a large magnitude of $\|X_k \Gamma_k\|_2$ tends to make the intermediate step $\bar{x}_k$ overshoot the trust region around $x_k$. Thus it is more reasonable to force $\|X_k \Gamma_k\|_2$ rather than $\|\Gamma_k\|_2$ to be small. We formulate this idea as

$$\min_{\Gamma} \|r_k - R_k \Gamma\|_2^2 + \delta_k \|X_k \Gamma\|_2^2, \tag{11}$$

where $\delta_k \geq 0$ is a variable determined in each iteration. Explicitly solving (11) leads to

$$\Gamma_k = \left( R_k^{\mathrm{T}} R_k + \delta_k X_k^{\mathrm{T}} X_k \right)^{\dagger} R_k^{\mathrm{T}} r_k. \tag{12}$$

We call AM with this regularization and damped projection as SAM, i.e. the prototype algorithm given in Algorithm 1.

**Positive definiteness.** From (9) and (12), the SAM update is $x_{k+1} = x_k + H_k r_k$, where $H_k = \beta_k I - \alpha_k Y_k Z_k^{\dagger} R_k^{\mathrm{T}}, Y_k = X_k + \beta_k R_k, Z_k = R_k^{\mathrm{T}} R_k + \delta_k X_k^{\mathrm{T}} X_k$. $H_k$ is generally not symmetric. A critical condition for the convergence analysis of SAM is the *positive definiteness* of $H_k$, i.e.

$$p_k^{\mathrm{T}} H_k p_k \geq \beta_k \mu \|p_k\|_2^2, \quad \forall p_k \in \mathbb{R}^d, \tag{13}$$

where $\mu \in (0,1)$ is a constant. Next, we give an approach to guarantee it.

Let $\lambda_{min}(\cdot)$ denote the smallest eigenvalue, $\lambda_{max}(\cdot)$ denote the largest eigenvalue. Since $p_k^{\mathrm{T}} H_k p_k = \frac{1}{2} p_k^{\mathrm{T}} (H_k + H_k^{\mathrm{T}}) p_k$, Condition (13) is equivalent to $\lambda_{min} \left( \frac{1}{2} \left( H_k + H_k^{\mathrm{T}} \right) \right) \geq \beta_k \mu$. With some simple algebraic operations, we obtain $\lambda_{min} \left( \frac{1}{2} \left( H_k + H_k^{\mathrm{T}} \right) \right) = \beta_k - \frac{1}{2} \alpha_k \lambda_{max} (Y_k Z_k^{\dagger} R_k^{\mathrm{T}} + R_k Z_k^{\dagger} Y_k^{\mathrm{T}})$. Let $\lambda_k \stackrel{\text{def}}{=} \lambda_{max} (Y_k Z_k^{\dagger} R_k^{\mathrm{T}} + R_k Z_k^{\dagger} Y_k^{\mathrm{T}})$, then Condition (13) is equivalent to

$$\alpha_k \lambda_k \leq 2\beta_k (1 - \mu). \tag{14}$$

To check Condition (14), note that

$$\lambda_k = \lambda_{max} \left( \begin{pmatrix} Y_k & R_k \end{pmatrix} \begin{pmatrix} 0 & Z_k^{\dagger} \\ Z_k^{\dagger} & 0 \end{pmatrix} \begin{pmatrix} Y_k^{\mathrm{T}} \\ R_k^{\mathrm{T}} \end{pmatrix} \right) = \lambda_{max} \left( \begin{pmatrix} Y_k^{\mathrm{T}} \\ R_k^{\mathrm{T}} \end{pmatrix} \begin{pmatrix} Y_k & R_k \end{pmatrix} \begin{pmatrix} 0 & Z_k^{\dagger} \\ Z_k^{\dagger} & 0 \end{pmatrix} \right). \tag{15}$$

**Algorithm 2** Stochastic Anderson Mixing with variance reduction (SAM-VR)

---

**Input**: $\tilde{x}_0 \in \mathbb{R}^d$; $\beta_t^k, \alpha_t^k, \delta_t^k$ for $SAM\_update(x_t^k, g_t^k)$; Batch size $n \geq 1$ .
**Output**: $x \in \mathbb{R}^d$

1: **for** $k = 0, \ldots, N - 1$ **do**
2:     $x_0^k = \tilde{x}_k$
3:     $\nabla f(\tilde{x}_k) = \frac{1}{T} \sum_{i=1}^{T} \nabla f_{\xi_i}(\tilde{x}_k)$
4:     **for** $t = 0, \ldots, q - 1$ **do**
5:        Sample a subset $\mathcal{K} \subseteq [T]$ with $|\mathcal{K}| = n$
6:        $g_t^k = \nabla f_{\mathcal{K}}(x_t^k) - \nabla f_{\mathcal{K}}(\tilde{x}_k) + \nabla f(\tilde{x}_k)$ where $\nabla f_{\mathcal{K}}(x_t^k) = \frac{1}{|\mathcal{K}|} \sum_{i \in \mathcal{K}} \nabla f_{\xi_i}(x_t^k)$
7:        $x_{t+1}^k = SAM\_update(x_t^k, g_t^k)$
8:     **end for**
9:     Set $\tilde{x}_{k+1} = x_q^k$
10: **end for**
11: **return** Iterate $x$ chosen uniformly random from $\{\{x_t^k\}_{t=0}^{q-1}\}_{k=0}^{N-1}$

---

Since $\begin{pmatrix} Y_k^{\mathrm{T}} \\ R_k^{\mathrm{T}} \end{pmatrix} (Y_k \quad R_k)$, $\begin{pmatrix} 0 & Z_k^{\dagger} \\ Z_k^{\dagger} & 0 \end{pmatrix} \in \mathbb{R}^{2m \times 2m}$, and $m \ll d$, $\lambda_k$ can be computed efficiently, say, using an eigenvalue decomposition algorithm with the time complexity of $O(m^3)$. This cost is negligible compared with those to form $X_k^{\mathrm{T}} X_k, R_k^{\mathrm{T}} R_k$. After that, to guarantee the positive definiteness, we check if $\alpha_k$ satisfies (14) and use a smaller $\alpha_k$ if necessary.

**AdaSAM method.** We choose the $\delta_k$ in (11) as

$$\delta_k = \max \left\{ \frac{c_1 \|r_k\|_2^2}{\|x_k - x_{k-1}\|_2^2 + \epsilon}, c_2 \beta_k^{-2} \right\}, \tag{16}$$

where $c_1, c_2, \epsilon$ are constants. Such choice reflects the curvature change in the vicinity of $x_k$. A large $\|r_k\|_2$ indicates a potential dramatic change in landscape, suggesting using a precautious tiny stepsize. The denominator in (16) behaves like *annealing*, which can measure the noise in gradients like that in secant penalized BFGS [27].

## 2.3 Enhancement of Stochastic Anderson Mixing

We introduce the *variance reduction* and *preconditioned mixing* techniques to further enhance SAM.

**Variance reduction.** Variance reduction techniques turn out to be effective if a scan over the full dataset is feasible [1, 44]. Similar to SdLBFGS-VR proposed in [60], we also incorporate SVRG to SAM, which we call SAM-VR (Algorithm 2), for solving (2). To simplify the description, we denote one iteration of SAM in Algorithm 1 as $SAM\_update(x_k, g_k)$, i.e. one update of $x_k$ given the gradient estimate $g_k$.

**Preconditioned mixing.** Motivated by the great success of preconditioning in solving linear systems and eigenvalue computation [18], we present a preconditioned version of SAM. The key modification is the mixing step (4b). We replace the simple mixing $x_{k+1} = \bar{x}_k + \beta_k \bar{r}_k$ with $x_{k+1} = \bar{x}_k + M_k^{-1} \bar{r}_k$ where $M_k$ approximates the Hessian. Combining it with (8) and (12), we obtain

$$x_{k+1} = x_k + \left( M_k^{-1} - \alpha_k \left( X_k + M_k^{-1} R_k \right) \left( R_k^{\mathrm{T}} R_k + \delta_k X_k^{\mathrm{T}} X_k \right)^{\dagger} R_k^{\mathrm{T}} \right) r_k. \tag{17}$$

Setting $\alpha_k \equiv 1$ and $\delta_k \equiv 0$, (17) reduces to a preconditioned AM update, which can be recast as the solution to the constrained optimization problem: $\min_{H_k} \|H_k - M_k^{-1}\|_F$ subject to $H_k R_k = -X_k$, a direct extension of (7). This preconditioned version of AM is related to quasi-Newton updates [20]. We also point out that the action of $M_k^{-1}$ can be implicitly done via an update of any optimizer at hand, i.e. $x_{k+1} = optim(\bar{x}_k, -\bar{r}_k)$, where $optim$ updates $\bar{x}_k$ given the *extragradient* $-\bar{r}_k$. If $R_k = -\nabla^2 f(x_k) X_k$, which is the case in deterministic quadratic optimization, the projection step in preconditioned AM is still a minimal residual procedure.

**Remark 2.** *The same as SdLBFGS [60], SAM needs another $2md$ space to store $X_k$ and $R_k$. The extra main computational cost for SAM compared with SGD is $O(m^2 d) + O(m^3)$, which accounts for the matrix multiplications ($\mathbb{R}^{m \times d} \times \mathbb{R}^{d \times m}$) and matrix decomposition of a small $\mathbb{R}^{m \times m}$ matrix.*

*Since dense matrix multiplication can be ideally parallelized and the cost of gradient evaluations often dominates the computing, the benefit from SAM pays for this extra cost. Also, this cost can be further reduced to $O(md) + O(m^3)$ (see Appendix B.2). In our implementation, we incorporate sanity check of the positive definiteness, alternating iteration and moving average, and the details are given in the supplementary materials.*

## 3    Theory

In this section, we analyze the convergence and complexity of SAM. All the proofs are deferred to the Appendix A.1. We first give two assumptions about the objective function $f$ that are widely used in stochastic programming [17, 60, 44].

**Assumption 1.** $f : \mathbb{R}^d \to \mathbb{R}$ *is continuously differentiable.* $f(x) \geq f^{low} > -\infty$ *for any* $x \in \mathbb{R}^d$. $\nabla f$ *is globally L-Lipschitz continuous; namely* $\|\nabla f(x) - \nabla f(y)\|_2 \leq L\|x - y\|_2$ *for any* $x, y \in \mathbb{R}^d$.

**Assumption 2.** *For any iteration $k$, the stochastic gradient $\nabla f_{\xi_k}(x_k)$ satisfies* $\mathbb{E}_{\xi_k}[\nabla f_{\xi_k}(x_k)] = \nabla f(x_k)$, $\mathbb{E}_{\xi_k}[\|\nabla f_{\xi_k}(x_k) - \nabla f(x_k)\|_2^2] \leq \sigma^2$, *where $\sigma > 0$, and $\xi_k, k = 0, 1, \ldots$, are independent samples that are independent of $\{x_j\}_{j=0}^k$.*

Throughout our analysis, we always assume both Assumptions 1 and 2 hold. Moreover, we state the following conditions about the three parameters $\alpha_k, \beta_k, \delta_k$ of SAM.

$$\sum_{k=0}^{+\infty} \beta_k = +\infty, \quad \sum_{k=0}^{+\infty} \beta_k^2 < +\infty, \tag{18a}$$

$$\delta_k \geq C\beta_k^{-2}, \quad \alpha_k \in [0, \min\{1, \beta_k^{\frac{1}{2}}\}] \text{ and statisfies (14)}, \tag{18b}$$

where $C > 0$ is a constant.

**Convergence and complexity.**    We give the convergence of SAM by analyzing the iterations $\{x_k\}$ generated by Algorithm 1.

**Theorem 1.** *Let $\{x_k\}$ be the sequence generated by Algorithm 1 with fixed batchsize $n$. Given a positive constant $C$, if $\beta_k \in (0, \frac{\mu}{4L(1+C^{-1})}]$, then under the conditions* (18a) *and* (18b)*, it has*

$$\liminf_{k \to \infty} \|\nabla f(x_k)\|_2 = 0 \text{ with probability } 1. \tag{19}$$

*Moreover, there exists a positive constant $M_f$ such that*

$$\mathbb{E}[f(x_k)] \leq M_f \quad \forall k. \tag{20}$$

**Theorem 2.** *Under the same conditions as Theorem 1, if we require that the noisy gradient is bounded, i.e.,*

$$\mathbb{E}_{\xi_k}[\|\nabla f_{\xi_k}(x_k)\|_2^2] \leq M_g, \tag{21}$$

*where $M_g > 0$ is a constant, we have*

$$\lim_{k \to \infty} \|\nabla f(x_k)\|_2 = 0 \text{ with probability } 1. \tag{22}$$

Now, we give the iteration complexity of SAM.

**Theorem 3.** *Let $\{x_k\}$ be the sequence generated by Algorithm 1 with batchsize $n$. Given constants $C > 0$ and $r \in (0.5, 1)$, we choose $\beta_k = \frac{\mu}{4L(1+C^{-1})}(k+1)^{-r}$. Under the condition* (18b)*, it has*

$$\frac{1}{N} \sum_{k=0}^{N-1} \mathbb{E}\|\nabla f(x_k)\|_2^2 \leq \frac{16L(1+C^{-1})(M_f - f^{low})}{\mu^2} N^{r-1} + \frac{(1+L^{-1}\mu^{-1})\sigma^2}{(1-r)n}(N^{-r} - rN^{-1}), \tag{23}$$

*where $n$ is the batchsize, and $N$ is the iteration number. Moreover, for a given $\epsilon \in (0, 1)$, to guarantee that $\frac{1}{N} \sum_{k=0}^{N-1} \mathbb{E}\|\nabla f(x_k)\|_2^2 < \epsilon$, the number of iterations $N$ needed is at most $O(\epsilon^{-\frac{1}{1-r}})$.*

If the output $x_R$ is randomly chosen from previous iterates according to the certain probability distribution, SAM has the same complexity $O(\epsilon^{-2})$ as RSG [17] and SQN [60].

**Theorem 4.** *Let $\{x_k\}$ be the sequence generated by Algorithm 1 with batchsize $n$ and $x_R$ be the random output of the first $N$ iterations where the random variable $R$ follows the uniform distribution $P_R(k) \stackrel{\text{def}}{=} Prob\{R = k\} = 1/N$. Given constant $C > 0$, we choose $\beta_k = \frac{\mu}{4L(1+C^{-1})}$ and assume condition* (18b) *holds. For any $\epsilon > 0$, there exist some $n$ and $N = O(\epsilon^{-2})/n$ such that we have $\mathbb{E}\left[\|\nabla f(x_R)\|_2^2\right] \leq \epsilon$, where the expectation is taken with respect to $R$ and $\{S_j\}_{j=0}^{N-1}$.*

In fact, from the proof of Theorem 4, we can find the explicit choice of the batchsize. Define $D_f \stackrel{\text{def}}{=} f(x_0) - f^{low}$ and $\tilde{D}$ is a problem-independent positive constant. The batchsize $n$ can be chosen as $n = \left\lceil \min\left\{\bar{N}, \max\left\{1, \frac{\sigma}{L}\sqrt{\frac{\bar{N}}{\tilde{D}}}\right\}\right\}\right\rceil$, where $\bar{N}$ is the total number of $\mathcal{SFO}$-calls satisfying $\bar{N} \geq \max\left\{\frac{C_1^2}{\epsilon^2} + \frac{4C_2}{\epsilon}, \frac{\sigma^2}{L^2\tilde{D}}\right\}$, and $C_1 = \frac{32D_f(1+C^{-1})\sigma}{\mu^2\sqrt{\tilde{D}}} + (L+\mu^{-1})\sigma\sqrt{\tilde{D}}, C_2 = \frac{32D_f L(1+C^{-1})}{\mu^2}$.
Thus, to ensure $\mathbb{E}\left[\|\nabla f(x_R)\|_2^2\right] \leq \epsilon$, the total number of $\mathcal{SFO}$-calls is $O(\epsilon^{-2})$. Moreover, we give the $\mathcal{SFO}$-calls complexity of the variance reduced version of SAM.

**Theorem 5.** *Let $x$ be the output of Algorithm 2 with batchsize $n$. Given a positive constant $C$, suppose $\nu, \mu_0 \in (0, 1)$ to be two constants satisfying*

$$\frac{\mu_0\mu}{2(1+C^{-1})^{1/2}} - \frac{\mu_0^2(2L+\mu^{-1})(e-1)}{L} - 2\mu_0^2 n - \frac{2\mu_0^3(2L+\mu^{-1})(e-1)n}{L} \geq \nu,$$

*where $e$ is the Euler's number. Set $\beta_t^k = \beta := \frac{\mu_0 n}{L(1+C^{-1})^{1/2}T^{2/3}}, \delta_t^k \geq C\beta^{-2}, \alpha_t^k \in [0, \min\{1, \beta^{\frac{1}{2}}\}]$ and satisfies* (14)*, and the number of iteration $q = \left\lfloor \frac{T}{\mu_0 n d_0}\right\rfloor$, where $d_0 = 6 + \frac{2}{L(1+C^{-1})^{1/2}}$. Then,*

$$\mathbb{E}\left[\|\nabla f(x)\|_2^2\right] \leq \frac{T^{2/3}L\left(f(x_0) - f^{low}\right)}{qNn\nu}. \tag{24}$$

*To ensure $\mathbb{E}\left[\|\nabla f(x)\|_2^2\right] \leq \epsilon$, the total number of $\mathcal{SFO}$-calls is $O(T + (T^{2/3}/\epsilon))$.*

**Remark 3.** *It is worth noting that the approximated Hessian $H_k$ in SAM depends on the data samples $\{\xi_i\}_{i\in S_k}$ of the current mini-batch, which violates the assumption AS.4 in* [60]*.*

**Analysis of AdaSAM.** All the results in Theorems 1-5 are applicable for AdaSAM as (16) satisfies the condition (18b). We further discuss the rationality of the first term in (16). Since $r_k^{\mathrm{T}} H_k r_k \geq \beta_k\mu\|r_k\|_2^2$, and $\|H_k r_k\|_2^2 \leq 2\left(\beta_k^2 + \alpha_k^2\delta_k^{-1}\right)\|r_k\|_2^2 \leq 2\beta_k^2\left(1 + C^{-1}\right)\|r_k\|_2^2$ if $\alpha_k \leq 1, \delta_k \geq C\beta_k^{-2}$ (see Corollary 1 in Appendix A.1), it is sensible to suppose $\|x_k - x_{k-1}\|_2 \approx \|x_{k+1} - x_k\|_2 = \|H_k r_k\|_2 = \beta_k h_k\|r_k\|_2$, where $h_k \in (0, h)$ $(h < +\infty)$ is related to $H_k$. Therefore,

$$\frac{c_1\|r_k\|_2^2}{\|x_k - x_{k-1}\|_2^2} \approx \frac{c_1\|r_k\|_2^2}{\beta_k^2 h_k^2\|r_k\|_2^2} \geq c_1 h^{-2}\beta_k^{-2}, \tag{25}$$

which coincides with the requirement that $\delta_k \geq C\beta_k^{-2}$. A further discussion is in Appendix D.3.

## 4 Experiments

We used AdaSAM (AdaSAM-VR) for the experiments, which is a special case of SAM (SAM-VR) with $\delta_k$ specified as (16). The pseudocodes are given in Appendix B.1. To evaluate the effectiveness of our methods, we compared them with several first-order and second-order optimizers for mini-batch training of neural networks on different machine learning tasks. The datasets were MNIST [32], CIFAR-10/CIFAR-100 [31] for image classification and Penn Treebank [35] for language model. The experimental details and hyper-parameter tuning are referred to Appendix C.

**Experiments on MNIST.** We trained a simple convolutional neural network (CNN) [3] on MNIST, for which we were only concerned about the minimization of the empirical risk (2), i.e. the training loss, with large batch sizes. The training dataset was preprocessed by randomly selecting 12k images from the total 60k images to facilitate large mini-batch training. Neither weight-decay nor dropout was used. We compared AdaSAM with SGDM [43] (SGD with momentum), Adam [29], SdLBFGS [60], and RAM (cf. (10)). The learning rate was tuned and fixed for each optimizer. The historical lengths for SdLBFGS, RAM and AdaSAM were set as 20. $\delta = 10^{-6}$ for RAM and $c_1 = 10^{-4}$ for AdaSAM.

---

[3]Based on the official PyTorch implementation https://github.com/pytorch/examples/blob/master/mnist.

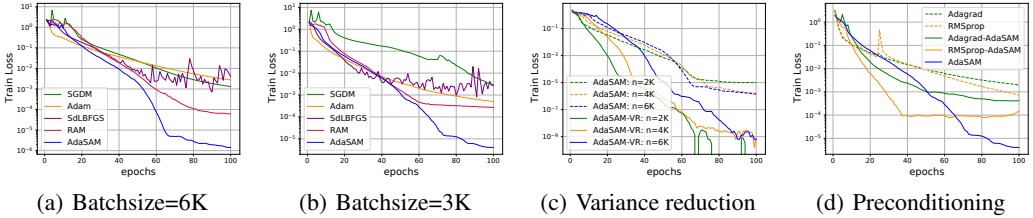

| (a) Batchsize=6K | (b) Batchsize=3K | (c) Variance reduction | (d) Preconditioning |

Figure 1: Experiments on MNIST. (a) Train Loss using batchsize = 6K; (b) Train Loss using batchsize = 3K; (c) AdaSAM with variance reduction; (d) Preconditioned AdaSAM with batchsize = 3K.

Figure 1 (a) and (b) show the curves of training loss when training 100 epochs with batch sizes of 6K and 3K, which indicate that AdaSAM can significantly minimize the empirical risk in large mini-batch training. The comparison with RAM verifies the benefit of adaptive regularization. We notice that there hardly exists any oscillation in AdaSAM during training except for the first few epochs, which suggests that AdaSAM is tolerant to noise. We also tested the effectiveness of variance reduction and preconditioning introduced in Section 2.3. The variance reduced extension AdaSAM-VR was compared with AdaSAM for different batch sizes. The variants of AdaSAM preconditioned by Adagrad [12] and RMSprop [55] are denoted as Adagrad-AdaSAM and RMSprop-AdaSAM, respectively. Though AdaSAM-VR is more costly per iteration and the preconditioned variants seem to deteriorate the final training loss, we point out that AdaSAM-VR can achieve lower training loss ($10^{-9}$) and the preconditioned variants converge faster to an acceptable training loss (e.g. $10^{-3}$).

Table 1: Final TOP1 test accuracy (mean ± standard deviation) (%) on CIFAR10/CIFAR100. The bold numbers highlight the best results. WideResNet is abbreviated as WResNet.

| Method | CIFAR10 | | | | | | CIFAR100 | | |
|---|---|---|---|---|---|---|---|---|---|
| | ResNet18 | ResNet20 | ResNet32 | ResNet44 | ResNet56 | WResNet | ResNet18 | ResNeXt | DenseNet |
| SGDM | 94.82±.15 | 92.03±.16 | 92.86±.15 | 93.10±.23 | 93.47±.28 | 94.90±.09 | 77.27±.09 | 78.41±.54 | 78.49±.12 |
| Adam | 93.03±.07 | 91.17±.13 | 92.03±.28 | 92.28±.62 | 92.39±.23 | 92.45±.11 | 72.41±.17 | 73.57±.17 | 70.80±.23 |
| AdaBelief | 94.65±.13 | 91.15±.21 | 92.15±.17 | 92.79±.24 | 93.30±.07 | 94.46±.13 | 76.25±.06 | 78.27±.16 | 78.83±.15 |
| Lookahead | 94.92±.33 | 92.07±.04 | 92.86±.15 | 93.26±.24 | 93.36±.13 | 94.90±.15 | 77.63±.35 | 78.93±.12 | 79.37±.16 |
| AdaHessian | 94.36±.09 | 91.92±.32 | 92.18±.18 | 92.74±.11 | 92.40±.06 | 94.04±.12 | 76.59±.42 | - | - |
| RNA | 93.45±.21 | 90.73±.12 | 91.08±.51 | 91.61±.37 | 91.23±.14 | 93.85±.24 | 75.12±.39 | 75.88±.40 | 75.70±.49 |
| RAM | 95.10±.05 | 92.21±.09 | 93.05±.43 | 93.42±.13 | 93.76±.16 | 95.04±.09 | 76.19±.12 | 78.65±.20 | 78.28±.62 |
| AdaSAM | **95.17±.10** | **92.43±.19** | **93.22±.32** | **93.57±.14** | **93.77±.12** | **95.23±.07** | **78.13±.14** | **79.31±.27** | **80.09±.52** |

**Experiments on CIFAR**. For CIFAR-10 and CIFAR-100, both datasets have 50K images for training and 10K images for test. The test accuracy at the final epoch was reported as the evaluation metric. We trained ResNet18/20/32/44/56 [21] and WideResNet16-4 [64] on CIFAR-10, and ResNet18, ResNeXt50 [61] and DenseNet121 [24] on CIFAR-100. The baseline optimizers were SGDM, Adam, AdaBelief [66], Lookahead [65], AdaHessian [63] and RNA [52]. The hyperparameters were kept unchanged across different tests. We trained 160 epochs with batch size of 128 and decayed the learning rate at the 80th and 120th epoch. For AdaSAM/RAM, $\alpha_k$ and $\beta_k$ were decayed at the 80th and 120th epoch.

Table 1 demonstrates the generalization ability of AdaSAM (AdaHessian ran out of memory for training ResNeXt50 and DenseNet121). It shows AdaSAM consistently outperforms other optimizers for various neural

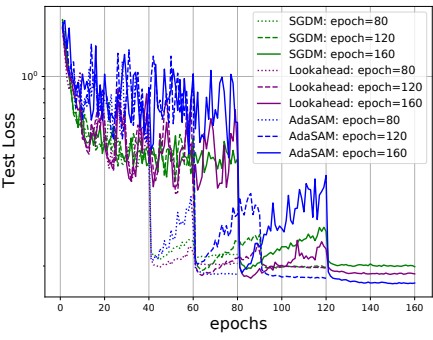

Figure 2: Test loss of training ResNet18 on CIFAR-10.

networks on CIFAR10/CIFAR100. Compared with SGDM/Lookahead, AdaSAM is built on a noisy quadratic model to extrapolate historical iterates more elaborately, which may account for its effectiveness. We also conducted tests on training for 120 epochs and 80 epochs. Figure 2 shows AdaSAM

Table 2: The memory and computation cost compared with SGDM and AdaHessian. Memory, per-epoch time and total running time are abbreviated as "m","t/e" and "t", respectively.

| Cost ($\times$ SGDM) | CIFAR10/ResNet20 | | | CIFAR10/WResNet | | | CIFAR100/ResNeXt50 | | | CIFAR100/DenseNet121 | | |
|---|---|---|---|---|---|---|---|---|---|---|---|---|
| | m | t/e | t | m | t/e | t | m | t/e | t | m | t/e | t |
| SGDM | 1.00 | 1.00 | 1.00 | 1.00 | 1.00 | 1.00 | 1.00 | 1.00 | 1.00 | 1.00 | 1.00 | 1.00 |
| AdaHessian | 1.78 | 3.59 | 3.59 | 2.35 | 5.97 | 5.97 | >1.6 | - | - | >1.8 | - | - |
| AdaSAM | 1.14 | 1.34 | 0.83 | 1.26 | 1.28 | 0.80 | 1.30 | 1.16 | 0.58 | 1.16 | 1.19 | 0.60 |

can achieve comparable or even lower test loss than SGDM/Lookahead when training with fewer epochs, thus saving a large number of iterations. More results can be found in Appendix C.3.

To compare the total training time, we set SGDM as the baseline and terminated the training of AdaSAM when the accuracy difference AdaSAM − SGDM > −0.05%. The memory, per-epoch computational cost and total training time of training ResNet20/WideResNet16-4 (abbr. WResNet) on CIFAR10 and ResNeXt50/DenseNet121 on CIFAR100 are reported in Table 2. It shows AdaSAM is more efficient than AdaHessian and saves more than 15% training time than SGDM in these models.

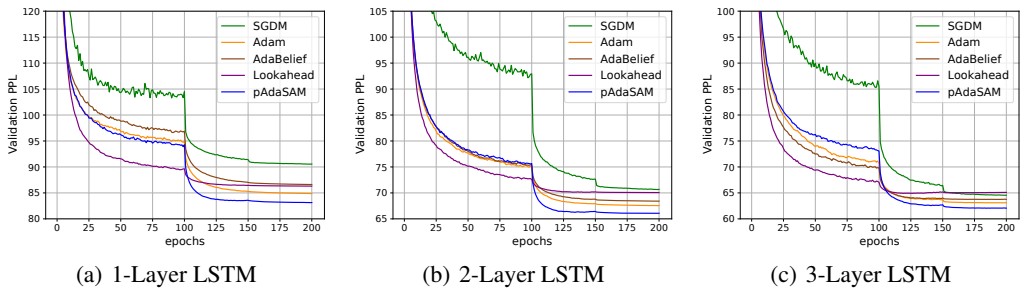

| (a) 1-Layer LSTM | (b) 2-Layer LSTM | (c) 3-Layer LSTM |
|---|---|---|

Figure 3: Experiments on Penn Treebank. Validation perplexity of training 1,2,3-Layer LSTM.

**Experiments on Penn Treebank**. We trained LSTM [22] on Penn Treebank and reported the perplexity on the validation set in Figure 3 and perplexity on the test set in Table 3, where pAdaSAM denotes the variant of AdaSAM preconditioned by Adam. The experimental setting was the same as that in AdaBelief [66]. In our practice, we found that the vanilla AdaSAM with default hyperparameter setting is not suitable for this task. Nevertheless, a suitable preconditioner (e.g. Adam) can largely

Table 3: Test perplexity on Penn Treebank for 1,2,3-layer LSTM. Lower is better.

| Method | 1-Layer | 2-Layer | 3-Layer |
|---|---|---|---|
| SGDM | 85.21±.36 | 67.12±.14 | 61.56±.14 |
| Adam | 80.88±.15 | 64.54±.18 | 60.34±.22 |
| AdaBelief | 82.41±.46 | 65.07±.02 | 60.64±.14 |
| Lookahead | 82.01±.07 | 66.43±.33 | 61.80±.10 |
| pAdaSAM | **79.34±.09** | **63.18±.22** | **59.47±.08** |

improve the behaviour of AdaSAM. Conversely, AdaSAM can also enhance a optimizer when the latter is used as a preconditioner.

## 5 Discussion

In this section, we provide discussions that compare SAM with SQN [60] and SGD.

**SAM versus SQN**. Our theoretical results are similar to those of SQN [60], but there are some fundamental differences:

(i) The basic assumptions are different. (a) Recall that the update is $x_{k+1} = x_k - H_k g_k$, where $g_k = \nabla f_{S_k}(x_k)$. Let $S_{[k-1]} := (S_0, \ldots, S_{k-1})$ denote the random samplings in the first $k$ iterations. A key assumption in [60] is that $H_k$ depends only on $S_{[k-1]}$, which guarantees $\mathbb{E}[H_k g_k | S_{[k-1]}] = H_k \nabla f(x_k)$ that can simplify the proof. In contrast, in our methods SAM and SAM-VR, $H_k$ depends on $S_k$ since $r_k = -g_k$ participates in constructing $H_k$, thus violating this assumption and complicating our analysis. (b) The proposed algorithm SdLBFGS in [60] needs twice continuously differentiable assumption of $f$ and the boundedness of $\nabla^2 f$, which are not required in our algorithm.

(ii) The ways to construct the secant equations are different. In the deterministic case, a secant equation is $H_k s_k = y_k$, where $s_k = x_k - x_{k-1}$ and $y_k = g_k - g_{k-1}$. In the stochastic case, various choices exist in forming $y_k$. For SAM, the choice of $y_k$ is the same as that of the deterministic case, thus making $H_k$ depend on $S_k$. For the SQN methods in [60], $y_k = \nabla f_{S_{k-1}}(x_k) - \nabla f_{S_{k-1}}(x_{k-1})$, which ensures $H_k$ is independent of $S_k$. Such choice can reduce variance of noise in secant equations but demands evaluating gradients on the same mini-batch $S_{k-1}$ twice. Some other SQN methods [7] use subsampled Hessian-vector products to form $y_k = \nabla^2 f_{S'_k}(x_k - x_{k-1})$, where $S'_k$ denotes the data samples independent of $S_k$, which also ensures $H_k$ is independent of $S_k$.

**SAM versus SGD**. We compare SAM with SGD from theoretical and practical perspectives.

(i) Theoretical perspective. (a) For general nonconvex stochastic optimization, SAM achieves the same worst-case complexity $O(1/\epsilon^2)$ as SGD. (b) There are some specific cases that SAM can be theoretically better than SGD. For strongly convex quadratic minimization, we prove that SAM is essentially equivalent to GMRES (see Appendix A.2) that can exhibit superlinear convergence [58]. For nonlinear optimization, SAM can also have superlinear convergence rate in some cases as it is closely related to the multisecant quasi-Newton method [14], as also pointed out in Remark 1.

(ii) Practical perspective. (a) With the same number of epochs as SGD, it is shown in Table 1 and Table 3 that our method has better generalization performance than SGD in both image classification and language tasks. (b) Our method can largely reduce the number of epochs to achieve a comparable solution as SGD, as shown in Figure 2 in Section 4 and Figure 8 in Appendix C.3. The total running time can reduce correspondingly, as shown in Table 2. (c) SAM needs additional memory and computational cost, but it is still much more efficient and economical than other second-order optimizers, as shown in Table 2 in Section 4 and Figure 22 in Appendix D.8.

Table 4: Comparison between SAM and SGD. '+' ('-') means SAM is better (worse) than SGD, and '=' means SAM is similar to SGD.

| Criterion | general | specific | memory | time/epochs | epochs | time | generalization |
|---|---|---|---|---|---|---|---|
| SAM vs. SGD | = | + | - | - | + | + | + |

Based on the above analysis, we summarize the comparisons of our method and SGD in Table 4. There are 7 types of criterion. "general": the worse-case complexity in nonconvex stochastic optimization; "specific": some specific problems; "memory": memory usage; "time/epochs": per-epoch running time; "epochs" and "time": the total epochs and running time to achieve comparable test performance as SGD; "generalization": test performance with the same number of epochs as SGD.

# 6 Conclusion

In this paper, we develop an extension of Anderson mixing, namely Stochastic Anderson Mixing, for nonconvex stochastic optimization. By introducing damped projection and adaptive regularization, we establish the convergence theory of our new method. We also analyze its work complexity in terms of $\mathcal{SFO}$-calls and show it can achieve the $O(1/\epsilon^2)$ complexity for an $\epsilon$-accurate solution. We also give a specific form of adaptive regularization. Then we propose two techniques to further enhance our method. One is the variance reduction technique, which can further improve the work complexity of our method theoretically and help achieve lower empirical risk in our experiments. The other one is the preconditioned mixing strategy that directly extends Anderson mixing. Experiments show encouraging results of our method and its enhanced versions in terms of convergence rate or generalization ability in training different neural networks in different machine learning tasks. These results confirm the suitability of Anderson mixing for nonconvex stochastic optimization.

## Acknowledgments and Disclosure of Funding

This work was supported by the National Key R&D Program of China (No. 2018YFB1005103), Technology and Innovation Major Project (No. 2020AAA0108403), National Natural Science Foundation of China (No.61925601, No. 61772302, No. 11901338) and Huawei Noah's Ark Lab. We thank all anonymous reviewers for their valuable comments and suggestions on this work.

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
