# A Proofs

## A.1 Nonconvex stochastic optimization

We give proofs of the theorems in section 3.

From Assumption 2, for the mini-batch gradient $f_{S_k}(x_k) = \frac{1}{n_k} \sum_{i \in S_k} f_{\xi_i}(x_k)$, where $n_k = |S_k|$, we have

$$\mathbb{E}[\nabla f_{S_k}(x)|x_k] = \nabla f(x_k), \tag{26a}$$

$$\mathbb{E}[\|\nabla f_{S_k}(x_k) - \nabla f(x_k)\|_2^2|x_k] \leq \frac{\sigma^2}{n_k}. \tag{26b}$$

Note that the update of SAM (9) can be written as $x_{k+1} = x_k + H_k r_k$, where $r_k = -\nabla f_{S_k}(x_k)$, $H_0 = \beta_0 I$ and for $k \geq 1$,

$$H_k = \beta_k I - (\alpha_k X_k + \alpha_k \beta_k R_k) \left(R_k^\mathrm{T} R_k + \delta_k X_k^\mathrm{T} X_k\right)^{-1} R_k^\mathrm{T}. \tag{27}$$

Theorems 1-5 state the same convergence and complexity results as the ones proved in [60]. To prove these theorems, the critical points are (i) the positive definiteness of the approximate Hessian $H_k$ and (ii) an adequate suppression of the noise in the gradient estimation.

We first give some lemmas.

**Lemma 1.** *Suppose that $\{x_k\}$ is generated by SAM. If $\alpha_k \geq 0, \beta_k > 0, \delta_k > 0$, then for any $v_k \in \mathbb{R}^d$, we have*

$$\|H_k v_k\|_2^2 \leq 2 \left(\beta_k^2 \left(1 + 2\alpha_k^2 - 2\alpha_k\right) + \alpha_k^2 \delta_k^{-1}\right) \|v_k\|_2^2. \tag{28}$$

*Proof.* The result clearly holds when $k = 0$ as $H_0 = \beta_0 I$. For $k \geq 1$,

$$H_k v_k = \beta_k v_k - (\alpha_k X_k + \alpha_k \beta_k R_k)\Gamma_k, \tag{29}$$

where

$$\Gamma_k = \min_\Gamma \|v_k - R_k \Gamma\|_2^2 + \delta_k \|X_k \Gamma\|_2^2. \tag{30}$$

Taking $\Gamma = 0$, we have

$$\|v_k - R_k \Gamma_k\|_2^2 + \delta_k \|X_k \Gamma_k\|_2^2 \leq \|v_k\|_2^2. \tag{31}$$

Therefore,

$$\begin{aligned}
&\|H_k v_k\|_2^2 \\
&= \|\beta_k v_k - (\alpha_k X_k + \alpha_k \beta_k R_k)\Gamma_k\|_2^2 \\
&= \|\beta_k (v_k - \alpha_k R_k \Gamma_k) - \alpha_k X_k \Gamma_k\|_2^2 \\
&= \|\beta_k (1 - \alpha_k) v_k + \beta_k \alpha_k (v_k - R_k \Gamma_k) - \alpha_k \delta_k^{-\frac{1}{2}} \delta_k^{\frac{1}{2}} X_k \Gamma_k\|_2^2 \\
&\leq \left(\beta_k^2 (1 - \alpha_k)^2 + \beta_k^2 \alpha_k^2 + \alpha_k^2 \delta_k^{-1}\right) \cdot \left(\|v_k\|_2^2 + \|v_k - R_k \Gamma_k\|_2^2 + \delta_k \|X_k \Gamma_k\|_2^2\right) \\
&\leq \left(\beta_k^2 \left(1 + 2\alpha_k^2 - 2\alpha_k\right) + \alpha_k^2 \delta_k^{-1}\right) \left(\|v_k\|_2^2 + \|v_k\|_2^2\right) \\
&= 2 \left(\beta_k^2 \left(1 + 2\alpha_k^2 - 2\alpha_k\right) + \alpha_k^2 \delta_k^{-1}\right) \|v_k\|_2^2. \tag{32}
\end{aligned}$$

In the above, the first inequality uses the inequality

$$\| \sum_{i=1}^n a_i \mathbf{x_i}\|_2^2 \leq \left(\sum_{i=1}^n |a_i| \|\mathbf{x_i}\|_2\right)^2 \leq \left(\sum_{i=1}^n a_i^2\right) \left(\sum_{i=1}^n \|\mathbf{x_i}\|_2^2\right), \tag{33}$$

where $a_i \in \mathbb{R}, x_i \in \mathbb{R}^d$. The second inequality is based on inequality (31). $\qquad\square$

**Lemma 2.** *Suppose that Assumption 2 holds for $\{x_k\}$ generated by SAM. In addition, if $\beta_k > 0, \delta_k > 0$, and $\alpha_k \geq 0$ and satisfies (14), then*

$$\mathbb{E}_{S_k}[\|H_k r_k\|_2^2] \leq 2 \left(\beta_k^2 \left(1 + 2\alpha_k^2 - 2\alpha_k\right) + \frac{\alpha_k^2}{\delta_k}\right) \cdot \left(\|\nabla f(x_k)\|_2^2 + \frac{\sigma^2}{n_k}\right), \tag{34a}$$

$$\nabla f(x_k)^\mathrm{T} \mathbb{E}_{S_k}[H_k r_k] \leq -\frac{1}{2} \beta_k \mu \|\nabla f(x_k)\|_2^2 + \frac{1}{2} \frac{\alpha_k^2 (\delta_k^{-\frac{1}{2}} + \beta_k)^2}{\beta_k \mu} \cdot \frac{\sigma^2}{n_k}, \tag{34b}$$

where $\mu > 0$ is the constant introduced in (13). If further assuming $H_k$ is independent of $S_k$, a better upper bound can be obtained:

$$\nabla f(x_k)^{\mathrm{T}} \mathbb{E}_{S_k}[H_k r_k] \leq -\beta_k \mu \|\nabla f(x_k)\|_2^2. \tag{35}$$

*Proof.* (i) From Lemma 1, we have

$$\mathbb{E}_{S_k}[\|H_k r_k\|_2^2] \leq 2\left(\beta_k^2\left(1 + 2\alpha_k^2 - 2\alpha_k\right) + \frac{\alpha_k^2}{\delta_k}\right)\mathbb{E}_{S_k}[\|r_k\|_2^2]. \tag{36}$$

From Assumption 2, we have

$$\mathbb{E}_{S_k}[\|r_k\|_2^2] = \mathbb{E}_{S_k}[\|r_k - \mathbb{E}_{S_k}[r_k]\|_2^2] + \|\mathbb{E}_{S_k}[r_k]\|_2^2 \leq \|\nabla f(x_k)\|_2^2 + \sigma^2/n_k. \tag{37}$$

With (36), (37), we obtain (34a).

(ii) Recalling that $H_0 = \beta_0 I$, the result holds for $k = 0$. Define $\epsilon_k = \nabla f_{S_k}(x_k) - \nabla f(x_k) = -r_k - \nabla f(x_k)$, then $H_k r_k = H_k\left(-\epsilon_k - \nabla f(x_k)\right)$. Since $\alpha_k$ satisfies (14), we can ensure $\lambda_{min}\left(\frac{1}{2}\left(H_k + H_k^{\mathrm{T}}\right)\right) \geq \beta_k \mu$. Thus

$$\nabla f(x_k)^{\mathrm{T}} H_k \nabla f(x_k) = \frac{1}{2}\nabla f(x_k)^{\mathrm{T}}\left(H_k + H_k^{\mathrm{T}}\right)\nabla f(x_k) \geq \beta_k \mu \|\nabla f(x_k)\|_2^2,$$

which implies

$$\mathbb{E}_{S_k}[\nabla f(x_k)^{\mathrm{T}} H_k \nabla f(x_k)] \geq \beta_k \mu \|\nabla f(x_k)\|_2^2. \tag{38}$$

Let $M_k = \alpha_k\left(X_k + \beta_k R_k\right)\left(R_k^{\mathrm{T}} R_k + \delta_k X_k^{\mathrm{T}} X_k\right)^\dagger R_k^{\mathrm{T}}$, then $H_k = \beta_k I - M_k$. With the assumption (26a), i.e. $\mathbb{E}_{S_k}[\epsilon_k] = 0$, we have

$$\begin{aligned}
\mathbb{E}_{S_k}[\nabla f(x_k)^{\mathrm{T}} H_k \epsilon_k] &= \mathbb{E}_{S_k}[\nabla f(x_k)^{\mathrm{T}}\left(\beta_k \epsilon_k - M_k \epsilon_k\right)] \\
&= \beta_k \nabla f(x_k)^{\mathrm{T}} \mathbb{E}_{S_k}[\epsilon_k] - \mathbb{E}_{S_k}[\nabla f(x_k)^{\mathrm{T}} M_k \epsilon_k] = -\mathbb{E}_{S_k}[\nabla f(x_k)^{\mathrm{T}} M_k \epsilon_k].
\end{aligned}$$

Using the *Cauchy-Schwarz inequality with expectations*, we obtain

$$\begin{aligned}
|\mathbb{E}_{S_k}[\nabla f(x_k)^{\mathrm{T}} H_k \epsilon_k]| = |\mathbb{E}_{S_k}[\nabla f(x_k)^{\mathrm{T}} M_k \epsilon_k]| &\leq \sqrt{\mathbb{E}_{S_k}[\|\nabla f(x_k)\|_2^2]}\sqrt{\mathbb{E}_{S_k}[\|M_k \epsilon_k\|_2^2]} \\
&\leq \|\nabla f(x_k)\|_2 \sqrt{\mathbb{E}_{S_k}[\|M_k \epsilon_k\|_2^2]}. \tag{39}
\end{aligned}$$

We now bound $\|M_k \epsilon_k\|_2^2$. For brevity, let $Z_k = R_k^{\mathrm{T}} R_k + \delta_k X_k^{\mathrm{T}} X_k$, and $N_1 = X_k Z_k^\dagger R_k^{\mathrm{T}}, N_2 = \beta_k R_k Z_k^\dagger R_k^{\mathrm{T}}$, then

$$\|M_k\|_2 = \|\alpha_k\left(N_1 + N_2\right)\|_2 \leq \alpha_k\left(\|N_1\|_2 + \|N_2\|_2\right). \tag{40}$$

Clearly, $X_k^{\mathrm{T}} X_k, R_k^{\mathrm{T}} R_k$ and $Z_k$ are symmetric positive semidefinite. Also, we have $\delta_k X_k^{\mathrm{T}} X_k \preceq Z_k, R_k^{\mathrm{T}} R_k \preceq Z_k$, where the notation "$\preceq$" denotes the *Loewner partial order*, i.e., $A \preceq B$ with $A, B \in \mathbb{R}^{m \times m}$ means that $B - A$ is positive semidefinite.

First we point out that

$$Z_k^\dagger = \lim_{t \to 0^+} Z_k^{\frac{1}{2}}\left(Z_k^2 + tI\right)^{-1} Z_k^{\frac{1}{2}}, \tag{41}$$

where $t > 0$, which can be verified as follows:
Since $Z_k$ is symmetric positive semidefinite, we have the eigenvalue decomposition: $Z_k = U \wedge U^{\mathrm{T}}$, where $UU^{\mathrm{T}} = I, 0 \preceq \wedge = diag\{\wedge_1, 0\} \in \mathbb{R}^{m \times m}$, and $\wedge_1$ is diagonal and nonsingular. Hence

$$Z_k^{\frac{1}{2}}\left(Z_k^2 + tI\right)^{-1} Z_k^{\frac{1}{2}} = U \wedge^{\frac{1}{2}}\left(\wedge^2 + tI\right)^{-1} \wedge^{\frac{1}{2}} U^{\mathrm{T}} = U\begin{pmatrix} \wedge_1(\wedge_1^2 + tI)^{-1} & 0 \\ 0 & 0 \end{pmatrix} U^{\mathrm{T}}.$$

It follows that $\lim_{t \to 0^+} Z_k^{\frac{1}{2}}\left(Z_k^2 + tI\right)^{-1} Z_k^{\frac{1}{2}} = U \wedge^\dagger U^{\mathrm{T}}$, where $\wedge^\dagger = diag\{\wedge_1^{-1}, 0\}$. From the definition of Penrose-Moore inverse, we know Equation (41) holds.

Since $\delta_k X_k^{\mathrm{T}} X_k \preceq Z_k, R_k^{\mathrm{T}} R_k \preceq Z_k$, we have

$$\delta_k Z_k^{\frac{1}{2}} X_k^{\mathrm{T}} X_k Z_k^{\frac{1}{2}} \preceq Z_k^2 \preceq Z_k^2 + tI, \quad Z_k^{\frac{1}{2}} R_k^{\mathrm{T}} R_k Z_k^{\frac{1}{2}} \preceq Z_k^2 \preceq Z_k^2 + tI. \tag{42}$$

Hence, we have
$$(Z_k^2 + tI)^{-\frac{1}{2}} \delta_k Z_k^{\frac{1}{2}} X_k^{\mathrm{T}} X_k Z_k^{\frac{1}{2}} (Z_k^2 + tI)^{-\frac{1}{2}} \preceq I, \quad (Z_k^2 + tI)^{-\frac{1}{2}} Z_k^{\frac{1}{2}} R_k^{\mathrm{T}} R_k Z_k^{\frac{1}{2}} (Z_k^2 + tI)^{-\frac{1}{2}} \preceq I,$$
which implies
$$\|(Z_k^2 + tI)^{-\frac{1}{2}} Z_k^{\frac{1}{2}} \left( X_k^{\mathrm{T}} X_k \right) Z_k^{\frac{1}{2}} (Z_k^2 + tI)^{-\frac{1}{2}}\|_2 \le \delta_k^{-1},$$
$$\|(Z_k^2 + tI)^{-\frac{1}{2}} Z_k^{\frac{1}{2}} \left( R_k^{\mathrm{T}} R_k \right) Z_k^{\frac{1}{2}} (Z_k^2 + tI)^{-\frac{1}{2}}\|_2 \le 1.$$
With Equation (41), we also have
$$N_1 = \lim_{t \to 0^+} N_1(t) := X_k Z_k^{\frac{1}{2}} \left( Z_k^2 + tI \right)^{-1} Z_k^{\frac{1}{2}} R_k^{\mathrm{T}},$$
$$N_2 = \lim_{t \to 0^+} N_2(t) := \beta_k R_k Z_k^{\frac{1}{2}} \left( Z_k^2 + tI \right)^{-1} Z_k^{\frac{1}{2}} R_k^{\mathrm{T}}.$$
Therefore,
$$\|N_1(t)\|_2^2 = \lambda_{max} \left( N_1(t) N_1(t)^{\mathrm{T}} \right)$$
$$= \lambda_{max} \left( X_k Z_k^{\frac{1}{2}} \left( Z_k^2 + tI \right)^{-1} Z_k^{\frac{1}{2}} R_k^{\mathrm{T}} \cdot R_k Z_k^{\frac{1}{2}} \left( Z_k^2 + tI \right)^{-1} Z_k^{\frac{1}{2}} X_k^{\mathrm{T}} \right)$$
$$= \lambda_{max} \left( Z_k^{\frac{1}{2}} X_k^{\mathrm{T}} X_k Z_k^{\frac{1}{2}} \left( Z_k^2 + tI \right)^{-1} Z_k^{\frac{1}{2}} R_k^{\mathrm{T}} R_k Z_k^{\frac{1}{2}} \left( Z_k^2 + tI \right)^{-1} \right)$$
$$= \lambda_{max}((Z_k^2 + tI)^{-\frac{1}{2}} Z_k^{\frac{1}{2}} X_k^{\mathrm{T}} X_k Z_k^{\frac{1}{2}} (Z_k^2 + tI)^{-\frac{1}{2}}$$
$$\cdot (Z_k^2 + tI)^{-\frac{1}{2}} Z_k^{\frac{1}{2}} R_k^{\mathrm{T}} R_k Z_k^{\frac{1}{2}} (Z_k^2 + tI)^{-\frac{1}{2}})$$
$$\le \|(Z_k^2 + tI)^{-\frac{1}{2}} Z_k^{\frac{1}{2}} X_k^{\mathrm{T}} X_k Z_k^{\frac{1}{2}} (Z_k^2 + tI)^{-\frac{1}{2}}$$
$$\cdot (Z_k^2 + tI)^{-\frac{1}{2}} Z_k^{\frac{1}{2}} R_k^{\mathrm{T}} R_k Z_k^{\frac{1}{2}} (Z_k^2 + tI)^{-\frac{1}{2}})\|_2$$
$$\le \|(Z_k^2 + tI)^{-\frac{1}{2}} Z_k^{\frac{1}{2}} X_k^{\mathrm{T}} X_k Z_k^{\frac{1}{2}} (Z_k^2 + tI)^{-\frac{1}{2}}\|_2$$
$$\cdot \|(Z_k^2 + tI)^{-\frac{1}{2}} Z_k^{\frac{1}{2}} R_k^{\mathrm{T}} R_k Z_k^{\frac{1}{2}} (Z_k^2 + tI)^{-\frac{1}{2}})\|_2 \le \delta_k^{-1},$$
$$\|N_2(t)\|_2 = \beta_k \lambda_{max} \left( R_k Z_k^{\frac{1}{2}} \left( Z_k^2 + tI \right)^{-1} Z_k^{\frac{1}{2}} R_k^{\mathrm{T}} \right)$$
$$= \beta_k \lambda_{max} \left( Z_k^{\frac{1}{2}} R_k^{\mathrm{T}} R_k Z_k^{\frac{1}{2}} \left( Z_k^2 + tI \right)^{-1} \right)$$
$$= \beta_k \lambda_{max} \left( \left( Z_k^2 + tI \right)^{-\frac{1}{2}} Z_k^{\frac{1}{2}} R_k^{\mathrm{T}} R_k Z_k^{\frac{1}{2}} \left( Z_k^2 + tI \right)^{-\frac{1}{2}} \right) \le \beta_k,$$
which implies $\|N_1\|_2^2 \le \delta_k^{-1}$ and $\|N_2\|_2 \le \beta_k$ from the continuity of singular value (e.g. Theorem 2.6.4 in [23]). With (40), we have
$$\|M_k\|_2 \le \alpha_k(\delta_k^{-\frac{1}{2}} + \beta_k). \tag{43}$$
Then $\|M_k \epsilon_k\|_2 \le \alpha_k(\delta_k^{-\frac{1}{2}} + \beta_k)\|\epsilon_k\|_2$, which implies
$$\mathbb{E}_{S_k}[\|M_k \epsilon_k\|_2^2] \le \alpha_k^2(\delta_k^{-\frac{1}{2}} + \beta_k)^2 \mathbb{E}_{S_k}[\|\epsilon_k\|_2^2] \le \alpha_k^2(\delta_k^{-\frac{1}{2}} + \beta_k)^2 \frac{\sigma^2}{n_k}, \tag{44}$$
where the last inequality is due to (26b). Now we can obtain the bound of $|\mathbb{E}_{S_k}[\nabla f(x_k)^{\mathrm{T}} H_k \epsilon_k]|$ as follows (cf. (39)):
$$|\mathbb{E}_{S_k}[\nabla f(x_k)^{\mathrm{T}} H_k \epsilon_k]|$$
$$\le \|\nabla f(x_k)\|_2 \sqrt{\mathbb{E}_{S_k}[\|M_k \epsilon_k\|_2^2]}$$
$$\le \alpha_k(\delta_k^{-\frac{1}{2}} + \beta_k)\|\nabla f(x_k)\|_2 \sqrt{\mathbb{E}_{S_k}[\|\epsilon_k\|_2^2]}$$
$$\le \alpha_k(\delta_k^{-\frac{1}{2}} + \beta_k)\frac{\sigma}{\sqrt{n_k}}\|\nabla f(x_k)\|_2$$
$$= \sqrt{\beta_k \mu}\|\nabla f(x_k)\|_2 \cdot \frac{\alpha_k(\delta_k^{-\frac{1}{2}} + \beta_k)}{\sqrt{\beta_k \mu}} \frac{\sigma}{\sqrt{n_k}}$$
$$\le \frac{1}{2}\beta_k \mu\|\nabla f(x_k)\|_2^2 + \frac{1}{2}\frac{\alpha_k^2(\delta_k^{-\frac{1}{2}} + \beta_k)^2}{\beta_k \mu} \cdot \frac{\sigma^2}{n_k}. \tag{45}$$

With the inequality (38) and (45), we obtain

$$
\begin{aligned}
&\nabla f(x_k)^{\mathrm{T}} \mathbb{E}_{S_k}[H_k r_k] \\
&= -\nabla f(x_k)^{\mathrm{T}} \mathbb{E}_{S_k}[H_k(\epsilon_k + \nabla f(x_k))] \\
&= -\mathbb{E}_{S_k}[\nabla f(x_k)^{\mathrm{T}} H_k \nabla f(x_k)] - \mathbb{E}_{S_k}[\nabla f(x_k)^{\mathrm{T}} H_k \epsilon_k] \\
&\leq -\mathbb{E}_{S_k}[\nabla f(x_k)^{\mathrm{T}} H_k \nabla f(x_k)] + |\mathbb{E}_{S_k}[\nabla f(x_k)^{\mathrm{T}} H_k \epsilon_k]| \\
&\leq -\beta_k \mu \|\nabla f(x_k)\|_2^2 + \frac{1}{2}\beta_k \mu \|\nabla f(x_k)\|_2^2 + \frac{1}{2}\frac{\alpha_k^2(\delta_k^{-\frac{1}{2}} + \beta_k)^2}{\beta_k \mu} \cdot \frac{\sigma^2}{n_k} \\
&= -\frac{1}{2}\beta_k \mu \|\nabla f(x_k)\|_2^2 + \frac{1}{2}\frac{\alpha_k^2(\delta_k^{-\frac{1}{2}} + \beta_k)^2}{\beta_k \mu} \cdot \frac{\sigma^2}{n_k}.
\end{aligned}
\tag{46}
$$

If $H_k$ is independent of $S_k$, then

$$
\mathbb{E}_{S_k}[\nabla f(x_k)^{\mathrm{T}} H_k \epsilon_k] = \nabla f(x_k)^{\mathrm{T}} H_k \mathbb{E}_{S_k}[\epsilon_k] = 0.
$$

Thus $\nabla f(x_k)^{\mathrm{T}} \mathbb{E}_{S_k}[H_k r_k] \leq -\beta_k \mu \|\nabla f(x_k)\|_2^2$. □

By imposing more restrictions to $\alpha_k, \beta_k, \delta_k$, we can obtain a convenient corollary:

**Corollary 1.** *Suppose that Assumption 2 holds for $\{x_k\}$ generated by SAM. $C > 0$ is a constant. If $\beta_k > 0, \delta_k \geq C\beta_k^{-2}, 0 \leq \alpha_k \leq \min\{1, \beta_k^{\frac{1}{2}}\}$ and satisfies (14), then*

$$
\mathbb{E}_{S_k}[\|H_k r_k\|_2^2] \leq 2\beta_k^2(1 + C^{-1}) \cdot \left(\|\nabla f(x_k)\|_2^2 + \frac{\sigma^2}{n_k}\right),
\tag{47a}
$$

$$
\nabla f(x_k)^{\mathrm{T}} \mathbb{E}_{S_k}[H_k r_k] \leq -\frac{1}{2}\beta_k \mu \|\nabla f(x_k)\|_2^2 + \beta_k^2 \cdot \mu^{-1}(1 + C^{-1})\frac{\sigma^2}{n_k}.
\tag{47b}
$$

*Proof.* The first result (47a) is easy to obtain by considering (34a) and noticing that $1 + 2\alpha_k^2 - 2\alpha_k \leq 1$ when $\alpha_k \in [0, 1]$ and $\delta_k^{-1} \leq C^{-1}\beta_k^2$. Since $\alpha_k \leq \beta_k^{\frac{1}{2}}, \delta_k \geq C\beta_k^{-2}$ and $(1 + C^{-\frac{1}{2}})^2 \leq 2(1 + C^{-1})$ we have

$$
\begin{aligned}
\frac{1}{2}\frac{\alpha_k^2(\delta_k^{-\frac{1}{2}} + \beta_k)^2}{\beta_k \mu} \cdot \frac{\sigma^2}{n_k} &\leq \frac{1}{2}\frac{\beta_k(C^{-\frac{1}{2}}\beta_k + \beta_k)^2}{\beta_k \mu} \cdot \frac{\sigma^2}{n_k} \\
&= \frac{1}{2}\mu^{-1}(C^{-\frac{1}{2}} + 1)^2 \beta_k^2 \cdot \frac{\sigma^2}{n_k} \\
&\leq \beta_k^2 \mu^{-1}(1 + C^{-1})\frac{\sigma^2}{n_k}.
\end{aligned}
$$

Substituting it into (34b), we obtain (47b). □

Using Corollary 1 we obtain the descent property of SAM:

**Lemma 3.** *Suppose that Assumptions 1 and 2 hold for $\{x_k\}$ generated by SAM. $C > 0$ is a constant. If $0 < \beta_k \leq \frac{\mu}{4L(1+C^{-1})}, \delta_k \geq C\beta_k^{-2}, 0 \leq \alpha_k \leq \min\{1, \beta_k^{\frac{1}{2}}\}$ and satisfies (14), then*

$$
\mathbb{E}_{S_k}[f(x_{k+1})] \leq f(x_k) - \frac{1}{4}\beta_k \mu \|\nabla f(x_k)\|_2^2 + \beta_k^2\left((L + \mu^{-1})(1 + C^{-1})\right)\frac{\sigma^2}{n_k}.
\tag{48}
$$

*Proof.* According to Assumption 1, we have

$$
\begin{aligned}
f(x_{k+1}) &\leq f(x_k) + \nabla f(x_k)^{\mathrm{T}}(x_{k+1} - x_k) + \frac{L}{2}\|x_{k+1} - x_k\|_2^2 \\
&= f(x_k) + \nabla f(x_k)^{\mathrm{T}} H_k r_k + \frac{L}{2}\|H_k r_k\|_2^2.
\end{aligned}
\tag{49}
$$

Taking expectation with respect to the mini-batch $S_k$ on both sides of (49) and using Corollary 1 we obtain

$$\mathbb{E}_{S_k}[f(x_{k+1})]$$

$$\leq f(x_k) + \nabla f(x_k)^{\mathrm{T}} \mathbb{E}_{S_k}[H_k r_k] + \frac{L}{2} \mathbb{E}_{S_k} \|H_k r_k\|_2^2$$

$$\leq f(x_k) - \frac{1}{2}\beta_k \mu \|\nabla f(x_k)\|_2^2 + \beta_k^2 \mu^{-1}(1 + C^{-1})\frac{\sigma^2}{n_k} + L\beta_k^2(1 + C^{-1})\left(\|\nabla f(x_k)\|_2^2 + \frac{\sigma^2}{n_k}\right)$$

$$= f(x_k) - \beta_k\left(\frac{1}{2}\mu - \beta_k L(1 + C^{-1})\right)\|\nabla f(x_k)\|_2^2 + \beta_k^2(L + \mu^{-1})(1 + C^{-1})\frac{\sigma^2}{n_k}. \tag{50}$$

Then (50) combined with the assumption $\beta_k \leq \frac{\mu}{4L(1+C^{-1})}$ implies (48). $\qquad\square$

Following the proofs in [60], we introduce the definition of a *supermartingale*.

**Definition 1.** *Let $\{\mathcal{F}_k\}$ be an increasing sequence of $\sigma$-algebras. If $\{X_k\}$ is a stochastic process satisfying (i) $\mathbb{E}[|X_k|] < \infty$, (ii) $X_k \in \mathcal{F}_k$ for all k, and (iii) $\mathbb{E}[X_{k+1}|\mathcal{F}_k] \leq X_k$ for all k, then $\{X_k\}$ is called a supermartingale.*

**Proposition 1** (Supermartingale convergence theorem, see, e.g., Theorem 4.2.12 in [13]). *If $\{X_k\}$ is a nonnegative supermartingale, then $\lim_{k\to\infty} X_k \to X$ almost surely and $\mathbb{E}[X] \leq \mathbb{E}[X_0]$.*

Now, we prove our theorems.

***Proof of Theorem 1.*** Define $\zeta_k := \frac{\beta_k \mu}{4}\|\nabla f(x_k)\|_2^2$ and $\tilde{L} := (L + \mu^{-1})(1 + C^{-1})$, $\gamma_k := f(x_k) + \tilde{L}\frac{\sigma^2}{n}\sum_{i=k}^{\infty}\beta_i^2$. Let $\mathcal{F}_k$ be the $\sigma$-algebra measuring $\zeta_k, \gamma_k$, and $x_k$. From (48) we know that for any $k$,

$$\mathbb{E}[\gamma_{k+1}|\mathcal{F}_k] = \mathbb{E}[f(x_{k+1})|\mathcal{F}_k] + \tilde{L}\frac{\sigma^2}{n}\sum_{i=k+1}^{\infty}\beta_i^2$$

$$\leq f(x_k) - \frac{1}{4}\beta_k\mu\|\nabla f(x_k)\|_2^2 + \tilde{L}\frac{\sigma^2}{n}\sum_{i=k}^{\infty}\beta_i^2 = \gamma_k - \zeta_k, \tag{51}$$

which implies that $\mathbb{E}[\gamma_{k+1} - f^{low}|\mathcal{F}_k] \leq \gamma_k - f^{low} - \zeta_k$. Since $\zeta_k \geq 0$, we have $0 \leq \mathbb{E}[\gamma_k - f^{low}] \leq \gamma_0 - f^{low} < +\infty$. As the diminishing condition (18a) holds, we obtain (20). According to Definition 1, $\{\gamma_k - f^{low}\}$ is a supermartingale. Therefore, Proposition 1 indicates that there exists a $\gamma$ such that $\lim_{k\to\infty}\gamma_k = \gamma$ with probability 1, and $\mathbb{E}[\gamma] \leq \mathbb{E}[\gamma_0]$. Note that from (51) we have $\mathbb{E}[\zeta_k] \leq \mathbb{E}[\gamma_k] - \mathbb{E}[\gamma_{k+1}]$. Thus,

$$\mathbb{E}\left[\sum_{k=0}^{\infty}\zeta_k\right] \leq \sum_{k=0}^{\infty}(\mathbb{E}[\gamma_k] - \mathbb{E}[\gamma_{k+1}]) < +\infty,$$

which further yields that

$$\sum_{k=0}^{\infty}\zeta_k = \frac{\mu}{4}\sum_{k=0}^{\infty}\beta_k\|\nabla f(x_k)\|_2^2 < +\infty \text{ with probability 1}. \tag{52}$$

Since $\sum_{k=0}^{\infty}\beta_k = +\infty$, it follows that (19) holds. $\qquad\square$

***Proof of Theorem 2.*** For any give $\epsilon > 0$, according to (19), there exist infinitely many iterates $x_k$ such that $\|\nabla f(x_k)\|_2 \leq \epsilon$. Then if (22) does not hold, there must exist two infinite sequences of indices $\{s_i\}, \{t_i\}$ with $t_i > s_i$, such that for $i = 0, 1, \ldots, k = s_i + 1, \ldots, t_i - 1$,

$$\|\nabla f(x_{s_i})\|_2 \geq 2\epsilon, \|\nabla f(x_{t_i})\|_2 < \epsilon, \|\nabla f(x_k)\|_2 \geq \epsilon. \tag{53}$$

Then from (52) it follows that

$$+\infty > \sum_{k=0}^{\infty}\beta_k\|\nabla f(x_k)\|_2^2 \geq \sum_{i=0}^{+\infty}\sum_{k=s_i}^{t_i-1}\beta_k\|\nabla f(x_k)\|_2^2 \geq \epsilon^2\sum_{i=0}^{+\infty}\sum_{k=s_i}^{t_i-1}\beta_k \text{ with probability 1},$$

which implies that

$$\sum_{k=s_i}^{t_i-1} \beta_k \to 0 \text{ with probability } 1, \text{ as } i \to +\infty. \tag{54}$$

According to (37) and (32), we have

$$
\begin{aligned}
&\mathbb{E}[\|x_{k+1} - x_k\|_2 | x_k] \\
&= \mathbb{E}[\|H_k r_k\|_2 | x_k] \\
&\leq \sqrt{2\left(\beta_k^2\left(1 + 2\alpha_k^2 - 2\alpha_k\right) + \alpha_k^2 \delta_k^{-1}\right)} \mathbb{E}[\|r_k\|_2 | x_k] \\
&\leq \beta_k \sqrt{2(1 + C^{-1})} \mathbb{E}[\|r_k\|_2 | x_k] \\
&\leq \beta_k \sqrt{2(1 + C^{-1})} (\mathbb{E}[\|r_k\|_2^2 | x_k])^{\frac{1}{2}} \\
&\leq \beta_k \sqrt{2(1 + C^{-1})} M_g^{\frac{1}{2}}, \tag{55}
\end{aligned}
$$

where the last inequalities are due to *Cauchy-Schwarz inequality* and (21). Then it follows from (55) that

$$\mathbb{E}[\|x_{t_i} - x_{s_i}\|_2] \leq \sqrt{2(1 + C^{-1})} M_g^{\frac{1}{2}} \sum_{k=s_i}^{t_i-1} \beta_k,$$

which together with (54) implies that $\|x_{t_i} - x_{s_i}\|_2 \to 0$ with probability 1, as $i \to +\infty$. Hence, from the Lipschitz continuity of $\nabla f$, it follows that $\|\nabla f(x_{t_i}) - \nabla f(x_{s_i})\|_2 \to 0$ with probability 1 as $i \to +\infty$. However, this contradicts (53). Therefore, the assumption that (22) does not hold is not true. $\qquad \square$

***Proof of Theorem 3.*** Define $\tilde{L} := (L + \mu^{-1})(1 + C^{-1})$. Taking expectation on both sides of (48) and summing over $k = 0, 1, \ldots, N-1$ yields

$$
\begin{aligned}
&\frac{1}{4}\mu \sum_{k=0}^{N-1} \mathbb{E}[\|\nabla f(x_k)\|_2^2] \\
&\leq \sum_{k=0}^{N-1} \frac{1}{\beta_k}\left(\mathbb{E}[f(x_k)] - \mathbb{E}[f(x_{k+1})]\right) + \tilde{L}\frac{\sigma^2}{n} \sum_{k=0}^{N-1} \beta_k \\
&= \frac{1}{\beta_0} f(x_0) + \sum_{k=1}^{N-1}\left(\frac{1}{\beta_k} - \frac{1}{\beta_{k-1}}\right)\mathbb{E}[f(x_k)] - \frac{1}{\beta_{N-1}}\mathbb{E}[f(x_N)] + \tilde{L}\frac{\sigma^2}{n} \sum_{k=0}^{N-1} \beta_k \\
&\leq \frac{M_f}{\beta_0} + M_f \sum_{k=1}^{N-1}\left(\frac{1}{\beta_k} - \frac{1}{\beta_{k-1}}\right) - \frac{f^{low}}{\beta_{N-1}} + \tilde{L}\frac{\sigma^2}{n} \sum_{k=0}^{N-1} \beta_k \\
&= \frac{M_f - f^{low}}{\beta_{N-1}} + \tilde{L}\frac{\sigma^2}{n} \sum_{k=0}^{N-1} \beta_k \\
&\leq \frac{4L(1 + C^{-1})(M_f - f^{low})}{\mu}N^r + \frac{(\mu + L^{-1})\sigma^2}{4n(1-r)}(N^{1-r} - r),
\end{aligned}
$$

which results in (23), where the second inequality is due to (20) and the last inequality is due to the choice of $\beta_k$. Then for a give $\epsilon > 0$, to guarantee that $\frac{1}{N}\sum_{k=0}^{N-1}\mathbb{E}\|\nabla f(x_k)\|_2^2 < \epsilon$, it suffices to require that

$$\frac{16L(1 + C^{-1})(M_f - f^{low})}{\mu^2}N^{r-1} + \frac{(1 + L^{-1}\mu^{-1})\sigma^2}{(1-r)n}(N^{-r} - rN^{-1}) < \epsilon.$$

Since $r \in (0.5, 1)$, it follows that the number of iterations $N$ needed is at most $O(\epsilon^{-\frac{1}{1-r}})$. $\qquad \square$

***Proof of Theorem 4.*** According to (50) in Lemma 3, we have

$$\sum_{k=0}^{N-1} \beta_k \left( \frac{1}{2}\mu - \beta_k L(1 + C^{-1}) \right) \mathbb{E}\|\nabla f(x_k)\|_2^2$$

$$\leq f(x_0) - f^{low} + \sum_{k=0}^{N-1} \beta_k^2 (L + \mu^{-1})(1 + C^{-1}) \frac{\sigma^2}{n_k}, \qquad (56)$$

where the expectation is taken with respect to $\{S_j\}_{j=0}^{N-1}$. Define

$$P_R(k) \stackrel{\text{def}}{=} Prob\{R = k\} = \frac{\beta_k \left( \frac{1}{2}\mu - \beta_k L(1 + C^{-1}) \right)}{\sum_{j=0}^{N-1} \beta_j \left( \frac{1}{2}\mu - \beta_j L(1 + C^{-1}) \right)}, \quad k = 0, \ldots, N-1, \qquad (57)$$

then

$$\mathbb{E}\left[\|\nabla f(x_R)\|_2^2\right] = \frac{\sum_{k=0}^{N-1} \beta_k \left( \frac{1}{2}\mu - \beta_k L(1 + C^{-1}) \right) \mathbb{E}\left[\|\nabla f(x_k)\|_2^2\right]}{\sum_{j=0}^{N-1} \beta_j \left( \frac{1}{2}\mu - \beta_j L(1 + C^{-1}) \right)}$$

$$\leq \frac{D_f + \sigma^2 (L + \mu^{-1})(1 + C^{-1}) \sum_{k=0}^{N-1} \beta_k^2/n_k}{\sum_{j=0}^{N-1} \beta_j \left( \frac{1}{2}\mu - \beta_j L(1 + C^{-1}) \right)}. \qquad (58)$$

If we choose $\beta_k = \beta := \frac{\mu}{4L(1+C^{-1})}$ and $n_k = n$, then the definition of $P_R$ simplifies to $P_R(k) = 1/N$ and we have

$$\mathbb{E}\left[\|\nabla f(x_R)\|_2^2\right] \leq \frac{D_f + \sigma^2 (L + \mu^{-1})(1 + C^{-1}) N \frac{\beta^2}{n}}{\frac{1}{4}\mu N \beta}$$

$$= \frac{4D_f}{\mu N \beta} + \frac{4(L + \mu^{-1})(1 + C^{-1})\sigma^2 \frac{\beta}{n}}{\mu}$$

$$= \frac{16 D_f L(1 + C^{-1})}{N\mu^2} + \frac{(L + \mu^{-1})\sigma^2}{nL}. \qquad (59)$$

Let $\bar{N}$ be the total number of $\mathcal{SFO}$-calls needed to calculate stochastic gradients in SAM. Then the number of iterations of SAM is at most $N = \lceil \bar{N}/n \rceil$. Obviously, $N \geq \bar{N}/(2n)$.

For a given accuracy tolerance $\epsilon > 0$, we assume that

$$\bar{N} \geq \max \left\{ \frac{C_1^2}{\epsilon^2} + \frac{4C_2}{\epsilon}, \frac{\sigma^2}{L^2 \tilde{D}} \right\}, \qquad (60)$$

where

$$C_1 = \frac{32 D_f (1 + C^{-1})\sigma}{\mu^2 \sqrt{\tilde{D}}} + (L + \mu^{-1})\sigma \sqrt{\tilde{D}}, \quad C_2 = \frac{32 D_f L(1 + C^{-1})}{\mu^2}, \qquad (61)$$

where $\tilde{D}$ is a problem-independent positive constant. Moreover, we assume that the batch size satisfies

$$n_k = n := \left\lceil \min \left\{ \bar{N}, \max \left\{ 1, \frac{\sigma}{L}\sqrt{\frac{\bar{N}}{\tilde{D}}} \right\} \right\} \right\rceil. \qquad (62)$$

The we can prove $\mathbb{E}\left[\|\nabla f(x_R)\|_2^2\right] \leq \epsilon$ as follows.
From (59) we have that

$$\mathbb{E}\left[\|\nabla f(x_R)\|_2^2\right] \leq \frac{32 D_f L(1 + C^{-1})n}{\mu^2 \bar{N}} + \frac{L + \mu^{-1}}{L} \frac{\sigma^2}{n}$$

$$\leq \frac{32 D_f L(1 + C^{-1})}{\mu^2 \bar{N}} \left( 1 + \frac{\sigma}{L}\sqrt{\frac{\bar{N}}{\tilde{D}}} \right) + \frac{L + \mu^{-1}}{L} \cdot \max \left\{ \frac{\sigma^2}{\bar{N}}, \frac{\sigma L \sqrt{\tilde{D}}}{\sqrt{\bar{N}}} \right\}. \qquad (63)$$

Equation (60) implies that

$$\frac{\sigma^2}{\bar{N}} = \frac{\sigma}{\sqrt{\bar{N}}} \cdot \frac{\sigma}{\sqrt{\bar{N}}} \le L\sqrt{\widetilde{D}} \cdot \frac{\sigma}{\sqrt{\bar{N}}} = \frac{\sigma L\sqrt{\widetilde{D}}}{\sqrt{\bar{N}}}. \tag{64}$$

Then from (63) and (61), we have

$$\mathbb{E}\left[\|\nabla f(x_R)\|_2^2\right] \le \frac{32 D_f L (1 + C^{-1})}{\mu^2 \bar{N}} \left(1 + \frac{\sigma}{L}\sqrt{\frac{\bar{N}}{\widetilde{D}}}\right) + \frac{L + \mu^{-1}}{L}\frac{\sigma L\sqrt{\widetilde{D}}}{\sqrt{\bar{N}}} = \frac{C_1}{\sqrt{\bar{N}}} + \frac{C_2}{\bar{N}}. \tag{65}$$

To ensure $\mathbb{E}\left[\|\nabla f(x_R)\|_2^2\right] \le \epsilon$, it is sufficient to let the upper bound $\frac{C_1}{\sqrt{\bar{N}}} + \frac{C_2}{\bar{N}} \le \epsilon$, which implies

$$\sqrt{\bar{N}} \ge \frac{\sqrt{C_1^2 + 4C_2\epsilon} + C_1}{2\epsilon}.$$

This is guaranteed by Condition (60) since

$$\sqrt{\bar{N}} \ge \frac{\sqrt{C_1^2 + 4C_2\epsilon}}{\epsilon} \ge \frac{\sqrt{C_1^2 + 4C_2\epsilon} + C_1}{2\epsilon}.$$

$\square$

To prove Theorem 5, we first prove the following lemma.

**Lemma 4.** *Suppose that Assumptions 1 and 2 hold. $C > 0$ is a constant. $\beta_t^k > 0, \delta_t^k \ge C(\beta_t^k)^{-2}$, $0 \le \alpha_t^k \le \min\{1, (\beta_t^k)^{\frac{1}{2}}\}$ and satisfies (14). For any $\eta_t > 0$, set*

$$c_t^k = c_{t+1}^k \left(1 + 2\beta_t^k \eta_t + 4(\beta_t^k)^2(1 + C^{-1})\frac{L^2}{n} + 2(\beta_t^k)^2\eta_t^{-1}(1 + C^{-1})\frac{L^2}{n}\right)$$

$$+ (\beta_t^k)^2 (L + \frac{\mu^{-1}}{2})(1 + C^{-1})\frac{2L^2}{n}.$$

*Then*

$$\Psi_t^k \mathbb{E}\left[\|\nabla f(x_t^k)\|_2^2\right] \le \Phi_t^k - \Phi_{t+1}^k, \tag{66}$$

*where $\Phi_t^k = \mathbb{E}\left[f(x_t^k) + c_t^k\|x_t^k - \tilde{x}_k\|_2^2\right]$ and $\Psi_t^k = \frac{1}{2}\beta_t^k\mu - 2c_{t+1}^k\beta_t^k\eta_t^{-1}(1 + C^{-1}) - 2L(\beta_t^k)^2(1 + C^{-1}) - 4c_{t+1}^k(\beta_t^k)^2(1 + C^{-1})$.*

*(**Note**: Here and in the following proof of Theorem 5, the definition of notation $c_t^k$ has no relation with $c_1, c_2$ in (16) of AdaSAM.)*

*Proof.* Define $\epsilon_t^k = -r_t^k - \nabla f(x_t^k), M_t^k = \beta_t^k I - H_t^k$. Then
$$\mathbb{E}\left[\nabla f(x_t^k)^{\mathrm{T}} H_t^k \epsilon_t^k\right] = \mathbb{E}\left[\nabla f(x_t^k)^{\mathrm{T}}(\beta_t^k \epsilon_t^k - M_t^k \epsilon_t^k)\right] = -\mathbb{E}\left[f(x_t^k)^{\mathrm{T}} M_t^k \epsilon_t^k\right].$$

According to Lemma 1, for any $v_k \in \mathbb{R}^d$,
$$\|H_t^k v_k\|_2^2 \le 2\left((\beta_t^k)^2\left(1 + 2(\alpha_t^k)^2 - 2\alpha_t^k\right) + (\alpha_t^k)^2(\delta_t^k)^{-1}\right)\|v_k\|_2^2 \le 2(\beta_t^k)^2(1 + C^{-1})\|v_k\|_2^2. \tag{67}$$

We also have

$$\begin{aligned}
&\nabla f(x_t^k)^{\mathrm{T}} \mathbb{E}\left[H_t^k r_t^k\right] \\
&= -\nabla f(x_t^k)^{\mathrm{T}} H_t^k \nabla f(x_t^k) - \mathbb{E}\left[\nabla f(x_t^k)^{\mathrm{T}} H_t^k \epsilon_t^k\right] \\
&\le -\beta_t^k \mu \|\nabla f(x_t^k)\|_2^2 + |\mathbb{E}\left[\nabla f(x_t^k)^{\mathrm{T}} M_t^k \epsilon_t^k\right]| \\
&\le -\beta_t^k \mu \|\nabla f(x_t^k)\|_2^2 + \|\nabla f(x_t^k)\|_2 \sqrt{\mathbb{E}\left[\|M_t^k \epsilon_t^k\|_2^2\right]} \\
&\le -\beta_t^k \mu \|\nabla f(x_t^k)\|_2^2 + \alpha_t^k((\delta_t^k)^{-\frac{1}{2}} + \beta_t^k)\|\nabla f(x_t^k)\|_2 \sqrt{\mathbb{E}\left[\|\epsilon_t^k\|_2^2\right]} \quad \text{(cf. (43))} \\
&= -\beta_t^k \mu \|\nabla f(x_t^k)\|_2^2 + \sqrt{\beta_t^k \mu}\|\nabla f(x_t^k)\|_2 \cdot \frac{\alpha_t^k((\delta_t^k)^{-\frac{1}{2}} + \beta_t^k)}{\sqrt{\beta_t^k \mu}}\sqrt{\mathbb{E}\left[\|\epsilon_t^k\|_2^2\right]} \\
&\le -\frac{1}{2}\beta_t^k \mu \|\nabla f(x_t^k)\|_2^2 + \frac{1}{2}\frac{(\alpha_t^k)^2((\delta_t^k)^{-\frac{1}{2}} + \beta_t^k)^2}{\beta_t^k \mu}\mathbb{E}\left[\|\epsilon_t^k\|_2^2\right] \\
&\le -\frac{1}{2}\beta_t^k \mu \|\nabla f(x_t^k)\|_2^2 + (\beta_t^k)^2\mu^{-1}(1 + C^{-1})\mathbb{E}\|\epsilon_t^k\|_2^2. \tag{68}
\end{aligned}$$

Hence, with Assumption 1 we obtain

$$\mathbb{E}\left[f(x_{t+1}^k)\right]$$
$$\leq \mathbb{E}\left[f(x_t^k) + \nabla f(x_t^k)^{\mathrm{T}}(x_{t+1}^k - x_t^k) + \frac{L}{2}\|x_{t+1}^k - x_t^k\|_2^2\right]$$
$$= \mathbb{E}\left[f(x_t^k) + \nabla f(x_t^k)^{\mathrm{T}} H_t^k r_t^k + \frac{L}{2}\|H_t^k r_t^k\|_2^2\right]$$
$$\leq \mathbb{E}\left[f(x_t^k) - \frac{1}{2}\beta_t^k \mu\|\nabla f(x_t^k)\|_2^2 + (\beta_t^k)^2 \mu^{-1}(1+C^{-1})\mathbb{E}\|\epsilon_t^k\|_2^2 + L(\beta_t^k)^2(1+C^{-1})\|r_t^k\|_2^2\right].$$
$$\tag{69}$$

Next, we give a bound of $(x_t^k - \tilde{x}_k)^{\mathrm{T}} H_t^k r_t^k$. Since

$$-(x_t^k - \tilde{x}_k)^{\mathrm{T}} H_t^k \nabla f(x_t^k) = -\beta_t^k \cdot (\beta_t^k)^{-1}(x_t^k - \tilde{x}_k)^{\mathrm{T}} H_t^k \nabla f(x_t^k)$$
$$\leq \beta_t^k \|x_t^k - \tilde{x}_k\|_2 \|(\beta_t^k)^{-1} H_t^k \nabla f(x_t^k)\|_2$$
$$= \beta_t^k \eta_t^{1/2}\|x_t^k - \tilde{x}_k\|_2 \cdot \eta_t^{-1/2}\|(\beta_t^k)^{-1} H_t^k \nabla f(x_t^k)\|_2$$
$$\leq \frac{1}{2}\beta_t^k \left(\eta_t\|x_t^k - \tilde{x}_k\|_2^2 + \eta_t^{-1}(\beta_t^k)^{-2}\|H_t^k \nabla f(x_t^k)\|_2^2\right)$$
$$\leq \frac{1}{2}\beta_t^k \left(\eta_t\|x_t^k - \tilde{x}_k\|_2^2 + 2\eta_t^{-1}(1+C^{-1})\|\nabla f(x_t^k)\|_2^2\right),$$

and

$$-\mathbb{E}[(x_t^k - \tilde{x}_k)^{\mathrm{T}} H_t^k \epsilon_t^k] = -\mathbb{E}[(x_t^k - \tilde{x}_k)^{\mathrm{T}}(\beta_t^k I - M_t^k)\epsilon_t^k]$$
$$= \mathbb{E}[(x_t^k - \tilde{x}_k)^{\mathrm{T}} M_t^k \epsilon_t^k]$$
$$\leq \|x_t^k - \tilde{x}_k\|_2 \cdot \sqrt{\mathbb{E}[\|M_t^k \epsilon_t^k\|_2^2]}$$
$$= \sqrt{\beta_t^k \eta_t}\|x_t^k - \tilde{x}_k\|_2 \cdot \frac{\alpha_t^k((\delta_t^k)^{-1/2} + \beta_t^k)}{\sqrt{\beta_t^k \eta_t}}\sqrt{\mathbb{E}\|\epsilon_t^k\|_2^2}$$
$$\leq \frac{1}{2}\beta_t^k \eta_t\|x_t^k - \tilde{x}_k\|_2^2 + \frac{1}{2}\frac{(\alpha_t^k)^2((\delta_t^k)^{-1/2} + \beta_t^k)^2}{\beta_t^k \eta_t}\mathbb{E}\|\epsilon_t^k\|_2^2$$
$$\leq \frac{1}{2}\beta_t^k \eta_t\|x_t^k - \tilde{x}_k\|_2^2 + \eta_t^{-1}(\beta_t^k)^2(1+C^{-1})\mathbb{E}\|\epsilon_t^k\|_2^2,$$

we obtain

$$\mathbb{E}[(x_t^k - \tilde{x}_k)^{\mathrm{T}} H_t^k r_t^k]$$
$$= \mathbb{E}[-(x_t^k - \tilde{x}_k)^{\mathrm{T}} H_t^k \nabla f(x_t^k) - (x_t^k - \tilde{x}_k)^{\mathrm{T}} H_t^k \epsilon_t^k]$$
$$\leq \mathbb{E}[\beta_t^k \eta_t^{-1}(1+C^{-1})\|\nabla f(x_t^k)\|_2^2 + \beta_t^k \eta_t\|x_t^k - \tilde{x}_k\|_2^2 + \eta_t^{-1}(\beta_t^k)^2(1+C^{-1})\mathbb{E}\|\epsilon_t^k\|_2^2].$$

Hence, we have

$$\mathbb{E}[\|x_{t+1}^k - \tilde{x}_k\|_2^2]$$
$$= \mathbb{E}[\|x_{t+1}^k - x_t^k\|_2^2 + \|x_t^k - \tilde{x}_k\|_2^2 + 2(x_t^k - \tilde{x}_k)^{\mathrm{T}}(x_{t+1}^k - x_t^k)]$$
$$= \mathbb{E}[\|H_t^k r_t^k\|_2^2] + \|x_t^k - \tilde{x}_k\|_2^2 + 2(x_t^k - \tilde{x}_k)^{\mathrm{T}} H_t^k r_t^k]$$
$$\leq \mathbb{E}[2(\beta_t^k)^2(1+C^{-1})\|r_t^k\|_2^2 + \|x_t^k - \tilde{x}_k\|_2^2$$
$$+ 2\beta_t^k \eta_t^{-1}(1+C^{-1})\|\nabla f(x_t^k)\|_2^2 + 2\beta_t^k \eta_t\|x_t^k - \tilde{x}_k\|_2^2 + 2\eta_t^{-1}(\beta_t^k)^2(1+C^{-1})\mathbb{E}\|\epsilon_t^k\|_2^2]. \tag{70}$$

Also, the following inequalities hold:

$$\mathbb{E}\left[\|\epsilon_t^k\|_2^2\right] = \mathbb{E}\left[\|\nabla f_{\mathcal{K}}(x_t^k) - \nabla f_{\mathcal{K}}(\tilde{x}_k) + \nabla f(\tilde{x}_k) - \nabla f(x_t^k)\|_2^2\right]$$
$$= \mathbb{E}\left[\|\nabla f_{\mathcal{K}}(x_t^k) - \nabla f_{\mathcal{K}}(\tilde{x}_k) - \mathbb{E}\left[\nabla f_{\mathcal{K}}(x_t^k) - \nabla f_{\mathcal{K}}(\tilde{x}_k)\right]\|_2^2\right]$$
$$= \frac{1}{n}\mathbb{E}\left[\|\nabla f_i(x_t^k) - \nabla f_i(\tilde{x}_k) - \mathbb{E}\left[\nabla f_i(x_t^k) - \nabla f_i(\tilde{x}_k)\right]\|_2^2\right]$$
$$\leq \frac{1}{n}\mathbb{E}\left[\|\nabla f_i(x_t^k) - \nabla f_i(\tilde{x}_k)\|_2^2\right] \leq \frac{L^2}{n}\mathbb{E}\left[\|x_t^k - \tilde{x}_k\|_2^2\right], \tag{71}$$

$$\mathbb{E}\left[\|r_t^k\|_2^2\right] = \mathbb{E}\left[\|\epsilon_t^k + \nabla f(x_t^k)\|_2^2\right]$$
$$\leq \mathbb{E}\left[2\|\epsilon_t^k\|_2^2 + 2\|\nabla f(x_t^k)\|_2^2\right]$$
$$\leq 2\mathbb{E}\left[\|\nabla f(x_t^k)\|_2^2\right] + \frac{2L^2}{n}\mathbb{E}\left[\|x_t^k - \tilde{x}_k\|_2^2\right]. \tag{72}$$

Combining (69), (70), (71) and (72) yields that

$$\Phi_{t+1}^k = \mathbb{E}[f(x_{t+1}^k) + c_{t+1}^k\|x_{t+1}^k - \tilde{x}_k\|_2^2]$$

$$\leq \mathbb{E}[f(x_t^k) - \frac{1}{2}\beta_t^k\mu\|\nabla f(x_t^k)\|_2^2 + (\beta_t^k)^2\mu^{-1}(1+C^{-1})\mathbb{E}\|\epsilon_t^k\|_2^2 + L(\beta_t^k)^2(1+C^{-1})\|r_t^k\|_2^2$$

$$+ c_{t+1}^k 2(\beta_t^k)^2(1+C^{-1})\|r_t^k\|_2^2 + c_{t+1}^k\|x_t^k - \tilde{x}_k\|_2^2 + c_{t+1}^k 2\beta_t^k\eta_t^{-1}(1+C^{-1})\|\nabla f(x_t^k)\|_2^2$$

$$+ c_{t+1}^k 2\beta_t^k\eta_t\|x_t^k - \tilde{x}_k\|_2^2 + c_{t+1}^k 2\eta_t^{-1}(\beta_t^k)^2(1+C^{-1})\mathbb{E}\|\epsilon_t^k\|_2^2]$$

$$= \mathbb{E}[f(x_t^k) - (\frac{1}{2}\beta_t^k\mu - c_{t+1}^k 2\beta_t^k\eta_t^{-1}(1+C^{-1}))\|\nabla f(x_t^k)\|_2^2$$

$$+ (L(\beta_t^k)^2(1+C^{-1}) + c_{t+1}^k 2(\beta_t^k)^2(1+C^{-1}))\|r_t^k\|_2^2$$

$$+ (c_{t+1}^k + c_{t+1}^k 2\beta_t^k\eta_t)\|x_t^k - \tilde{x}_k\|_2^2$$

$$+ ((\beta_t^k)^2\mu^{-1}(1+C^{-1}) + c_{t+1}^k 2(\beta_t^k)^2\eta_t^{-1}(1+C^{-1}))\|\epsilon_t^k\|_2^2]$$

$$\leq \mathbb{E}[f(x_t^k) - (\frac{1}{2}\beta_t^k\mu - c_{t+1}^k 2\beta_t^k\eta_t^{-1}(1+C^{-1}))\|\nabla f(x_t^k)\|_2^2$$

$$+ (L(\beta_t^k)^2(1+C^{-1}) + c_{t+1}^k 2(\beta_t^k)^2(1+C^{-1}))(2\|\nabla f(x_t^k)\|_2^2 + \frac{2L^2}{n}\|x_t^k - \tilde{x}_k\|_2^2)$$

$$+ (c_{t+1}^k + c_{t+1}^k 2\beta_t^k\eta_t)\|x_t^k - \tilde{x}_k\|_2^2$$

$$+ ((\beta_t^k)^2\mu^{-1}(1+C^{-1}) + c_{t+1}^k 2(\beta_t^k)^2\eta_t^{-1}(1+C^{-1}))\frac{L^2}{n}\|x_t^k - \tilde{x}_k\|_2^2]$$

$$= \mathbb{E}[f(x_t^k) + (c_{t+1}^k(1 + 2\beta_t^k\eta_t + 4(\beta_t^k)^2(1+C^{-1})\frac{L^2}{n} + 2(\beta_t^k)^2\eta_t^{-1}(1+C^{-1})\frac{L^2}{n})$$

$$+ (\beta_t^k)^2(1+C^{-1})\frac{2L^3}{n} + (\beta_t^k)^2\mu^{-1}(1+C^{-1})\frac{L^2}{n})\|x_t^k - \tilde{x}_k\|_2^2$$

$$- (\frac{1}{2}\beta_t^k\mu - 2c_{t+1}^k\beta_t^k\eta_t^{-1}(1+C^{-1}) - 2L(\beta_t^k)^2(1+C^{-1})$$

$$- 4c_{t+1}^k(\beta_t^k)^2(1+C^{-1}))\|\nabla f(x_t^k)\|_2^2] = \Phi_t^k - \Psi_t^k\mathbb{E}[\|\nabla f(x_t^k)\|_2^2],$$

which further implies (66). $\qquad\square$

***Proof of Theorem 5.*** Let $\eta_t = \eta := \frac{L(1+C^{-1})^{1/2}}{T^{1/3}}$. Denote $\theta = 2\beta\eta + 4\beta^2(1+C^{-1})L^2/n + 2\beta^2\eta^{-1}(1+C^{-1})L^2/n$. It then follows that

$$\theta = \frac{2\mu_0 n}{T} + \frac{4\mu_0^2 n}{T^{4/3}} + \frac{2\mu_0^2 n}{TL(1+C^{-1})^{1/2}}$$

$$\leq \frac{\mu_0 n}{T}\left(2 + 4\mu_0 + \frac{2\mu_0}{L(1+C^{-1})^{1/2}}\right) \leq \frac{\mu_0 n}{T}\left(6 + \frac{2}{L(1+C^{-1})^{1/2}}\right) = \frac{\mu_0 n}{T}d_0,$$

which implies $(1+\theta)^q \leq e$. Let $c_q^k = c_q := 0$, then for any $k \geq 0$, we have

$$c_t^k \leq c_0^k = \beta^2(L + \frac{\mu^{-1}}{2})(1+C^{-1})\frac{2L^2}{n} \cdot \frac{(1+\theta)^q - 1}{\theta}$$

$$= \frac{2\mu_0^2 n(L + \frac{\mu^{-1}}{2})((1+\theta)^q - 1)}{T^{\frac{4}{3}}\theta} = \frac{2\mu_0^2(L + \frac{\mu^{-1}}{2})((1+\theta)^q - 1)}{2\mu_0 T^{\frac{1}{3}} + 4\mu_0^2 + \frac{2\mu_0^2}{L(1+C^{-1})^{\frac{1}{2}}}T^{\frac{1}{3}}}$$

$$\leq \frac{2\mu_0^2(L + \frac{\mu^{-1}}{2})((1+\theta)^q - 1)}{2\mu_0 T^{\frac{1}{3}}} \leq \frac{\mu_0(L + \frac{\mu^{-1}}{2})(e - 1)}{T^{\frac{1}{3}}}.$$

Therefore, it follows that

$$
\begin{aligned}
\Psi_t^k &= \frac{1}{2}\beta\mu - 2c_{t+1}^k\beta\eta^{-1}(1+C^{-1}) - 2L\beta^2(1+C^{-1}) - 4c_{t+1}^k\beta^2(1+C^{-1}) \\
&= \frac{1}{2}\frac{\mu_0 n\mu}{L(1+C^{-1})^{\frac{1}{2}}T^{\frac{2}{3}}} - 2c_{t+1}^k\frac{\mu_0 n}{L^2 T^{\frac{1}{3}}} - \frac{2\mu_0^2 n^2}{LT^{\frac{4}{3}}} - 4c_{t+1}^k\frac{\mu_0^2 n^2}{L^2 T^{\frac{4}{3}}} \\
&\geq \frac{1}{2}\frac{\mu_0 n\mu}{L(1+C^{-1})^{\frac{1}{2}}T^{\frac{2}{3}}} - \frac{2\mu_0^2(L+\frac{\mu^{-1}}{2})(e-1)n}{L^2 T^{\frac{2}{3}}} - \frac{2\mu_0^2 n^2}{LT^{\frac{4}{3}}} - \frac{4\mu_0^3(L+\frac{\mu^{-1}}{2})(e-1)n^2}{L^2 T^{\frac{5}{3}}} \\
&\geq \frac{n}{LT^{\frac{2}{3}}}\left(\frac{\mu_0\mu}{2(1+C^{-1})^{\frac{1}{2}}} - \frac{2\mu_0^2(L+\frac{\mu^{-1}}{2})(e-1)}{L} - 2\mu_0^2 n - \frac{4\mu_0^3(L+\frac{\mu^{-1}}{2})(e-1)n}{L}\right) \\
&\geq \frac{n\nu}{LT^{\frac{2}{3}}}.
\end{aligned}
$$

As a result, we have

$$
\sum_{t=0}^{q-1}\mathbb{E}\left[\|\nabla f(x_t^k)\|_2^2\right] \leq \frac{\Phi_0^k - \Phi_q^k}{\min_t \Psi_t^k} = \frac{\mathbb{E}\left[f(\tilde{x}_k) - f(\tilde{x}_{k+1})\right]}{\min_t \Psi_t^k},
$$

which yields that

$$
\mathbb{E}\left[\|\nabla f(x)\|_2^2\right] = \frac{1}{qN}\sum_{k=0}^{N-1}\sum_{t=0}^{q-1}\mathbb{E}\left[\|\nabla f(x_t^k)\|_2^2\right] \leq \frac{f(x_0) - f^{low}}{qN\min_t \Psi_t^k} \leq \frac{T^{2/3}L(f(x_0) - f^{low})}{qNn\nu}.
$$

To achieve $\mathbb{E}\left[\|\nabla f(x)\|_2^2\right] \leq \epsilon$, the outer iteration number $N$ of Algorithm 2 should be in the order of $O\left(\frac{T^{2/3}}{qn\epsilon}\right) = O\left(\frac{T^{-1/3}}{\epsilon}\right)$, which is due to the fact that $qn = O(T)$. As as result, the total number of $\mathcal{SFO}$-calls is $T + (T + 2qn)N$ (taking the first iteration into account), which is $O(T + T^{2/3}/\epsilon)$. $\qquad\square$

## A.2 Relationship with GMRES

Although the previous worst-case analysis shows that Anderson mixing has similar convergence rate as SGD, Anderson mixing usually performs much better in practice. To explain this phenomenon, we briefly discuss the relationship of Anderson mixing with GMRES [46] for deterministic quadratic optimization since a twice continuously differentiable objective function can be approximated by a quadratic model in a local region, thus leading to a quadratic optimization. An optimization method that performs well in quadratic optimization is likely to have good convergence property as well in general nonlinear optimization.

We consider the following strongly convex quadratic objective function:

$$
f(x) = \frac{1}{2}x^{\mathrm{T}}Ax - b^{\mathrm{T}}x. \tag{73}
$$

where $A \in \mathbb{R}^{d\times d}$ is symmetric positive definite, $b \in \mathbb{R}^d$. Solving (73) is equivalent to solving linear system

$$
Ax = b. \tag{74}
$$

In this case, $\nabla f(x) = Ax - b, \nabla^2 f(x) = A$, $r_k = b - Ax_k$ is the residual. Hence the quadratic approximation in AM is always exact, i.e. $R_k = -AX_k = -\nabla^2 f(x_k)X_k$.

When neither regularization nor damping is used, i.e. $\delta_k = 0$ and $\alpha_k = 1$, SAM is identical to AM to accelerate fixed-point iteration $g(x) = (I - A)x + b$. It has been proved in [59] that in exact arithmetic, AA is essentially equivalent to GMRES when starting from the same initial point and no stagnation occurs. We restate the main result here.

Let $x_k^{\mathrm{G}}, r_k^{\mathrm{G}} \overset{\text{def}}{=} b - Ax_k^{\mathrm{G}}$ denote the $k$-th GMRES iterate and residual, respectively, and $\mathcal{K}_k(A, v) \overset{\text{def}}{=} span\{v, Av, \ldots, A^{k-1}v\}$ denotes the $k$-th Krylov subspace generated by $A$ and $v$. Define $e^j \overset{\text{def}}{=} (1, 1, \ldots, 1)^{\mathrm{T}} \in \mathbb{R}^j$ for $j \geq 1$. For brevity, let $span(X)$ denote the linear space spanned by the columns of $X$. Besides, $\{x_k\}$ are the iterates generated by AM (SAM), and $\{\bar{x}_k\}$ are the intermediate iterates generated by (4a). We have

**Proposition 2.** *To minimize* (73)*, suppose that for SAM,* $\delta_k = 0, \alpha_k = \beta_k = 1, m = k \geq 1$*. If* $x_0 = x_0^{\mathrm{G}}$ *and* $rank(R_k) = m$*, then* $\bar{x}_k = x_k^{\mathrm{G}}$*.*

*Proof.* Since $R_k = -AX_k$ and $A$ is nonsingular, we have $rank(X_k) = m$. We first show $span(X_k) = \mathcal{K}_k(A, r_0^{\mathrm{G}})$ by induction. We abbreviate $\mathcal{K}_k(A, r_0^{\mathrm{G}})$ as $\mathcal{K}_k$ in this proof.

First, $\Delta x_0 = r_0 = r_0^{\mathrm{G}}$. If $k = 1$, then the proof is complete. Then, suppose that $k > 1$ and, as an inductive hypothesis, that $span(X_{k-1}) = \mathcal{K}_{k-1}$. With (9) and noting that $\alpha_k = \beta_k = 1$, we have

$$
\begin{aligned}
\Delta x_{k-1} &= x_k - x_{k-1} \\
&= r_{k-1} - (X_{k-1} + R_{k-1})\Gamma_{k-1} \\
&= b - Ax_{k-1} - (X_{k-1} - AX_{k-1})\Gamma_{k-1} \\
&= b - A(x_0 + \Delta x_0 + \cdots + \Delta x_{k-2}) - (X_{k-1} - AX_{k-1})\Gamma_{k-1} \\
&= r_0 - AX_{k-1}e^{k-1} - (X_{k-1} - AX_{k-1})\Gamma_{k-1}.
\end{aligned}
\tag{75}
$$

Since $r_0 \in \mathcal{K}_{k-1}$, and by the inductive hypothesis $span(X_{k-1}) \subseteq \mathcal{K}_{k-1}$ which also implies $span(AX_{k-1}) \subseteq \mathcal{K}_k$, we know $\Delta x_{k-1} \in \mathcal{K}_k$, which implies $span(X_k) \subseteq \mathcal{K}_k$. Since we assume $rank(X_k) = m = k$ which implies $\dim(span(X_k)) = \dim(\mathcal{K}_k)$, we have $span(X_k) = \mathcal{K}_k$, thus completing the induction.

Recalling that to determine $\Gamma_k$, we solve the least squares problem (5) and $R_k = -AX_k$, we have

$$
\Gamma_k = \arg\min_{\Gamma \in \mathbb{R}^m} \|r_k + AX_k\Gamma\|_2.
\tag{76}
$$

Since $rank(AX_k) = rank(X_k) = m$, (76) has a unique solution. Also, since $r_k = b - Ax_k = b - A(x_0 + X_k e^k) = r_0 - AX_k e^k$, we have $r_k + AX_k\Gamma = r_0 - AX_k e^k + AX_k\Gamma = r_0 - AX_k\tilde{\Gamma}$, where $\tilde{\Gamma} = e^k - \Gamma$. So $\Gamma_k$ solves (76) if and only if $\tilde{\Gamma}_k = e^k - \Gamma_k$ solves

$$
\min_{\tilde{\Gamma} \in \mathbb{R}^m} \|r_0 - AX_k\tilde{\Gamma}\|_2,
\tag{77}
$$

which is the GMRES minimization problem. Since the solution of (77) is also unique, we have

$$
\bar{x}_k = x_k - X_k\Gamma_k = x_k - X_k(e^k - \tilde{\Gamma}_k) = x_0 + X_k\tilde{\Gamma}_k = x_k^{\mathrm{G}}.
$$

$\square$

If $R_k$ is rank deficient, then a stagnation occur in AM (SAM).

**Proposition 3.** *To minimize* (73)*, suppose that for SAM,* $\delta_k = 0, \alpha_k = \beta_k = 1, m = k \geq 1$*. If* $rank(R_k) = k$ *holds for* $1 \leq k < s$ *while failing to hold for* $k = s$*, where* $s > 1$*, then* $\bar{x}_s = \bar{x}_{s-1}$*.*

*Proof.* The rank deficiency of $X_s$ implies $\Delta x_{s-1} \in span(X_{s-1})$. Therefore, there exists $\gamma_s \in \mathbb{R}^{s-1}$ such that $\Delta x_{s-1} = X_{s-1}\gamma_s$. Partitioning $\tilde{\Gamma} \in \mathbb{R}^s$ in (77) as $\tilde{\Gamma} = (\eta_1, \eta_2)^{\mathrm{T}} \in \mathbb{R}^s$, where $\eta_1 \in \mathbb{R}^{s-1}, \eta_2 \in \mathbb{R}$, we have

$$
\begin{aligned}
X_s\tilde{\Gamma} &= (X_{s-1} \quad X_{s-1}\gamma_s) \begin{pmatrix} \eta_1 \\ \eta_2 \end{pmatrix} \\
&= X_{s-1}(\eta_1 + \gamma_s\eta_2),
\end{aligned}
$$

which implies

$$
r_0 - AX_s\tilde{\Gamma} = r_0 - AX_{s-1}(\eta_1 + \gamma_s\eta_2).
$$

Hence

$$
\min_{\tilde{\Gamma} \in \mathbb{R}^s} \|r_0 - AX_s\tilde{\Gamma}\|_2
\tag{78a}
$$

$$
= \min_{\eta_1 \in \mathbb{R}^{s-1}, \eta_2 \in \mathbb{R}} \|r_0 - AX_{s-1}(\eta_1 + \gamma_s\eta_2)\|_2.
\tag{78b}
$$

Since $AX_{s-1}$ has full rank, $\tilde{\Gamma}_{s-1} = \eta_1 + \gamma_s\eta_2$ is the unique solution to minimize (78b) and $\tilde{\Gamma}_s = (\eta_1, \eta_2)^{\mathrm{T}}$ minimizes (78a) while being not unique. Therefore, from the equivalence of (76) and (77), we conclude that

$$
\begin{aligned}
\bar{x}_s &= x_0 + X_s\tilde{\Gamma}_s = x_0 + X_{s-1}(\eta_1 + \gamma_s\eta_2) \\
&= x_0 + X_{s-1}\tilde{\Gamma}_{s-1} = \bar{x}_{s-1}.
\end{aligned}
$$

$\square$

When the stagnation happens, further iterations of AM cannot make any improvement. This is a potential numerical weakness of AM relative to GMRES, which does not break down upon stagnation before the solution has been found. At this point, switching to applying several fixed-point iteration $x_{k+1} = g(x_k)$ may help jump out of the stagnation [42].

In Section 2.3, we introduce the preconditioned mixing strategy. This form of preconditioning for AM is new as far as we know. We reveal its relationship with right preconditioned GMRES [47] here. Let $x_k^{\text{RG}}, r_k^{\text{RG}}$ denote the $k$-th right preconditioned GMRES iterate and residual. According to Proposition 9.1 in [47], with a fixed preconditioner $M$, then $x_k^{\text{RG}}$ in the right preconditioned GMRES minimize residual in the affine subspace $x_0 + \mathcal{K}_k\{M^{-1}A, M^{-1}r_0^{\text{RG}}\}$.

**Proposition 4.** *To minimize* (73)*, suppose that for preconditioned SAM (cf.* (17)*),* $\delta_k = 0, \alpha_k = \beta_k = 1, m = k \geq 1$ *and* $M_k = M$ *where $M$ is nonsingular. If $x_0 = x_0^{\text{RG}}$ and $rank(R_k) = m$, then* $\bar{x}_k = x_k^{\text{RG}}$.

*Proof.* Since $R_k = -AX_k$ and $A$ is nonsingular, we have $rank(X_k) = m$. We first show that $span(X_k) = \mathcal{K}_k(M^{-1}A, M^{-1}r_0^{\text{RG}})$ by induction. We abbreviate $\mathcal{K}_k(M^{-1}A, M^{-1}r_0^{\text{RG}})$ as $\mathcal{K}_k$ in this proof.

First, $\Delta x_0 = M^{-1}r_0 = M^{-1}r_0^{\text{RG}}$. If $k = 1$, then the proof is complete. Then, suppose that $k > 1$ and, as an inductive hypothesis, that $span(X_{k-1}) = \mathcal{K}_{k-1}$. With (17) and noting that $\alpha_k = \beta_k = 1$, we have

$$\begin{aligned}
\Delta x_{k-1} &= x_k - x_{k-1} \\
&= M^{-1}r_{k-1} - (X_{k-1} + M^{-1}R_{k-1})\Gamma_{k-1} \\
&= M^{-1}(b - Ax_{k-1}) - (X_{k-1} - M^{-1}AX_{k-1})\Gamma_{k-1} \\
&= M^{-1}b - M^{-1}A(x_0 + \Delta x_0 + \cdots + \Delta x_{k-2}) - (X_{k-1} - M^{-1}AX_{k-1})\Gamma_{k-1} \\
&= M^{-1}r_0 - M^{-1}AX_{k-1}e^{k-1} - (X_{k-1} - M^{-1}AX_{k-1})\Gamma_{k-1}, \quad (79)
\end{aligned}$$

where $\Gamma_{k-1}$ minimizes (5) since $\delta_{k-1} = 0$.

Since $M^{-1}r_0 \in \mathcal{K}_{k-1}$, and by the inductive hypothesis $span(X_{k-1}) \subseteq \mathcal{K}_{k-1}$ which also implies $span(M^{-1}AX_{k-1}) \subseteq \mathcal{K}_k$, we know $\Delta x_{k-1} \in \mathcal{K}_k$, which implies $span(X_k) \subseteq \mathcal{K}_k$. Since we assume $rank(X_k) = m = k$ which implies $\dim(span(X_k)) = \dim(\mathcal{K}_k)$, we have $span(X_k) = \mathcal{K}_k$, thus completing the induction.

Recalling that to determine $\Gamma_k$, we solve the least squares problem (5) and $R_k = -AX_k$, we have

$$\Gamma_k = \arg\min_{\Gamma \in \mathbb{R}^m} \|r_k + AX_k\Gamma\|_2. \quad (80)$$

Since $rank(AX_k) = rank(X_k) = m$, (80) has a unique solution. Also, since $r_k = b - Ax_k = b - A(x_0 + X_k e^k) = r_0 - AX_k e^k$, we have $r_k + AX_k\Gamma = r_0 - AX_k e^k + AX_k\Gamma = r_0 - AX_k\tilde{\Gamma}$, where $\tilde{\Gamma} = e^k - \Gamma$. So $\Gamma_k$ solves (80) if and only if $\tilde{\Gamma}_k = e^k - \Gamma_k$ solves

$$\min_{\tilde{\Gamma} \in \mathbb{R}^m} \|r_0 - AX_k\tilde{\Gamma}\|_2, \quad (81)$$

which is the right preconditioned GMRES minimization problem. Since the solution of (81) is also unique, we have

$$\bar{x}_k = x_k - X_k\Gamma_k = x_k - X_k(e^k - \tilde{\Gamma}_k) = x_0 + X_k\tilde{\Gamma}_k = x_k^{\text{RG}}.$$

$\square$

For preconditioned SAM, the preconditioner $M_k$ can vary from step to step, while the minimal residual property still holds, i.e. $\bar{x}_k = \arg\min_{x \in x_k + span\{X_k\}} \|b - Ax\|_2$.

**Remark 4.** *When the objective function is only approximately convex quadratic in a local region around the minimum, the relation between SAM and GMRES can only approximately hold. Nonetheless, SAM can often show superior behaviour in practice.*

# B  Implementation of AdaSAM/pAdaSAM

In this section, we give the implementation details of our methods AdaSAM and pAdaSAM, including the pseudocodes and the discussion of the extra computational cost.

**Algorithm 3** AdaSAM, namely our proposed SAM with $\delta_k$ chosen as (16). $optim(x_k, g_k)$ is an optimizer which updates $x_k$ given stochastic gradient $g_k$.

---

**Input**: $x_0 \in \mathbb{R}^d, m = 10, \alpha_k = 1, \beta_k = 1, c_1 = 10^{-2}, p = 1, \gamma = 0.9, \epsilon = 10^{-8}, max\_iter > 0$, optimizer $optim$.
**Output**: $x \in \mathbb{R}^d$

1:   $\Delta \hat{x}_0 = 0, \Delta \hat{r}_0 = 0$
2:   **for** $k = 0, 1, \cdots, max\_iter$ **do**
3:     $r_k = -\nabla f_{S_k}(x_k)$
4:     **if** $k > 0$ **then**
5:       $m_k = \min\{m, k\}$
6:       $\Delta \hat{x}_k = \gamma \cdot \Delta \hat{x}_{k-1} + (1 - \gamma) \cdot \Delta x_{k-1}$
7:       $\Delta \hat{r}_k = \gamma \cdot \Delta \hat{r}_{k-1} + (1 - \gamma) \cdot \Delta r_{k-1}$
8:       $\hat{X}_k = [\Delta \hat{x}_{k-m_k+1}, \Delta \hat{x}_{k-m_k+2}, \cdots, \Delta \hat{x}_k]$
9:       $\hat{R}_k = [\Delta \hat{r}_{k-m_k+1}, \Delta \hat{r}_{k-m_k+2}, \cdots, \Delta \hat{r}_k]$
10:    **end if**
11:    **if** $k > 0$ and $k \mod p = 0$ **then**
12:      $\delta_k = c_1 \|r_k\|_2^2 / \left( \|\Delta \hat{x}_k\|_2^2 + \epsilon \right)$
13:      $\Delta x_k = \beta_k r_k - \alpha_k \left( \hat{X}_k + \beta_k \hat{R}_k \right) \left( \hat{R}_k^{\mathrm{T}} \hat{R}_k + \delta_k \hat{X}_k^{\mathrm{T}} \hat{X}_k \right)^{\dagger} \hat{R}_k^{\mathrm{T}} r_k$
14:      **if** $(\Delta x_k)^{\mathrm{T}} r_k > 0$ **then**
15:        $x_{k+1} = x_k + \Delta x_k$
16:      **else**
17:        $x_{k+1} = optim(x_k, -r_k)$
18:      **end if**
19:    **else**
20:      $x_{k+1} = optim(x_k, -r_k)$
21:    **end if**
22:    Apply learning rate schedule of $\alpha_k, \beta_k$
23: **end for**
24: **return** $x_k$

---

### B.1   Pseudocode for AdaSAM/pAdaSAM

Algorithm 3 gives the pseudocode for AdaSAM. Based on the prototype Algorithm 1, we incorporate sanity check of the positive definiteness, alternating iteration and moving average in our implementation of AdaSAM:

1. **Sanity check of the positive definiteness**. Besides calculating the largest eigenvalue to check Condition (14), a rule of thumb is checking a necessary condition $r_k^T H_k r_k > 0$, i.e. the searching direction $\Delta x_k = H_k r_k$ is a descent direction with respect to the stochastic gradient $\nabla f_{S_k}(x_k)$. If this condition is violated, we switch to updating $x_k$ via $optim$. Although such rule of thumb is not theoretically justified, it causes no difficulty of convergence in our practice. (Line 14-17 in Algorithm 3.)

2. **Moving average**. In mini-batch training, moving average can be used to incorporate information from the past and reduce the variability. Specifically, we maintain the moving averages of $X_k, R_k$ by $\hat{X}_k, \hat{R}_k$ respectively. Here $\hat{X}_k = [\Delta \hat{x}_{k-m+1}, \Delta \hat{x}_{k-m+2}, \cdots, \Delta \hat{x}_k]$, $\hat{R}_k = [\Delta \hat{r}_{k-m+1}, \Delta \hat{r}_{k-m+2}, \cdots, \Delta \hat{r}_k]$, where $\Delta \hat{x}_k = \gamma \cdot \Delta \hat{x}_{k-1} + (1 - \gamma) \cdot \Delta x_{k-1}$, $\Delta \hat{r}_k = \gamma \cdot \Delta \hat{r}_{k-1} + (1 - \gamma) \cdot \Delta r_{k-1}$, $\Delta \hat{x}_0 = 0, \Delta \hat{r}_0 = 0$ and $\gamma \in [0, 1)$. For deterministic quadratic optimization, $\hat{R}_k = -\nabla^2 f(x_k) \hat{X}_k$ still holds. (Line 6-9 in Algorithm 3.)

3. **Alternating iteration**. To amortize the computational cost of SAM, it is reasonable to apply a form of alternating iteration like [42]. In each cycle, we iterate with $optim$ for $(p - 1)$ steps and update $X_k, R_k$ simultaneously, then apply SAM in the $p$-th step, the result of which is the starting point of the next cycle. (Line 11,19 in Algorithm 3.)

We point out that these three techniques are not required in our theoretical analysis in Section 3. Nonetheless, they may have positive effects in practice.

**Algorithm 4** pAdaSAM, namely the preconditioned AdaSAM. $optim(x_k, g_k)$ is an optimizer which updates $x_k$ given stochastic gradient $g_k$.

---

**Input**: $x_0 \in \mathbb{R}^d, m = 10, \alpha_k = 1, c_1 = 10^{-2}, p = 1, \gamma = 0.9, \epsilon = 10^{-8}, max\_iter > 0$, optimizer $optim$.
**Output**: $x \in \mathbb{R}^d$

1: $\Delta \hat{x}_0 = 0, \Delta \hat{r}_0 = 0$
2: **for** $k = 0, 1, \cdots, max\_iter$ **do**
3:     $r_k = -\nabla f_{S_k}(x_k)$
4:     **if** $k > 0$ **then**
5:         $m_k = \min\{m, k\}$
6:         $\Delta \hat{x}_k = \gamma \cdot \Delta \hat{x}_{k-1} + (1 - \gamma) \cdot \Delta x_{k-1}$
7:         $\Delta \hat{r}_k = \gamma \cdot \Delta \hat{r}_{k-1} + (1 - \gamma) \cdot \Delta r_{k-1}$
8:         $\hat{X}_k = [\Delta \hat{x}_{k-m_k+1}, \Delta \hat{x}_{k-m_k+2}, \cdots, \Delta \hat{x}_k]$
9:         $\hat{R}_k = [\Delta \hat{r}_{k-m_k+1}, \Delta \hat{r}_{k-m_k+2}, \cdots, \Delta \hat{r}_k]$
10:     **end if**
11:     **if** $k > 0$ and $k \bmod p = 0$ **then**
12:         $\delta_k = c_1 \|r_k\|_2^2 / \left( \|\Delta \hat{x}_k\|_2^2 + \epsilon \right)$
13:         $\Gamma_k = \left( \hat{R}_k^{\mathrm{T}} \hat{R}_k + \delta_k \hat{X}_k^{\mathrm{T}} \hat{X}_k \right)^\dagger \hat{R}_k^{\mathrm{T}} r_k$
14:         $\bar{x}_k = x_k - \alpha_k \hat{X}_k \Gamma_k$
15:         $\bar{r}_k = r_k - \alpha_k \hat{R}_k \Gamma_k$
16:         $\Delta x_k = optim(\bar{x}_k, -\bar{r}_k) - x_k$
17:         **if** $(\Delta x_k)^{\mathrm{T}} r_k > 0$ **then**
18:             $x_{k+1} = x_k + \Delta x_k$
19:         **else**
20:             $x_{k+1} = optim(x_k, -r_k)$
21:         **end if**
22:     **else**
23:         $x_{k+1} = optim(x_k, -r_k)$
24:     **end if**
25:     Apply learning rate schedule of $\alpha_k, \beta_k$
26: **end for**
27: **return** $x_k$

---

Our implementation of the RAM method, i.e. using constant regularization (cf. (10)), differs from AdaSAM by replacing Line 13 in Algorithm 3 with $\Delta x_k = \beta_k r_k - \alpha_k \left( \hat{X}_k + \beta_k \hat{R}_k \right) \left( \hat{R}_k^{\mathrm{T}} \hat{R}_k + \delta I \right)^\dagger \hat{R}_k^{\mathrm{T}} r_k$. In other words, we also incorporate the damped projection into the constant-regularized AM. Therefore, the comparison between AdaSAM and RAM can show the effect of adaptive regularization with $\delta_k$ chosen as (16).

We give the pseudocode of pAdaSAM in Algorithm 4, which is the preconditioned version of AdaSAM. (See Section 2.3.) The effect of the preconditioner $optim$ is reflected in Line 16 in Algorithm 4.

**Remark 5.** *In our implementations, we introduce several extra hyper-parameters to the prototype Algorithm 1. However, we will show in the additional experiments that the only hyper-parameter needed to be tuned is still the regularization parameter $c_1$, while setting the other hyper-parameters as default is proper. We also omit the second term $c_2 \beta_k^{-2}$ in (16), which we will justify in Appendix D.3.*

## B.2 Additional computational complexity of SAM

In Remark 2, we claim that the additional computational complexity of SAM compared with SGD is $O(m^2 d) + O(m^3)$, which accounts for the matrix multiplications ($\mathbb{R}^{m \times d} \times \mathbb{R}^{d \times m}$) and matrix decomposition of a small $\mathbb{R}^{m \times m}$ matrix. In fact, the computational complexity can be further reduced to $O(md) + O(m^3)$ as the matrix multiplications $X_k^{\mathrm{T}} X_k, R_k^{\mathrm{T}} R_k$ need not be calculated from scratch because we can reuse the submatrice of $X_{k-1}^{\mathrm{T}} X_{k-1}, R_{k-1}^{\mathrm{T}} R_{k-1}$ calculated in the last iteration. To

see this, consider $X_k^{\mathrm{T}} X_k (k \geq m)$. By column partitioning $X_k = ((X_k)_{1:m-1} \quad (X_k)_m)$, where we use $(A)_{i:j}$ to denote the submatrix formed by the $i$ to $j$ columns of matrix $A$, we have

$$
X_k^{\mathrm{T}} X_k = \begin{pmatrix} (X_k)_{1:m-1}^{\mathrm{T}}(X_k)_{1:m-1} & (X_k)_{1:m-1}^{\mathrm{T}}(X_k)_m \\ (X_k)_m^{\mathrm{T}}(X_k)_{1:m-1} & (X_k)_m^{\mathrm{T}}(X_k)_m \end{pmatrix}
$$
$$
= \begin{pmatrix} (X_{k-1})_{2:m}^{\mathrm{T}}(X_{k-1})_{2:m} & (X_k)_{1:m-1}^{\mathrm{T}}(X_k)_m \\ (X_k)_m^{\mathrm{T}}(X_k)_{1:m-1} & (X_k)_m^{\mathrm{T}}(X_k)_m \end{pmatrix}.
$$

Hence, the submatrix $(X_{k-1})_{2:m}^{\mathrm{T}}(X_{k-1})_{2:m}$ can be reused, and only $X_k^{\mathrm{T}}(X_k)_m$ needs to be calculated. $R_k^{\mathrm{T}} R_k$ can be similarly calculated. Thus, if we reuse the matrix multiplication as explained above, the additional computational time can be further reduced.

## C   Experimental details

Our codes were written in PyTorch1.4.0[4] and one GeForce RTX 2080 Ti GPU is used for each test. Our methods are AdaSAM and its preconditioned variant pAdaSAM. The RAM method serves as an ablation study for the regularization. Before describing the experimental details, we explain the hyperparameter setting of AdaSAM/pAdaSAM/RAM.

### C.1   Hyperparameter setting of AdaSAM/pAdaSAM/RAM

Since these methods are all the variants of AM with minor differences, their hyperparameter setting are similar. The only hyperparameter that needs to be carefully tuned is the regularization parameter, i.e. $c_1$ for AdaSAM/pAdaSAM, and $\delta$ for RAM. We explain reasons of the default setting of other hyperparamters:

- $\alpha_0 = 1$. Setting $\alpha_0 = 1$ corresponds to using no damping, which means that the minimal residual procedure is exact for AM in deterministic quadratic optimization. This setting follows the same philosophy of setting initial learning rate as 1 in Newton's method.

- $\beta_0 = 1$. Setting the mixing parameter $\beta_0 = 1$ is a standard setting in AM. (See e.g. [59].) Tuning $\beta_k$ may be of help [41, 14], but we abandon this possibility to reduce the work of hyperparameter tuning.

- $m = 10$. Since the extra space is $2md$ and the extra computational cost is $O(m^2 d) + O(m^3)$, using small $m$ is preferred. $m = 10$ or $20$ is also the default setting in restarted GMRES [47]. Moreover, large $m$, say, $m = 100$, can cause the solution of $\Gamma_k$ less stable as we solve the normal equation directly.

- $p = 1$. By default, no alternating iteration is used. When the extra computational cost (e.g. Line 13 in Algorithm 3) dominates the computation, this option can be helpful to alleviate the cost.

- $\epsilon = 10^{-8}$. $\epsilon$ only serves as the safe-guard to prevent the denominator in (16) from being zero. It does not have meaning like the constant regularization $\delta$ in RAM. Only when $\|\Delta \hat{x}_k\|_2^2 \approx \epsilon$, the effect of $\epsilon$ becomes obvious, but $x_k$ is supposed to converge at this point.

- $\gamma = 0.9$. This is a default setting for moving average [55, 29].

### C.2   Experiments on MNIST

Since SAM is expected to behave like the minimal residual method in deterministic quadratic optimization, this group of experiments focused on large mini-batch training where the variance of noise is relatively small, thus the curvature of the objective function rather than noise dominates the optimization. Moreover, using constant learning rate is proper in this situation.

The baselines are SGDM, Adam and SdLBFGS. For SGDM and Adam, we used the built-in PyTorch implementations. For SdLBFGS, in addition to the initial proposal [60], the Hessian is always

---

[4] Information about this framework is referred to https://pytorch.org.

initialized with the identity matrix and the calculated descent direction is normalized because such modifications were found to be more effective for SdLBFGS in our experiments.

We tuned the learning rates of the baseline optimizers by log-scale grid-searches from $10^{-3}$ to 100. The learning rates of SGDM, Adam and SdLBFGS were 0.1, 0.001 and 0.1, respectively. The historical lengths for SdLBFGS, RAM and AdaSAM were set as 20. $\delta = 10^{-6}$ for RAM and $c_1 = 10^{-4}$ for AdaSAM. The $optim$ in Algorithm 3 is $optim(x_k, g_k) = x_k - 0.1 * g_k$.

For the preconditioned AdaSAM, i.e. Adagrad-AdaSAM and RMSprop-AdaSAM, the learning rates of Adagrad and RMSprop were 0.01, 0.001, respectively.

For all the tests, the model was trained for 100 epochs.

In the main paper, we only report training loss. Here, we report both the training loss and the squared norm of the gradient (SNG) in Figure 4. By comparing the training loss and SNG, it can be observed that a smaller SNG typically indicates a smaller training loss, which confirms the way to minimize $\mathbb{E}\|\nabla f(x)\|_2^2$. Since AM is closely related to the minimal residual method, where the term "residual" is actually the negative gradient in optimization, AM is expected to achieve small SNG when the quadratic approximation of the objective function is accurate enough. From the experiments, we find the behaviour of AdaSAM is rather stable even in mini-batch training.

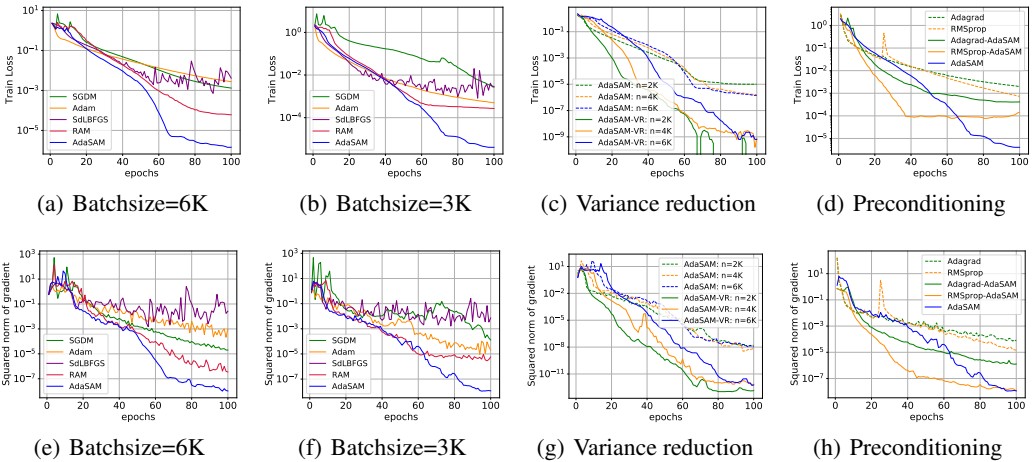

Figure 4: Experiments on MNIST. Train Loss: (a) Batchsize = 6K; (b) Batchsize = 3K; (c) AdaSAM with variance reduction; (d) Preconditioned AdaSAM with batchsize = 3K. Square norm of the gradient: (e) Batchsize = 6K; (f) Batchsize = 3K; (g) AdaSAM with variance reduction; (h) Preconditioned AdaSAM with batchsize = 3K.

## C.3 Experiments on CIFAR

For this group of experiments, we followed the same setting of training ResNet in [21]. The batchsize was 128 as commonly suggested. When training for $N$ iterations, the learning rate was decayed at the $\left(\lfloor \frac{N}{2} \rfloor\right)$-th and the $\left(\lfloor \frac{3N}{4} \rfloor\right)$-th iterations. For AdaSAM/RAM, the learning rate decay means decaying $\alpha_k, \beta_k$ simultaneously. The results in Table 1 come from repeated tests with 3 random seeds.

The baseline optimizers were SGDM, Adam, AdaBelief [66], Lookahead [65], RNA [52], AdaHessian [63]. AdaBelief is a recently proposed adaptive learning rate method to improve Adam. Lookahead is a $k$-step method, which can be seen as a simple sequence interpolation method. In each cycle, Lookahead iterates with an inner-optimizer $optim$ for $k$ steps and then interpolates the first and the last iterates to give the starting point of the next cycle. RNA is also an extrapolation method but based on the minimal polynomial extrapolation approach [5]. AdaHessian is a second-order optimizer which uses Hessian-vector products to approximate the diagonal of the Hessian.

We tuned the hyperparameters through experiments on CIFAR-10/ResNet20. For AdaSAM/RAM, we only tuned the regularization parameter as explained in Section C.1. For each optimizer, the hyperparameter setting that has the best final test accuracy on CIFAR-10/ResNet20 was kept unchanged

and used for the other tests. We list the hyperparameters of all the tested optimizers here. (Learning rate is abbreviated as lr.)

- **SGDM**: lr = 0.1, momentum = 0.9, weight-decay = $5 \times 10^{-4}$, lr-decay = 0.1.
- **Adam**: lr = 0.001, $(\beta_1, \beta_2) = (0.9, 0.999)$, weight-decay = $5 \times 10^{-4}$, lr-decay = 0.1.
- **AdaBelief**: lr = 0.001, $(\beta_1, \beta_2) = (0.9, 0.999)$, eps = $1 \times 10^{-8}$, weight-decay = $5 \times 10^{-4}$, lr-decay = 0.1.
- **Lookahead**: $optim$: SGDM (lr = 0.1, momentum = 0.9, weight-decay = $1 \times 10^{-3}$), $\alpha = 0.8$, steps = 10, lr-decay = 0.1.
- **AdaHessian**: lr = 0.15, $(\beta_1, \beta_2) = (0.9, 0.999)$, eps=$1 \times 10^{-4}$, hessian-power: 1, weight-decay: $5 \times 10^{-4}/0.15$, lr-decay: 0.1.
- **RNA**: lr = 0.1, momentum = 0.9, $\lambda = 0.1$, hist-length = 10, weight-decay = $5 \times 10^{-4}$, lr-decay = 0.1.
- **RAM**: $optim$: SGDM (lr = 0.1, momentum = 0, weight-decay = $1.5 \times 10^{-3}$), $\alpha_k = 1.0, \beta_k = 1.0, \delta = 0.1, p = 1, m = 10$, weight-decay = $1.5 \times 10^{-3}$, lr-decay = 0.06.
- **AdaSAM**: $optim$: SGDM (lr = 0.1, momentum = 0, weight-decay = $1.5 \times 10^{-3}$), $\alpha_k = 1.0, \beta_k = 1.0, c_1 = 0.01, p = 1, m = 10$, weight-decay = $1.5 \times 10^{-3}$, lr-decay = 0.06.

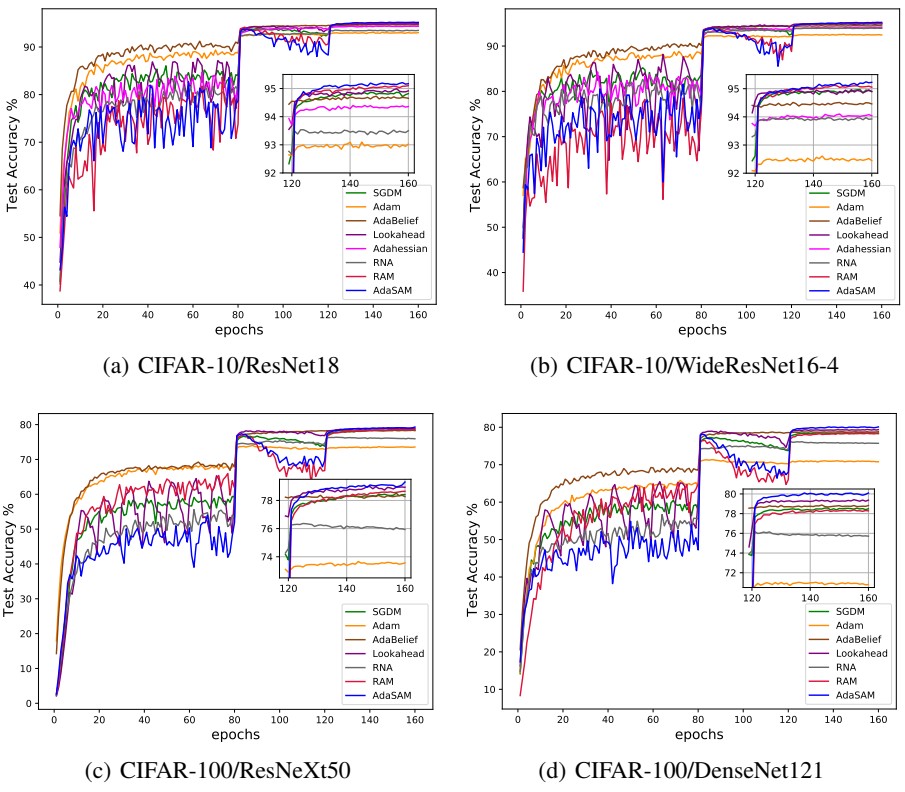

(a) CIFAR-10/ResNet18

(b) CIFAR-10/WideResNet16-4

(c) CIFAR-100/ResNeXt50

(d) CIFAR-100/DenseNet121

Figure 5: Test accuracy of ResNet18/WideResNet16-4 on CIFAR-10 and ResNeXt50/DenseNet121 on CIFAR-100.

Figure 5 shows the test accuracy of different optimizers for training ResNet18/WideResNet16-4 on CIFAR-10 and ResNeXt50/DenseNet121 on CIFAR-100. The full results of final test accuracy are listed in Table 1 in the main paper. From Figure 5, we find that the convergence behaviour of AdaSAM is rather erratic during the first 120 epochs. However, it always climbs up and stabilizes to the highest accuracy in the final 40 epochs. This phenomenon is due to the fact that AdaSAM uses a large weight-decay ($1.5 \times 10^{-3}$ vs. $5 \times 10^{-4}$ of SGDM) and large mixing parameter ($\beta_k = 1$). We verify this claim by doing tests on CIFAR-10/ResNet20. In Figure 6, we fixed other hyperparameters and tested

the effect of different weight-decays of AdaSAM. It is clear that a smaller weight-decay can lead to faster convergence on the training dataset, but often cause poorer generalization on the test dataset. In Figure 7, we only changed $\beta_0$ while fixing other hyperparameters (weight-decay=$1.5 \times 10^{-3}$). We can see a smaller $\beta_0$ can lead to faster and more stable convergence at the beginning, but the final test accuracy is suboptimal. This phenomenon coincides with the results in [34].

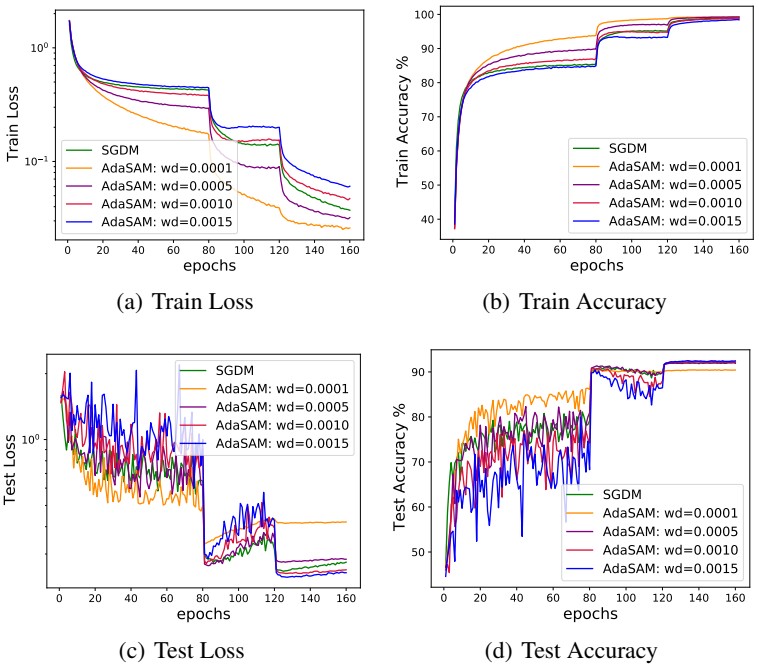

(a) Train Loss

(b) Train Accuracy

(c) Test Loss

(d) Test Accuracy

Figure 6: Experiments on CIFAR-10/ResNet20 with different weight-decay (abbreviated as wd in the legends). The weight-decay of SGDM is 0.0005.

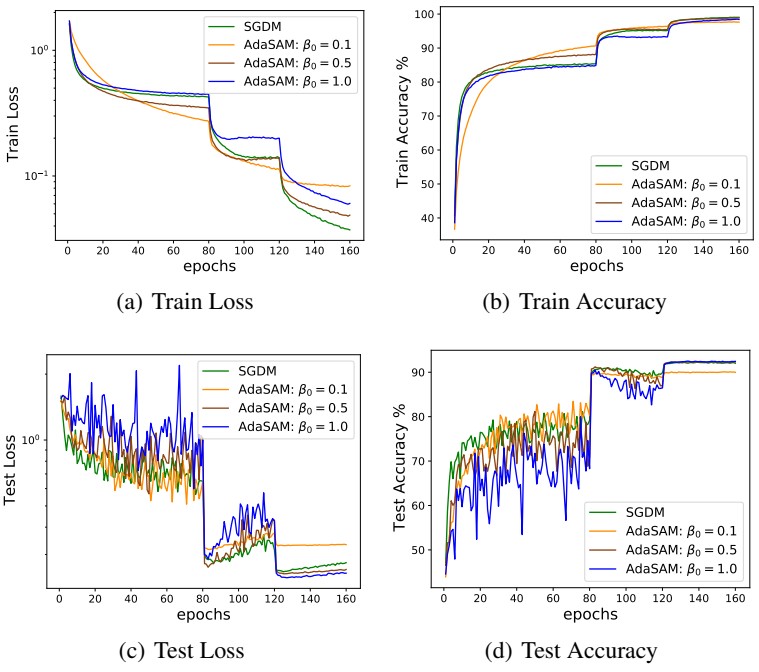

(a) Train Loss

(b) Train Accuracy

(c) Test Loss

(d) Test Accuracy

Figure 7: Experiments on CIFAR-10/ResNet20 with different $\beta_0$.

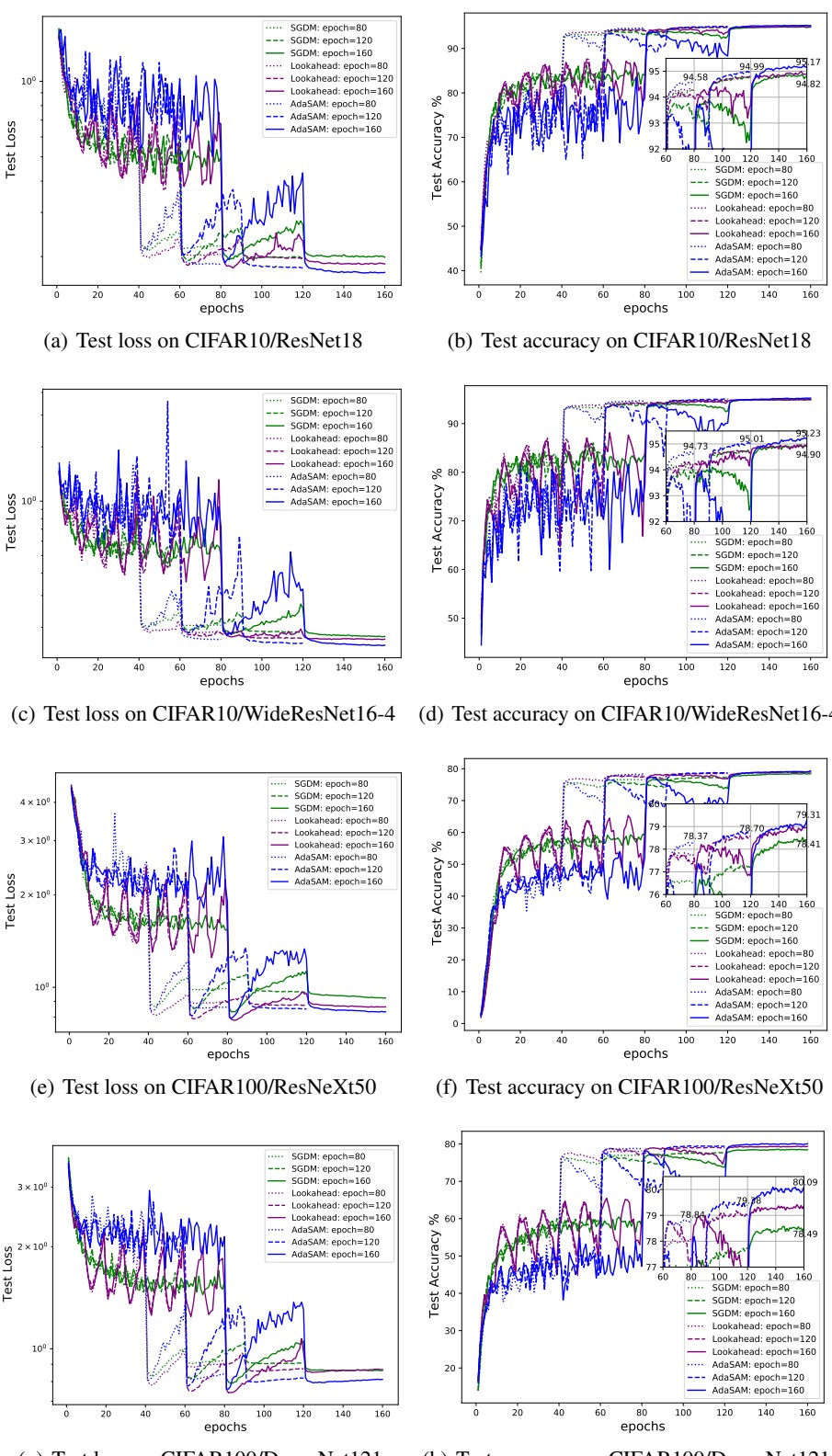

(a) Test loss on CIFAR10/ResNet18

(b) Test accuracy on CIFAR10/ResNet18

(c) Test loss on CIFAR10/WideResNet16-4

(d) Test accuracy on CIFAR10/WideResNet16-4

(e) Test loss on CIFAR100/ResNeXt50

(f) Test accuracy on CIFAR100/ResNeXt50

(g) Test loss on CIFAR100/DenseNet121

(h) Test accuracy on CIFAR100/DenseNet121

Figure 8: Training deep neural networks for 80,120,160 epochs. We report the final test accuracy of AdaSAM for training 80,120,160 epochs at nearby point in the nested figure. The final test accuracy of SGDM for training 160 epochs is also reported for comparison.

Since AdaSAM requires additional matrix computation in each iteration, it consumes more time if training for the same number of epochs as SGDM. Nonetheless, AdaSAM can achieve comparable test accuracy if decaying the learning rate earlier and stopping the training earlier. As indicated in Table 1, SGDM and Lookahead can serve as the strong baselines, so we conducted tests of comparisons between AdaSAM and SGDM/Lookahead to see the effectiveness of AdaSAM when training with fewer epochs. Results in Figure 8 show that the final test accuracy of AdaSAM for training 80 or 120 epochs can match or even surpass the test accuracy of SGDM for training 160 epochs. Therefore, the generalization benefit from AdaSAM pays for its additional cost.

In Table 2 in Section 4, we report the memory, per-epoch running time, and total running time of AdaSAM compared with SGDM. Here, we give more comprehensive results in Table 5, including the epochs and final test accuracy (AdaHessian ran out of memory in our device for training ResNeXt50 and DenseNet121). It shows that except for ResNet18, by using SGD as the baseline, AdaSAM can achieve a comparable solution within fewer training epochs and save computation time. Also, AdaSAM is far more efficient than the second-order optimizer AdaHessian.

**Remark 6.** *Since all the methods and neural networks were implemented and tested with the Python language, the measured memory and time may not exactly reflect the actual performance of each optimizer. Nonetheless, the observed results indicate that our method is efficient even though we did not do code optimization. Besides, we point out that the per-epoch running time of AdaSAM can be reduced by reusing matrix multiplications as discussed in Section B.2.*

Table 5: The cost and final test accuracy compared with SGDM. Memory, per-epoch time, total training epochs, total running time, and accuracy are abbreviated as "m","t/e", "e", "t", "a", respectively.

| Cost (× SGDM) & accuracy | CIFAR10/ResNet18 | | | | | CIFAR10/ResNet20 | | | | |
|---|---|---|---|---|---|---|---|---|---|---|
| | m | t/e | e | t | a(%) | m | t/e | e | t | a(%) |
| SGDM | 1.00 | 1.00 | 1.00 | 1.00 | 94.82 | 1.00 | 1.00 | 1.00 | 1.00 | 92.03 |
| AdaHessian | 2.33 | 5.78 | 1.00 | 5.78 | 94.36 | 1.78 | 3.59 | 1.00 | 3.59 | 91.92 |
| AdaSAM | 1.73 | 1.78 | 0.56 | 1.00 | 94.81 | 1.14 | 1.34 | 0.63 | 0.83 | 92.01 |

| Cost (× SGDM) & accuracy | CIFAR10/ResNet32 | | | | | CIFAR10/ResNet44 | | | | |
|---|---|---|---|---|---|---|---|---|---|---|
| | m | t/e | e | t | a(%) | m | t/e | e | t | a(%) |
| SGDM | 1.00 | 1.00 | 1.00 | 1.00 | 92.86 | 1.00 | 1.00 | 1.00 | 1.00 | 93.10 |
| AdaHessian | 2.01 | 4.10 | 1.00 | 4.10 | 92.18 | 2.15 | 4.66 | 1.00 | 4.66 | 92.74 |
| AdaSAM | 1.12 | 1.17 | 0.69 | 0.80 | 92.89 | 1.12 | 1.22 | 0.63 | 0.76 | 93.13 |

| Cost (× SGDM) & accuracy | CIFAR10/ResNet56 | | | | | CIFAR10/WideResNet16-4 | | | | |
|---|---|---|---|---|---|---|---|---|---|---|
| | m | t/e | e | t | a(%) | m | t/e | e | t | a(%) |
| SGDM | 1.00 | 1.00 | 1.00 | 1.00 | 93.47 | 1.00 | 1.00 | 1.00 | 1.00 | 94.90 |
| AdaHessian | 2.32 | 5.35 | 1.00 | 5.35 | 92.40 | 2.35 | 5.97 | 1.00 | 5.97 | 94.04 |
| AdaSAM | 1.21 | 1.35 | 0.63 | 0.84 | 93.47 | 1.26 | 1.28 | 0.63 | 0.80 | 94.94 |

| Cost (× SGDM) & accuracy | CIFAR100/ResNet18 | | | | | CIFAR100/ResNeXt50 | | | | |
|---|---|---|---|---|---|---|---|---|---|---|
| | m | t/e | e | t | a(%) | m | t/e | e | t | a(%) |
| SGDM | 1.00 | 1.00 | 1.00 | 1.00 | 77.27 | 1.00 | 1.00 | 1.00 | 1.00 | 78.41 |
| AdaHessian | 2.56 | 5.87 | 1.00 | 5.87 | 76.59 | >1.6 | - | - | - | - |
| AdaSAM | 1.85 | 1.79 | 0.56 | 1.00 | 77.33 | 1.30 | 1.16 | 0.50 | 0.58 | 78.37 |

| Cost (× SGDM) & accuracy | CIFAR100/DenseNet121 | | | | |
|---|---|---|---|---|---|
| | m | t/e | e | t | a(%) |
| SGDM | 1.00 | 1.00 | 1.00 | 1.00 | 78.49 |
| AdaHessian | >1.8 | - | - | - | - |
| AdaSAM | 1.16 | 1.19 | 0.50 | 0.60 | 78.84 |

## C.4 Experiments on Penn Treebank

Our experimental setting on training LSTM models on Penn Treebank dataset was based on the official implementation of AdaBelief [66]. Results in Table 3 were measured across 3 repeated runs with independent initialization. The parameter setting of the LSTM models are the same as that

of AdaBelief. The baseline optimizers are SGDM, Adam, AdaBelief and Lookahead. We tuned hyperparameters on the validation dataset for each optimizer.

For SGDM, we tuned the learning rate (abbr. lr) via grid-search in $\{1, 10, 30, 100\}$ and found that lr=10 performs best on 2,3-layer LSTM. For 1-layer LSTM, we set lr=30 and momentum=0 as that in AdaBelief because we found such setting is better.

For Adam, we tuned the learning rate via grid-search in $\{1 \times 10^{-3}, 2 \times 10^{-3}, 5 \times 10^{-3}, 8 \times 10^{-3}, 1 \times 10^{-2}, 2 \times 10^{-2}\}$ and found $5 \times 10^{-3}$ performs best.

For AdaBelief, we tuned the learning rate and found $5 \times 10^{-3}$ is better than $1 \times 10^{-2}$ used in [66].

For Lookahead, as suggested by the authors in [65], Adam with best hyperparameter setting is set as the inner optimizer, then the interpolation parameter $\alpha = 0.5$ and steps = 5.

The batch size is 20. We trained for 200 epochs and decayed the learning rate by 0.1 at the 100th and 150th epoch. For pAdaSAM, since the learning rate decay has been applied to the inner optimizer, we did not apply decay to $\alpha_k$, i.e. $\alpha_k = 1$ is kept unchanged during the training.

Table 6: Test perplexity on Penn Treebank for 1,2,3-layer LSTM. Lower is better. AdaSAM* denotes AdaSAM with $\beta_0 = 100$.

| Method | 1-Layer | 2-Layer | 3-Layer |
|---|---|---|---|
| SGDM | $85.21\pm.36$ | $67.12\pm.14$ | $61.56\pm.14$ |
| Adam | $80.88\pm.15$ | $64.54\pm.18$ | $60.34\pm.22$ |
| AdaBelief | $82.41\pm.46$ | $65.07\pm.02$ | $60.64\pm.14$ |
| Lookahead | $82.01\pm.07$ | $66.43\pm.33$ | $61.80\pm.10$ |
| AdaSAM | $155.38\pm.35$ | $159.07\pm1.58$ | $163.60\pm.81$ |
| AdaSAM* | $91.23\pm.69$ | $68.53\pm.13$ | $63.74\pm.09$ |
| pAdaSAM | $\mathbf{79.34\pm.09}$ | $\mathbf{63.18\pm.22}$ | $\mathbf{59.47\pm.08}$ |

Our method is pAdaSAM, which set $optim$ = Adam in Algorithm 4, where Adam is the tuned baseline. AdaSAM with the default setting is not suitable for this task. To give a full view of the vanilla AdaSAM, we also report the results of AdaSAM with default setting and the tuned AdaSAM ($\beta_0 = 100$) in Table 6.

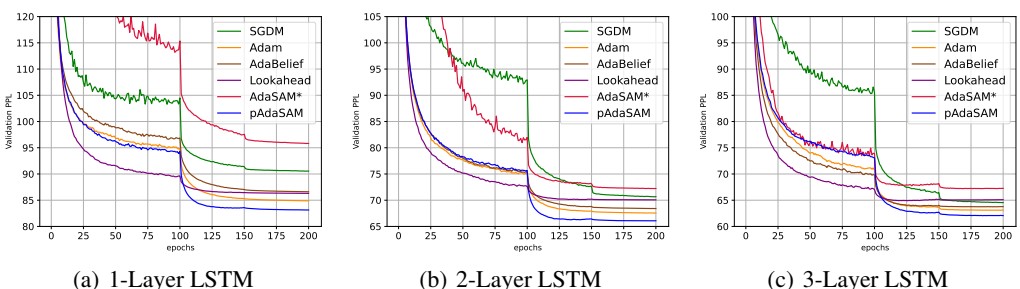

(a) 1-Layer LSTM         (b) 2-Layer LSTM         (c) 3-Layer LSTM

Figure 9: Experiments on Penn Treebank. Validation perplexity of training 1,2,3-Layer LSTM.

We think the scaling of the model's parameters is important for this problem. Since the batch size is very small, the gradient estimation is too noisy to capture the curvature information of the objective function. Hence the quadratic approximation in AdaSAM is rather inaccurate and further scaling by $\beta_k$ is required. For pAdaSAM, the scaling of the stochastic gradient is done by the inner optimizer Adam, so $\alpha_k$ can be set as default.

# D    Additional experiments

This section is about the techniques and hyperparameters used in our method. The computational costs with different batch sizes are also reported in the end.

## D.1 Check of positive definiteness

In Algorithm 3, we simplify the check of positive definiteness of $H_k$ described in Section 2.2 by $(\Delta x_k)^{\mathrm{T}} r_k > 0$. To see the effect of such simplification, we first give the pseudocode in Algorithm 5 that faithfully follows the procedure of checking positive definiteness in Section 2.2. We designate it as AdaSAM0. Note that the check of (14) is reflected in Line 17 in Algorithm 5.

We compared AdaSAM (Algorithm 3) with AdaSAM0 via experiments on MNIST and CIFAR-10/ResNet20. The experimental setting was the same as that in Section C.

The results on MNIST are shown in Figure 10. We trained 100 epochs with full-batch (batch-size=12K) and mini-batch (batchsize=3K). The evolution of $\alpha_k$ in Figure 10(c) implies that the Hessian approximation $H_k$ in AdaSAM ( $\alpha_k = \beta_k = 1$ ) is hardly positive definite. So AdaSAM0 reduces $\alpha_k$ to ensure Condition (13) holds. However, in full-batch training, there exists no noise in gradient evaluations, so $r_k^{\mathrm{T}} H_k r_k > 0$ is suffice to ensure $H_k r_k$ is a descent direction. In other words, Condition (13) may be too stringent to prevent the acceleration effect of AdaSAM. We also find that even running without checking of positive definiteness, the result is comparable. The result of AdaSAM0 with $\mu = 0.9$ suggests that the optimization is trapped in a local minima. In mini-batch training, Condition 13 is violated more frequently if using $\alpha_k = 1$, which can be inferred from the evolution of $\alpha_k$ of AdaSAM. Switching to *optim* when $(\Delta x_k)^{\mathrm{T}} r_k \leq 0$ is better than ignoring the violation of the positive definiteness. For AdaSAM0, we find using small $\mu$ is proper.

The results on CIFAR-10/ResNet20 are shown in Figure 11. We tested AdaSAM0 with different selections of $\mu$. We see smaller $\mu$ is better. We also find that the value of $\lambda_k$ is restrictive in training ResNet20. For $\mu = 0$, the Condition (14) is seldom violated during training, which means $\alpha_k$ is not need to be reduced to a smaller value to make $H_k$ positive definite. Therefore, AdaSAM0 with $\mu = 0$ has nearly the same behaviour as AdaSAM.

With these tests, we confirm that using the sanity check of positive definiteness of $H_k$ in Algorithm 3 does not lead to any deterioration.

---

**Algorithm 5** AdaSAM0. AdaSAM with the check of (13)

---

**Input**: $x_0 \in \mathbb{R}^d, m = 10, \alpha_k = 1, \beta_0 = 0.1, \beta_k = 1 (k \geq 1), \gamma = 0.9, \mu = 10^{-8}, \epsilon = 10^{-8}, max\_iter > 0$.
**Output**: $x \in \mathbb{R}^d$

1:  $\Delta \hat{x}_0 = 0, \Delta \hat{r}_0 = 0$
2:  **for** $k = 0, 1, \ldots, max\_iter$ **do**
3:      $r_k = -\nabla f_{S_k}(x_k)$
4:      **if** $k = 0$ **then**
5:          $x_{k+1} = x_k + \beta_k r_k$
6:      **else**
7:          $m_k = \min\{m, k\}$
8:          $\Delta \hat{x}_k = \gamma \cdot \Delta \hat{x}_{k-1} + (1 - \gamma) \cdot \Delta x_{k-1}$
9:          $\Delta \hat{r}_k = \gamma \cdot \Delta \hat{r}_{k-1} + (1 - \gamma) \cdot \Delta r_{k-1}$
10:         $\hat{X}_k = [\Delta \hat{x}_{k-m_k+1}, \Delta \hat{x}_{k-m_k+2}, \cdots, \Delta \hat{x}_k]$
11:         $\hat{R}_k = [\Delta \hat{r}_{k-m_k+1}, \Delta \hat{r}_{k-m_k+2}, \cdots, \Delta \hat{r}_k]$
12:         $\delta_k = c_1 \|r_k\|_2^2 / (\|\Delta \hat{x}_k\|_2^2 + \epsilon)$
13:         $Y_k = \hat{X}_k + \beta_k \hat{R}_k, Z_k = \hat{R}_k^{\mathrm{T}} \hat{R}_k + \delta_k \hat{X}_k^{\mathrm{T}} \hat{X}_k$
14:         Compute $\lambda_k = \lambda_{max} \left( \begin{pmatrix} Y_k^{\mathrm{T}} \\ \hat{R}_k^{\mathrm{T}} \end{pmatrix} \begin{pmatrix} Y_k & \hat{R}_k \end{pmatrix} \begin{pmatrix} 0 & Z_k^{\dagger} \\ Z_k^{\dagger} & 0 \end{pmatrix} \right)$.
15:         $\tilde{\alpha}_k = \alpha_k$
16:         **if** $\lambda_k > 0$ **then**
17:             $\tilde{\alpha}_k = \min\{\alpha_k, 2\beta_k(1 - \mu)/\lambda_k\}$
18:         **end if**
19:         $x_{k+1} = x_k + \beta_k r_k - \tilde{\alpha}_k Y_k Z_k^{\dagger} \hat{R}_k^{\mathrm{T}} r_k$
20:     **end if**
21:     Apply learning rate schedule of $\alpha_k, \beta_k$
22: **end for**
23: **return** $x_k$

---

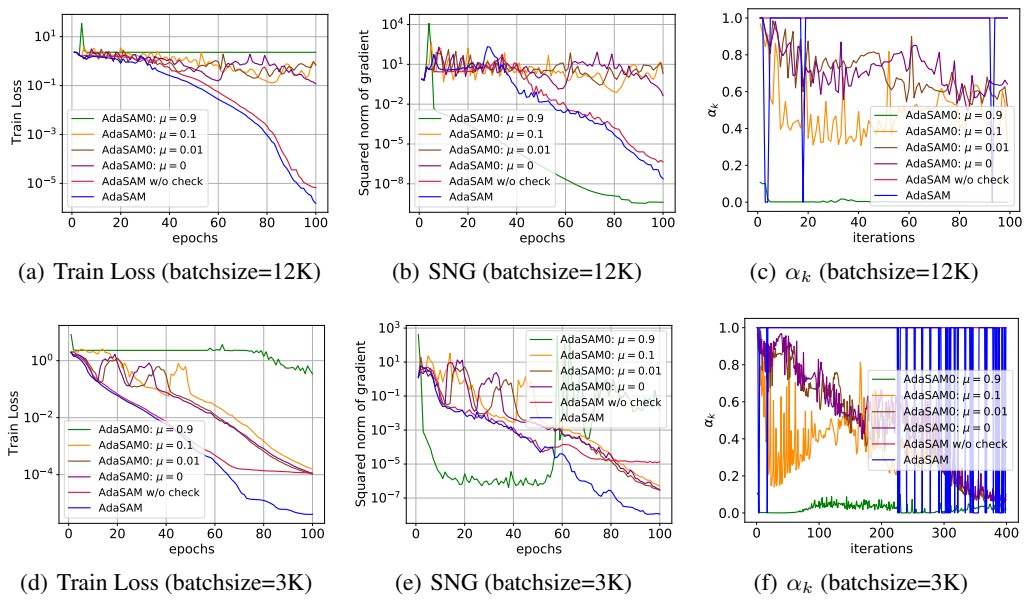

Figure 10: Experiments on MNIST. Training loss, squared norm of gradient (abbr. SNG) and $\alpha_k$ for batch size = 12K, 3K.

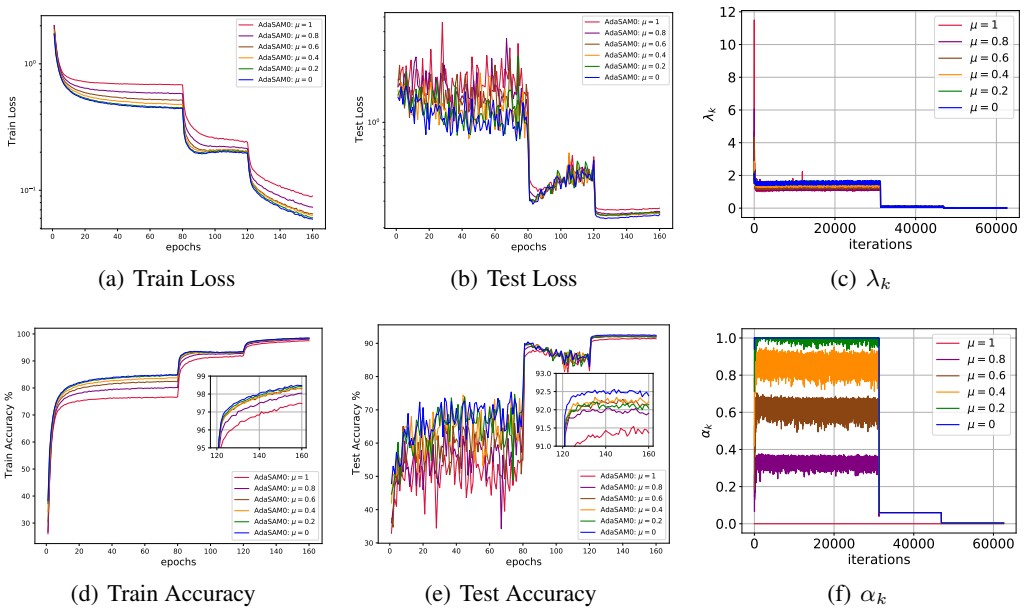

Figure 11: Experiments on CIFAT-10/ResNet20. Training with AdaSAM0 with $\mu = 0, 0.2, 0.4, 0.6, 0.8, 1$. (c) and (d) show the evolutions of $\lambda_k$ and $\alpha_k$ in AdaSAM0 during training.

## D.2 Effect of damped projection

Damped projection is introduced to overcome the weakness of the potential indefiniteness of $H_k$ in Anderson mixing. Its necessity has been justified in theory in Section 3. In practice, though we always initially set $\alpha_k = 1$ in AdaSAM, using damped projection can help improve the effectiveness. As shown in Figure 10, temporarily setting $\alpha_k = 0$ when $(\Delta x_k)^{\mathrm{T}} r_k \leq 0$ did improve convergence compared with the way of keeping $\alpha_k = 1$ unchanged. We also conducted tests on CIFAR-10/ResNet20,

where the learning rate decay of $\alpha_k$ was forbidden during training. The result is shown in Figure 12. We see the learning rate decay of $\alpha_k$ improves generalization.

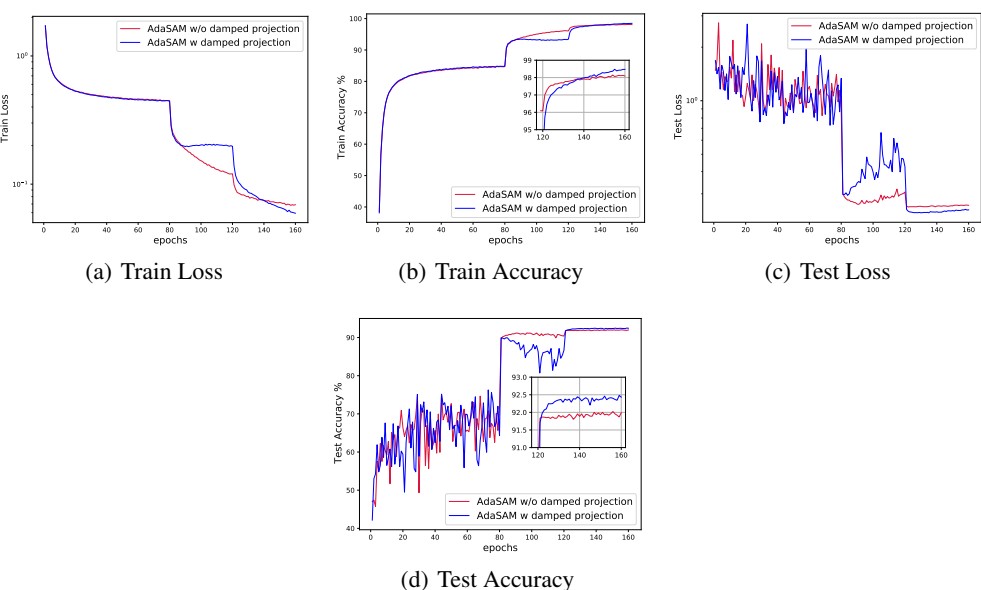

(a) Train Loss  (b) Train Accuracy  (c) Test Loss

(d) Test Accuracy

Figure 12: Experiments on CIFAT-10/ResNet20. Training with/without damped projection.

### D.3   Effect of adaptive regularization

AdaSAM is a special case of SAM with selection of $\delta_k$ as (16). Note that in our implementation Algorithm 3, we omit the term $c_2\beta_k^{-2}$. In fact, such special choice is important for SAM to be effective since it can roughly capture the curvature information. The comparison between AdaSAM and RAM in experiments on MNIST and CIFARs confirms the superiority of the regularization term of AdaSAM. Here, we further compare the choice of (16) with two other choices:

$$\delta_k = \delta, \qquad \text{(Option I)},$$
$$\delta_k = \delta\beta_k^{-2}, \quad \text{(Option II)},$$

We designate SAM with $\delta_k$ chosen as Option I (Option II) as SAM$^\dagger$ (SAM$^\ddagger$).

Table 7: Test accuracy on training CIFAR-10/ResNet20. The number on the first row are the regularization parameter $\delta$s.

|           | $10^3$        | $10^2$        | $10^1$        | $1$           | $10^{-1}$     | $10^{-2}$     | $10^{-3}$     |
|-----------|---------------|---------------|---------------|---------------|---------------|---------------|---------------|
| SAM$^\dagger$  | 91.57±.17 | 91.28±.27 | 91.40±.03 | 91.63±.01 | 91.65 ±.24 | 91.60±.24 | 91.67 ±.31 |
| SAM$^\ddagger$ | 91.48±.23 | 91.64±.30 | 91.74±.20 | 91.91±.19 | 91.62±.11 | 91.68±.34 | 91.48±.29 |

Experimental results on MNIST when training CNN with batch size of 12K, 3K are reported in Figure 13. Note that $c_1 = 10^{-4}$ was unchanged across two tests of different batch sizes. We see AdaSAM adaptively adjusts $\delta_k$ during training and always achieves the best result. On the contrary, the proper $\delta$ for SAM$^\dagger$ is dependent on the batch size.

For the tests on CIFAR-10/ResNet20, we made considerable efforts to tune $\delta$ in SAM$^\dagger$/SAM$^\ddagger$. The results corresponding to different $\delta$s are shown in Table 7. We also plot related curves of SAM$^\ddagger$ and AdaSAM in Figure 14, from which we see the $\delta_k$ determined in Line 12 in Algorithm 3 roughly matches the scheme of SAM$^\ddagger$, i.e. $\delta_k \geq C\beta_k^{-2}$ for some constant $C > 0$ , thus conforming our heuristic analysis about the convergence of AdaSAM in Section 3. Observed from Figure 14(c), we set $\delta = 0.5$ in SAM$^\ddagger$ to roughly match the evolution of $\delta_k$ in AdaSAM and obtain a slightly better test accuracy 92.05%. These results demonstrate the effectiveness of our choice of $\delta_k$ in AdaSAM.

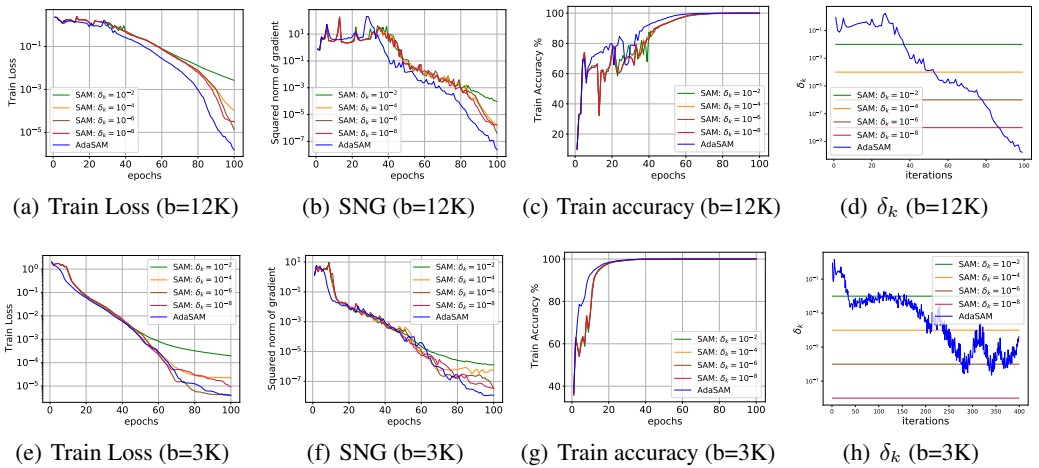

(a) Train Loss (b=12K)  (b) SNG (b=12K)  (c) Train accuracy (b=12K)  (d) $\delta_k$ (b=12K)

(e) Train Loss (b=3K)  (f) SNG (b=3K)  (g) Train accuracy (b=3K)  (h) $\delta_k$ (b=3K)

Figure 13: Experiments on MNIST. SAM$^\dagger$ with $\delta = 10^{-2}, 10^{-4}, 10^{-6}, 10^{-8}$ and AdaSAM. Training loss, training accuracy, squared norm of gradient (abbr. SNG) and $\delta_k$ with batchsize (abbr. b) of 12K, 3K are reported.

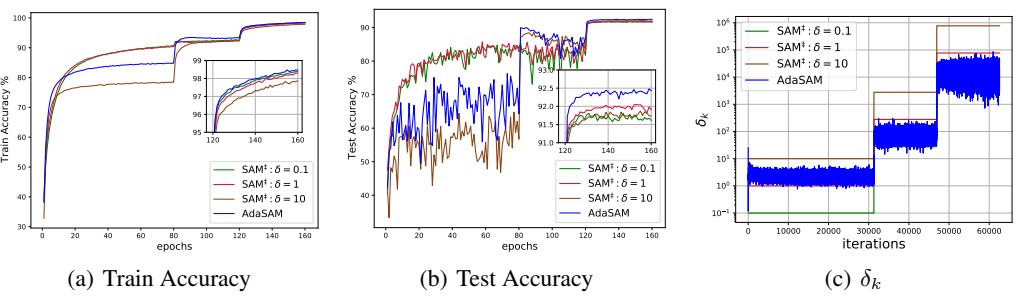

(a) Train Accuracy  (b) Test Accuracy  (c) $\delta_k$

Figure 14: Experiments on CIFAT-10/ResNet20. Training with SAM$^\ddagger$ with $\delta = 0.1, 1, 10$. (c) shows the evolution of $\delta_k$ in SAM$^\ddagger$ and AdaSAM during training.

## D.4 Moving average

For our implementation Algorithm 3 and Algorithm 4, we incorporate moving average as an option. In deterministic quadratic optimization, the minimal residual property still holds since the relation $\hat{R}_k = -\nabla f(x_k)\hat{X}_k$ is maintained. In general stochastic optimization, We find moving average may enhance the robustness to noise or generalization ability.

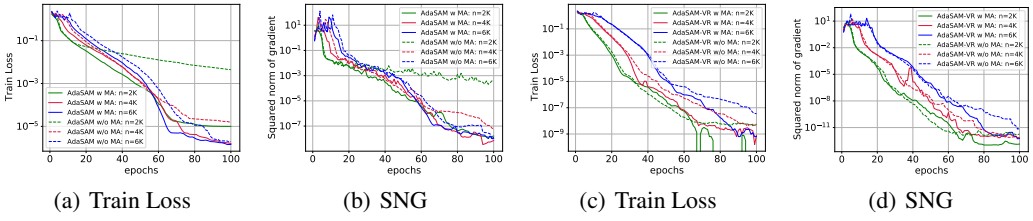

(a) Train Loss  (b) SNG  (c) Train Loss  (d) SNG

Figure 15: Experiments on MNIST. (a)(b) Training loss and square norm of gradient (abbr. SNG) of AdaSAM with/without moving average (abbr. MA). (c)(d) Training loss and SNG of AdaSAM-VR with/without MA. Batch size $n =$ 2K, 4K, 6K.

In Figure 15, we report AdaSAM/AdaSAM-VR with/without moving average for mini-batch training on MNIST. Figure 15(a) indicates that AdaSAM without moving average stagnates when batchsize is 2K due to noise in gradient estimates. By incorporating variance reduction, AdaSAM-VR without

moving average recovers the fast convergence rate. From this example, we conclude that moving average may help reduce the variability in gradient estimates and improve convergence.

We also reran the experiments on CIFAR-10/CIFAR-100 to see the effect of moving average. Results are reported in Table 8 and plotted in Figure 16. There seems to be no significant differences judging from final test accuracy, while AdaSAM without moving average can be faster at the beginning as indicated from Figure 16.

We reran the experiments on Penn Treebank. Results are shown in Table 9 and Figure 17. Similar to the phenomenon on CIFARs, pAdaSAM without moving average converges faster at the beginning. However, its final validation perplexity and test perplexity is slightly suboptimal compared with pAdaSAM with moving average.

With these experimental results, we think although moving average is not needed in our theoretical analysis, it may be beneficial in stabilizing the training or improving generalization ability.

Table 8: Experiments on CIFAR10/CIFAR100. WideResNet is abbreviated as WResNet.

| Method | CIFAR10 | | | | | | CIFAR100 | | |
|---|---|---|---|---|---|---|---|---|---|
| | ResNet18 | ResNet20 | ResNet32 | ResNet44 | ResNet56 | WResNet | ResNet18 | ResNeXt | DenseNet |
| AdaSAM w MA | 95.17±.10 | 92.43±.19 | 93.22±.32 | 93.57±.14 | 93.77±.12 | 95.23±.07 | 78.13±.14 | 79.31±.27 | 80.09±.52 |
| AdaSAM w/o MA | 95.22±.13 | 92.52±.09 | 93.08±.22 | 93.62±.05 | 93.89±.16 | 95.16±.04 | 78.09±.27 | 79.57±.21 | 80.03±.25 |

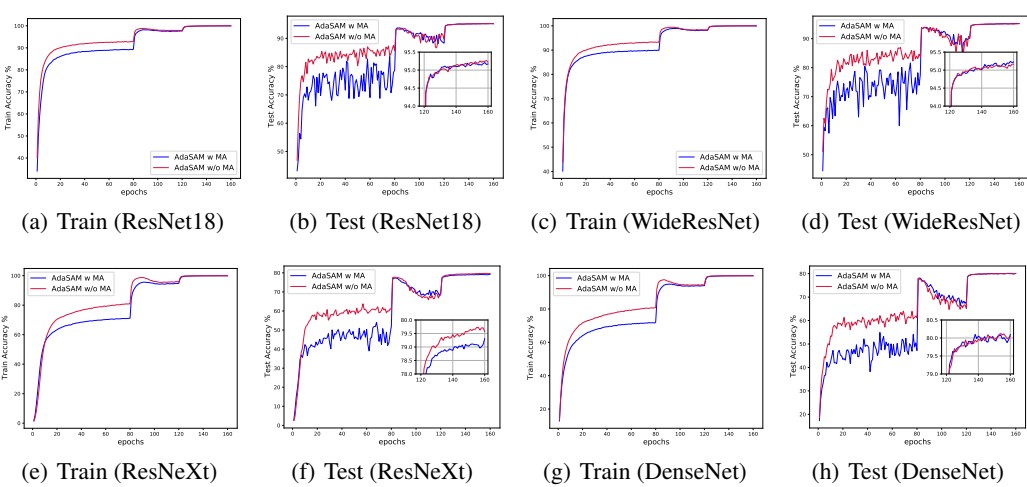

(a) Train (ResNet18)  (b) Test (ResNet18)  (c) Train (WideResNet)  (d) Test (WideResNet)

(e) Train (ResNeXt)  (f) Test (ResNeXt)  (g) Train (DenseNet)  (h) Test (DenseNet)

Figure 16: Experiments on CIFARs. Training CIFAR-10/ResNet18, CIFAR-10/WideResNet16-4, CIFAR-100/ResNeXt50, and CIFAR-100/DenseNet121 using AdaSAM with moving average (abbr. MA) or AdaSAM without moving average. Curves of training accuracy and test accuracy are reported.

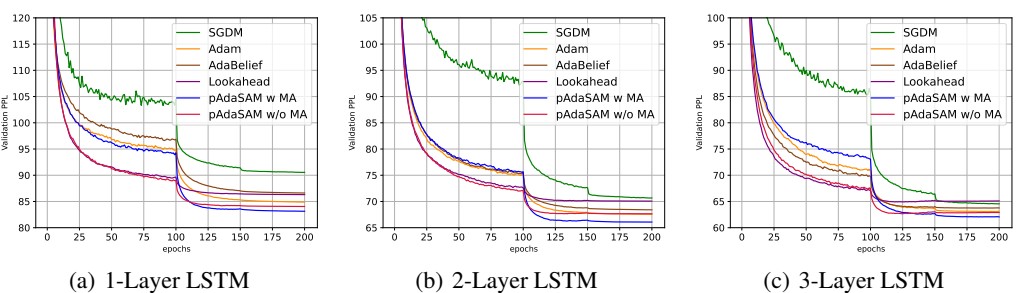

(a) 1-Layer LSTM  (b) 2-Layer LSTM  (c) 3-Layer LSTM

Figure 17: Experiments on Penn Treebank. Validation perplexity of training 1,2,3-Layer LSTM. Comparison between pAdaSAM with moving average (abbr. MA) and pAdaSAM without MA.

Table 9: Test perplexity on Penn Treebank for 1,2,3-layer LSTM. Comparison between pAdaSAM with moving average (abbr. MA) and pAdaSAM without MA.

| Method | 1-Layer | 2-Layer | 3-Layer |
|---|---|---|---|
| pAdaSAM w/o MA | 80.27±.09 | 64.74±.02 | 59.72±.05 |
| pAdaSAM w MA | 79.34±.09 | 63.18±.22 | 59.47±.08 |

## D.5 Alternating iteration

Alternating iteration is incorporated in Algorithm 3 and Algorithm 4: given an optimizer $optim$, in each cycle, we iterate with $optim$ for $(p-1)$ steps and then apply AdaSAM in the $p$-th step, the result of which is the starting point of the next cycle.

This option can be helpful to save per-iteration computational cost. We also tested this option in the experiments on CIFAR. We give some results here.

We tested vanilla SGD (momentum = 0) alternated with AdaSAM, and Adam alternated with AdaSAM, which are denoted as AdaSAM-SGD and AdaSAM-Adam respectively. The number of steps of one cycle is 5, i.e. $p = 5$. Results listed at the bottom of Table 10 show AdaSAM-Adam gives a thorough improvement over Adam. AdaSAM-SGD can even beat AdaSAM on CIFAR-100/ResNeXt50, and exceed the test accuracy of SGDM by 1.06%. Hence alternating iteration can reduce computational overhead while achieving comparable test accuracy.

Table 10: Final TOP1 test accuracy (%) on CIFAR10/CIFAR100. Alternating iterations are AdaSAM-SGD and AdaSAM-Adam.

| Method | CIFAR10 | | | | | | CIFAR100 | | |
|---|---|---|---|---|---|---|---|---|---|
| | ResNet18 | ResNet20 | ResNet32 | ResNet44 | ResNet56 | WResNet | ResNet18 | ResNeXt | DenseNet |
| SGDM | 94.82±.15 | 92.03±.16 | 92.86±.15 | 93.10±.23 | 93.47±.28 | 94.90±.09 | 77.27±.09 | 78.41±.54 | 78.49±.12 |
| Adam | 93.03±.07 | 91.17±.13 | 92.03±.28 | 92.28±.62 | 92.39±.23 | 92.45±.11 | 72.41±.17 | 73.57±.17 | 70.80±.23 |
| AdaSAM | **95.17±.10** | **92.43±.19** | **93.22±.32** | **93.57±.14** | **93.77±.12** | **95.23±.07** | **78.13±.14** | 79.31±.27 | **80.09±.52** |
| AdaSAM-SGD | 95.04±.22 | 92.26±.10 | 92.92±.28 | 93.01±.15 | 93.71±.15 | 94.99±.19 | 77.81±.12 | **79.47±.44** | 79.58±.39 |
| AdaSAM-Adam | 93.86±.23 | 92.27±.29 | 92.67±.09 | 92.94±.30 | 93.22±.12 | 93.88±.20 | 74.46±.51 | 75.34±.20 | 75.21±.49 |

## D.6 Additional experiments on MNIST

We provide some additional experiments on MNIST that is omitted in the main paper.

**Diminishing stepsize** Our theoretical analysis of SAM in Section 3 takes the diminishing condition (18a) as an assumption of $\beta_k$ in Theorems 1, 2, 3. Nonetheless, using constant stepsize and decaying after several epochs is a common way in practice. To test the diminishing condition, we set the $t$-th epoch learning rate for SGD/Adam/SdLBFGS and the $t$-th epoch mixing parameter $\beta_k$ for RAM/AdaSAM as $\eta_t = \eta_0(1 + \lfloor t/20 \rfloor)^{-1}$, where $t$ denotes the number of epochs, $\eta_0$ is tuned for each optimizer. For SGD, Adam and SdLBFGS, $\eta_0$ is 0.2, 0.001, 0.1, respectively. For RAM and AdaSAM, $\eta_0$ is 2. The results of training with batch sizes of 3K and 6K are reported in Figure 18. AdaSAM still shows the better convergence rate.

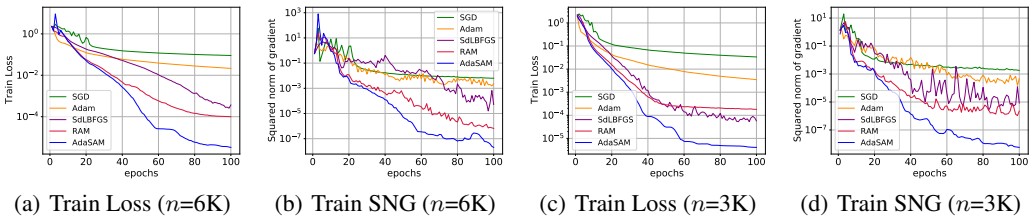

(a) Train Loss ($n$=6K)  (b) Train SNG ($n$=6K)  (c) Train Loss ($n$=3K)  (d) Train SNG ($n$=3K)

Figure 18: Experiments on MNIST (with diminishing stepsize). (a)(b) Training loss and square norm of gradient (abbr. SNG) using batchsize $n$= 6K; (c)(d) Training loss and SNG using $n$= 3K.

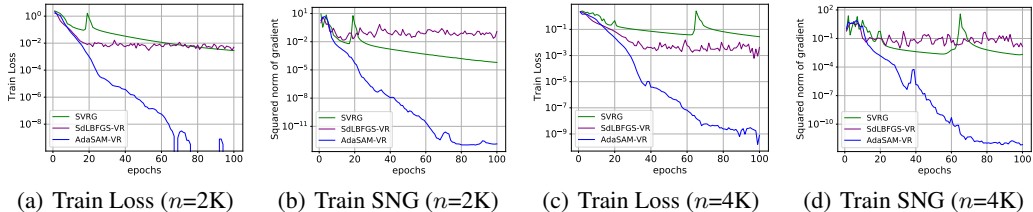

(a) Train Loss ($n$=2K)  (b) Train SNG ($n$=2K)  (c) Train Loss ($n$=4K)  (d) Train SNG ($n$=4K)

Figure 19: Experiments on MNIST (with variance reduction). (a)(b) Training loss and square norm of gradient (abbr. SNG) using batchsize $n$= 2K; (c)(d) Training loss and SNG using $n$= 4K.

**Comparisons with SVRG and SdLBFGS-VR**   We also compared with SVRG and SdLBFGS-VR [60]. The learning rates for SVRG and SdLBFGS-VR are 0.2 and 0.1. Results of training with batch sizes of 2K and 4K are shown in Figure 19. We find that AdaSAM-VR is more effective compared with the other two variance reduced optimizers.

### D.7   Discussion about the hyperparameters

As explained in Section C.1, though at first glance AdaSAM has several hyperparameters to tune, we actually only need to individually tune the regularization parameter $c_1$ besides other common hyperparameter such as weight-decay in almost all the cases. For example, setting $c_1 = 10^{-2}$ is fairly robust in our experiments in image classification on CIFARs and language model on Penn Treebank. We tested various deep neural networks on CIFAR-10 and CIFAR-100, while the hyperparameters were kept unchanged across different tests.

We conducted tests to see the effect of the historical length $m$ in AdaSAM and pAdaSAM. As pointed in [7] that the quasi-Newton updating is inherently an overwriting process rather than an average process, large noise in gradient estimates can make a secant method rather unstable. On the contrary, since AM is identified as a multisecant method, it leverages more secant conditions in one update which may alleviate the negative impact of a noisy secant condition. Hence, AM may be more tolerant to noise. The historical length $m$ determines how many secant conditions are taken into consideration at one time, so a larger $m$ is supposed to make AdaSAM more tolerant to noise.

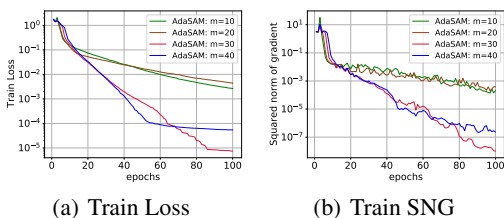

(a) Train Loss            (b) Train SNG

Figure 20: Experiments on MNIST. Training loss and square norm of gradient (abbr. SNG) using batchsize $n$= 2K. AdaSAM without moving average and $m = 10, 20, 30, 40$.

We set $m = 20$ in the experiments on MNIST. In Figure 15(a) and (b), we find AdaSAM without moving average stagnates when training with batchsize = 2K. We set $m$ to 10,30,40 to see if any difference happens. The result is shown in Figure 20, from which we see using a larger $m = 30$ did help convergence. Further increasing $m = 40$ does not lead to lower training loss, which may be due to the potential numerical weakness in solving (11) with (12) directly.

The results related to different $m$ in CIFAR-10/ResNet20 and 3-layer LSTM on Penn Treebank are reported in Figure 21. A larger $m$ seems to be beneficial to generalization ability. $m = 5$ or 10 is proper for these tests.

### D.8   Computational efficiency

The additional per-iteration computational cost of AdaSAM/pAdaSAM compared with SGD is mainly due to computing (12). The cost is a potential limitation of our method. Fortunately, this part of

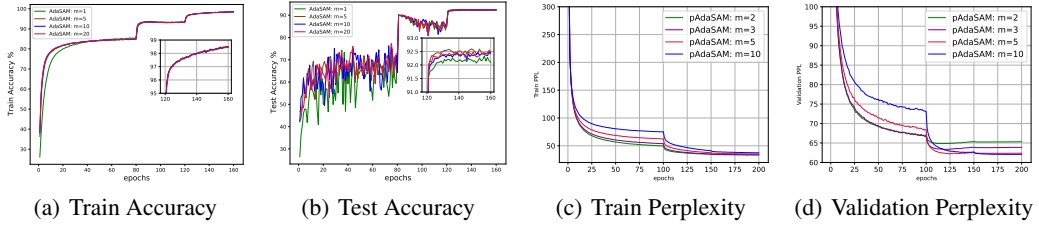

(a) Train Accuracy     (b) Test Accuracy     (c) Train Perplexity     (d) Validation Perplexity

Figure 21: (a)(b) Experiments on CIFAR-10/ResNet20. AdaSAM with $m = 1, 5, 10, 20$. (c)(d) Experiments on training a 3-layer LSTM on Penn Treebank. pAdaSAM with $m = 2, 3, 5, 10$.

computation is parallel-friendly since the main operation is dense matrix multiplications. Therefore, when the cost of function evaluations and gradient evaluations dominates the computation, the extra overhead incurred by AdaSAM is negligible. In high performance computing, we expect that the matrix computation in AdaSAM can be further optimized.

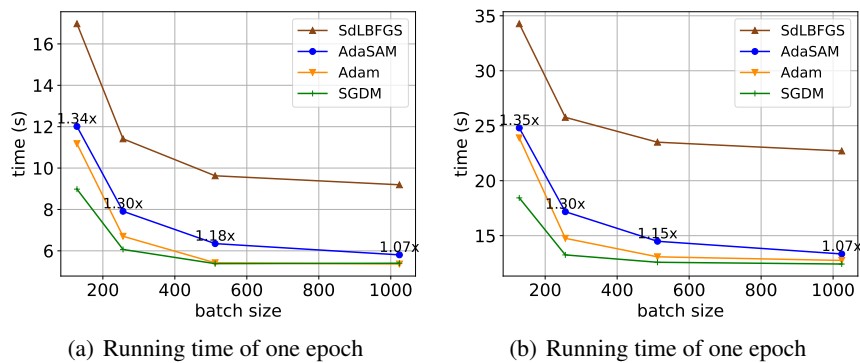

(a) Running time of one epoch        (b) Running time of one epoch

Figure 22: (a) Running time of one epoch of training CIFAR-10/ResNet20; (b) Running time of one epoch of training CIFAR-10/ResNet56. Batch size = 128, 256, 512, 1024. The numbers marked beside the curve of AdaSAM show the computational time of AdaSAM vs. SGDM.

Figure 22 reports the running time of one epoch of training ResNet20 and ResNet56 on CIFAR-10 with batch size of 128, 256, 512, 1024. Optimizers are SGDM, Adam, SdLBFGS and AdaSAM. It can be observed that the additional overhead of AdaSAM gradually becomes marginal with the increment of batch size. Therefore, AdaSAM is expected to be more computationally efficient in large mini-batch training. Also, AdaSAM is more efficient than the stochastic quasi-Newton method SdLBFGS, which is reasonable since SdLBFGS incurs 2x cost of gradient evaluations compared with AdaSAM, as discussed in Section 5. Finally, as confirmed by the experiments (Figure 8 and Table 5), AdaSAM can achieve comparable solutions while using fewer training epochs, which can save computation time. The proposed alternating iteration scheme can also serve as a trade-off between computational cost and final accuracy or loss.