# OpenReview forum: "Stochastic Anderson Mixing for Nonconvex Stochastic Optimization"
_NeurIPS.cc/2021/Conference — NeurIPS 2021 Poster_

### Official Review · Reviewer_TNM1 · 2021-07-16

**Rating:** 7
**Confidence:** 5

**Summary:**

The main idea of this paper is to introduce a stochastic Anderson mixing method which leads to faster convergence, accelerating the training. The main contribution of the paper is the convergence analysis part which gives the theoretical analysis of the proposed method.

**Ethics Review Area:**

["I don’t know"]

**Limitations And Societal Impact:**

The author(s) could extend the limitation part. Are there any situations where the convergence speed of SAM lags behind SGD? Could you please provide some specific analysis? So that readers can understand when to switch to SAM after reading.

**Main Review:**

Slow convergence is a notable and annoying problem during training, especially when training some nets with high complexity, such as GAN. The idea of using Anderson acceleration to improve the performance of the optimisation method is not novel, but I’m still impressed by the good performance of the AdaSAM. Actually, the core algorithm, using the damped projection and the adaptive regularisation, are not innovative enough. However, the convergence analysis and the abundant experiments shown the effectiveness of SAM, which gives me confidence.
Here are some questions that I had during the reading:
Line 41 - Why SAM is a second-order method?
The convergence analysis based on the assumption that f is continuously differentiable. Is it a realistic assumption?
Line 86 Remark 1 - The fixed point assumption is necessary actually, promising the equation in line 87 holds when j,k->\infinity.
Line 184 - Why the eq. 19 promises the convergence of the original problem? Is it possible to be stucked in a local minimal?
Figure 2: What is the training loss goes like when compared with Lookahead on CIFAR?
Figure 7 in appendix: It seems that the training loss of AdaSAM fell behind that of SGDM, does it means that AdaSAM cannot saving training time in some cases?
It seems that the cases shown here only needs hundreds epochs to converge. How well does the SAM perform in the more time-consuming cases?

**Time Spent Reviewing:**

3

---

> ### Author Response · Authors · 2021-08-10
> **Response to Reviewer TNM1**
>
> Thank you for your comments. We hope our following answers can address your questions.
>
> Q1. Why is SAM a second-order method?
> A1. Anderson mixing is identified as a multi-secant Newton method [13] which uses the past iterates and gradients to approximate the Hessian, i.e., the second-order information, of the objective function. From this perspective, since SAM is a stochastic version of Anderson mixing, we categorize SAM into the class of second-order methods.
>
> Q2. The continuously differentiable assumption of the objective function.
> A2: The continuously differentiable assumption is a standard assumption in stochastic optimization, and is assumed in many current optimizers, such as SGD [17], Adam [29], stochastic quasi-Newton method [8]. Without this assumption, we have to use the tool from nonsmooth analysis (e.g. subgradient), which makes the problem more complicated, and we leave it as a future work.
>
> Q3. The necessity of the fixed-point assumption.
> A3: In Remark 1, we claim that the fixed-point assumption is unnecessary because if $R_k \approx  -\nabla^2f(x_k)X_k$, the minimal residual property approximately holds for the projection step (Eq. (4a)). $X_k, R_k$ can be generated from any iteration as long as $x_{j-1}, x_j$ lie in a local small region around $x_k$ and the local quadratic approximation is nearly exact. For $j,k \rightarrow \infty$, $x_k, x_j$ should be close to a stationary point, which can be promised by a fixed-point assumption. From this viewpoint, the fixed-point assumption is useful. We will clarify it in revision.
>
> Q4. The concern about the convergence provided by $\\| \nabla f(x\_k)\\|\_2^2$.
> A4: Unlike convex problem, where $f(x)-f(x_*)$ or $ \\|x-x\_\*\\|\_2^2$ is used as the convergence criteria ($x_*$ denotes the optimum), the hardness of nonconvex problem makes it appropriate to use $ \\|\nabla f(x)\\|\_2^2<\epsilon $ as the convergence criteria, e.g., see [17] and Sec. 2 in [44]. This criteria for nonconvex problem can only guarantee the convergence to a stationary point.
>
> Q5. Training loss on CIFAR when compared with Lookahead.
> A5: In the CIFAR experiments, we empirically find that weight-decay can largely affect the training process. Larger weight-decay can cause slow training process due to its effect on alleviating overfitting. Following the same way of hyper-parameter tuning of Lookahead [64], the weight-decay hyper-parameter of each optimizer was tuned to achieve the best final performance. In our experiments, the weight decay parameter has the relationship: AdaSAM > Lookahead > SGDM. Thus, it can be found that the convergence rate for training loss is SGDM > Lookahead > AdaSAM.
>
> In Figure 6, we show the training loss with different weight decay in AdaSAM and it is shown that AdaSAM is faster than SGDM if both weight decay parameters are the same (purple vs. green). Furthermore, when the setting of the weight decay parameter is the same as that of SGDM, the detailed training and test profiles are given as follows.
>
> method  | training loss | training accuracy | test loss | test accuracy
> :-: | :-: | :-: | :-: | :-:
> SGDM        | 0.037±0.000 | 99.04±0.01 | 0.275±0.004 | 92.03±0.16
> Lookahead  | 0.043±0.000 | 98.81±0.03 | 0.293±0.03 | 91.62±0.17
> AdaSAM    | 0.032±0.000 | 99.25±0.04 | 0.284±0.007 | 91.93±0.18
>
>
> Q6. The efficiency of AdaSAM compared with SGDM.
> A6: Since AdaSAM uses much larger weight-decay ($1.5\times 10^{-3}$) than SGDM ($5\times 10^{-4}$), the training loss of AdaSAM falls behind that of SGDM. However, when considering the validation performance, Figure 8 in the appendix shows that with much less training epochs, AdaSAM can achieve comparable or even higher test accuracy than the fine-tuned SGDM, thus saving large computational cost. We list the final test accuracies of AdaSAM (with 120 epochs) and SGDM (with 160 epochs) here:
>
> method | CIFAR10/ResNet18 | CIFAR10/WideResNet16-4 | CIFAR100/ResNeXt50 | CIFAR100/DenseNet121
> :-: | :-: | :-: | :-: | :-:
> SGDM |  94.82±0.15 |  94.90±0.09 |  78.41±0.54 |   78.49±0.12
> AdaSAM | 94.99±0.09 |  95.01±0.13 |  78.70±0.32 |   79.38±0.05
>
> Q7. More time-consuming cases.
> A7: We test the performance of AdaSAM and SGDM for training ResNet18 on ImageNet on two GPUs. The hyperparameters in SGDM are set as the recommended setting in PyTorch. Due to the time and resource limit, we set the weight-decay parameter the same as the SGDM and do not tune the AdaSAM. Here are our experimental results.
>
> method | Top-1 training accuracy | Top-1 test accuracy | Top-5 training accuracy | Top-5 test accuracy | time (s)
> :-: | :-: | :-: | :-: | :-: | :-:
> SGDM |  69.04  |  69.81 | 87.30  |    89.25 |  202668
> AdaSAM |  69.13 | 70.16 | 87.33  |   89.34 | 195301
>
> Q8: The limitation of our work.
> A8: Thank you for suggesting us extending the limitation part. We will do it in revision.
> When the gradient estimation is very noisy, it is hard for SAM to capture any useful information from the past iterates and gradients, thus making the Hessian approximation very inaccurate. In this case, the potential acceleration brought by SAM can vanish. In our paper, we provide the preconditioned mixing strategy as a remedy, which can combine with SGDM or Adam as an implicit preconditioner.

---

### Official Review · Reviewer_wnT6 · 2021-07-16

**Rating:** 4
**Confidence:** 5

**Summary:**

In this manuscript, the authors proposes a Stochastic Anderson Mixing (SAM) scheme to solve nonconvex optimization problems. Under the smoothness, unbiased gradient estimate with bounded variance assumptions, the authors show different variants of the scheme (SAM, SAM with variance reduction, Ada-SAM) have asymptotic and non-asymptotic convergence. Experiments on MNIST, CIFAR and Penn Treebank seem to show the superior performance over other baselines.

**Limitations And Societal Impact:**

Yes.

**Main Review:**

Originality:
To the best my knowledge, the nonconvex Anderson mixing for nonconvex optimization does not appear in the previous literature.

Clarity:
The paper clearly states its contributions. I would also suggest the authors to create a table to compare the SAM-based approach with existing approaches in stochastic nonconvex optimization.

Significance:
This is my concern about this paper.

1. The motivation of this paper is vague. What are the limitations of SGD and SVRG for nonconvex optimization that make Anderson-Mixing approach important?

2. In my view, the SAM method (Algorithm 1) is worse than nonconvex SGD from every aspect. First, the convergence of nonconvex SGD does not require a large minibatch but it is required for SAM (Theorem 4). Second, Although SAM has the same iteration complexity compared with SGD, SAM has higher per-iteration time complexity (line 9 and line 10) and more memory (line 7 and line 8). I also doubt that it works better than SGD if the authors report running time in the experiments section (in section 4, only the loss versus epochs results are reported).

3. For Algorithm 2 (SAM-VR), it basically mimics the result of [Reddi et al. 2016], but again, the per-iteration complexity is still worse than nonconvex SVRG, since SAM-VR calls SAM as a subroutine. Also the number of SFO calls should be $O(T+T^{2/3}/\epsilon)$ in Theorem 5.

4. In section 3.2, it is important to have a formal theorem describing the convergence of AdaSAM. The current description of this section is vague and I am not sure if it is correct.

Experiments:

1. As I mentioned, since SAM-based approach has high per-iteration complexity and requires more memory, I would suggest the authors to report the running time comparison.

2. Only training loss results are reported. I understand the theory is only for the training loss, but test loss is also important for practitioners.

3. In Figure 1, only large batch size experiments are reported. I understand that the goal is to make it somehow consistent with the large batch requirement in the theory. However, it is known that large batch ends up with worse generalization in machine learning. What is the performance of this algorithm if small batch size is used?

4. In Figure 1(c), why AdaSAM-VR is used? What about the performance of SAM-VR?


**Time Spent Reviewing:**

2 hours

---

> ### Author Response · Authors · 2021-08-10
> **Response to Reviewer wnT6**
>
> Thank you for your comments and advice. We will follow your suggestion to create a table to compare our method with existing approaches in nonconvex stochastic optimization. We hope our following answers can clarify our contributions and address your concerns.
>
> For your concerns about our algorithm:
>
> Q1. The motivation of our paper.
> A1:
> (1) Due to the hardness of nonconvex stochastic optimization, it is known that the complexity $ 1/\epsilon^2 $ of SGD cannot be further improved even when the function is (non-strongly) convex (e.g. see the paper [44]). However, it does not mean SGD is the optimal optimizer in practice, especially for training neural networks. It is empirically observed that the networks trained by SGD achieve higher classification accuracy in CIFAR10/100 than those trained by Adam (Table 1), while in language modeling, Adam obtains lower validation and test loss than SGD (Table 2). Thus, the choice of optimizer is important for different applications, and it is desirable to develop optimization methods with promising results on both aforementioned tasks.
>
> (2) Anderson mixing (AM) is efficient in accelerating fixed-point iteration. It has several theoretical advantages over gradient based methods when solving linear equations [59] and nonlinear programming problems [13]. However, establishing the global convergence of AM is impossible as counter-examples exist [35]. Moreover, if the noisy gradient evaluations exist, the convergence analysis for the variants of AM is difficult. In the current paper, we propose SAM with the optimal complexity and show its advantages when training on both image classification and language modeling tasks.
>
> (3) Since the computation of the variance reduced gradient in SVRG is costly, it may be wasteful to use the gradient estimation only once and discard it in the next iteration. As a second-order method, SAM-VR is possible to benefit more from the less noisy gradient to capture more accurate curvature information, which may accelerate the convergence. Results from Figure 20 in the supplemental material validate this intuition.
>
> Q2. The concern about Theorem 4 and efficiency of SAM.
>
> Q2.1 The large mini-batch assumption in Theorem 4 for SAM.
> A2.1 Although the analysis of SGD (Corollary 2.2 in [17] and Theorem 1 in [44]) does not require the large mini-batch assumption, the choice of step sizes requires knowing the total number of iterations $N$ in advance. In contrast, the step size $\beta_k$ (Line 199) in Theorem 4 is independent of $N$. In fact, under the similar assumption in [17,44], i.e. the step size $\beta_k$ depends on $N$, we can remove the large mini-batch assumption from Theorem 4 for SAM. Here, we give the proof.
>
> Let $ \beta_k = \beta := \min\\{ \frac{\mu}{4L(1+C^{-1})}, \frac{\tilde{D}}{\sigma\sqrt{N}}\\}$, where $ \tilde{D}>0 $ is a problem-independent constant, and substitute it to Inequality (57) in the appendix (Line 607) in the supplemental material, we can finally obtain that
> $$
>  \mathbb{E}[ \\| \nabla f(x_R)\\|\_2^2] \leq \frac{16D_fL(1+C^{-1})}{N\mu^2}+\frac{\sigma}{\mu\sqrt{N}}\left( \frac{4D_f}{\tilde{D}}+\frac{4(L+\mu^{-1})(1+C^{-1})\tilde{D}}{n}\right).
> $$
> Thus, it is seen that $O(1/\epsilon^2)$ can be obtained regardless of the batch size $n$.
>
> Q2.2 The efficiency of our method.
> A2.2
> 1.  Compared to SGD, the additional memory of SAM is $2md$, and the additional computation cost is $O(m^2d)+O(m^3)$, where $m$ is the number of historical iterates.
>
> (a) Memory issue: Since $m=10<<d$ is small in our algorithm, the additional memory requirement $2md$ of SAM is marginal: in the tests on CIFAR10/ResNet20 and CIFAR10/ResNet56, our memory requirements are 1.14x and 1.21x as those of SGDM. From Figure 22 in the supplemental material, it shows that even $m=5$ can achieve comparable results, which can further reduce the memory requirement.
>
> (b) Computational time: In Sec. D.7 in the supplemental material, we have reported the computational time per epoch vs. batch size. From Fig. 23, it is seen that the running time of AdaSAM is much smaller than the second-order method SdLBFGS, and is a little bit slower than SGD. However, as our method achieves similar result with less training epochs, the total training time is less than SGD. In CIFAR10/ResNet 20, training by AdaSAM with 100 epochs (total time is 1188±2s) can achieve 92.01±0.23% accuracy while SGD needs 160 epochs (total time is 1437±15s) with 92.03±0.16% accuracy; in CIFAR100/DenseNet121, training by AdaSAM with 80 epochs (total time is 8776±47s) can achieve 78.84±0.22% accuracy while SGD needs 160 epochs (total time is 12074±112s) with 78.49±0.12% accuracy. Thus, the AdaSAM obtains comparable accuracy with SGD, and saving more than 15% training time.
>
> (c) In fact, the computational complexity can be further reduced to $O(md)+O(m^3)$ as the matrix multiplications $ X_k^\mathrm{T}X_k, R_k^\mathrm{T}R_k $ need not be calculated from scratch because we can reuse the submatrice of $ X_{k-1}^\mathrm{T}X_{k-1}, R_{k-1}^\mathrm{T}R_{k-1} $ calculated in the last iteration. To see this, consider $ X_k^\mathrm{T}X_k  (k\geq m)$.  By column partitioning
> $ X_k = \begin{pmatrix} (X_k)_{1:m-1} & (X_k)_m \end{pmatrix} $, where we use $(A)_\{i:j\}$ to denote the submatrix formed by the $i$ to $j$ columns of matrix $A$, we have
> $$
> X_k^\mathrm{T}X_k =
> \begin{pmatrix}
> (X_k)_\{1:m-1\}^\mathrm{T}(X_k)_\{1:m-1\} & (X_k)_\{1:m-1\}^\mathrm{T}(X_k)_m \\\\
> (X_k)_m^\mathrm{T}(X_k)_\{1:m-1\} & (X_k)_m^\mathrm{T}(X_k)_m
> \end{pmatrix}
> =\begin{pmatrix}
> (X_\{k-1\})_\{2:m\}^\mathrm{T}(X_\{k-1\})_\{2:m\} & (X_\{k\})_\{1:m-1\}^\mathrm{T}(X_k)_m \\\\
> (X_k)_m^\mathrm{T}(X_\{k\})_\{1:m-1\} & (X_k)_m^\mathrm{T}(X_k)_m
> \end{pmatrix}.
> $$
> Hence, the submatrix $ (X\_\{k-1\})_\{2:m\}^\mathrm{T}(X_\{k-1}\)_\{2:m\} $ can be reused, and only $ X_k^T(X_k)_m $ needs to be calculated.  $R_k^\mathrm{T}R_k$ can be similarly calculated. Thus, if we reuse the matrix multiplication as explained before, the additional computational time can be further reduced.
>
> Q3. The effectiveness and technical contribution of SAM-VR.
> We give a numerical comparison of SVRG and AdaSAM-VR on MNIST as shown in Figure 20. It shows SVRG falls far behind AdaSAM-VR.
> Our proof of SAM-VR follows a similar framework of [44, 60], but has fundamental difference with SVRG. Recalling that the update is $x_{k+1}=x_k-H_kg_k$, where $g_k=\nabla f_{S_k}(x_k)$ is the stochastic gradient evaluated at the mini-batch $S_k$. For SGD, $H_k=\eta_k I$. Let $S_{[k-1]}:=(S_0,\dots,S_{k-1})$ denote the random samplings in the first $k$ iterations. One of the key condition in [44, 60] is that $H_k$ is independent of $S_k$  and may depend only on $S_{[k-1]}$, which guarantees $\mathbb{E}[H_kg_k|S_{[k-1]}]=H_k\nabla f(x_k)$ that can largely simplify their derivations. In our algorithm, this assumption does not hold.  We also do not need twice continuously differentiable assumption as that in stochastic quasi-Newton methods [8, 60].  Moreover, AdaSAM does not need to do gradient estimation twice in the same mini-batch [60] or subsampled Hessian-vector products [8], which consumes more and is necessary for their analysis.
> Thanks for pointing out the more precise complexity of SAM-VR, we will revise it in revision.
>
> Q4. Convergence about AdaSAM.
> A4: AdaSAM is a specific instance of SAM with $\delta_k$ chosen as Eq. (13). In AdaSAM, $\delta_k \geq c_2 \beta_k^{-2}$ that satisfies the assumption about $\delta_k$ in Theorems 1-5. So, the convergence results of AdaSAM are the direct application of Theorems 1-5. We will give a more formal corollary and clarify it in revision.
>
> For your concerns about our experiments:
>
> Q5. Running time comparison.
> A5: See our answer to Q2.2.
>
> Q6. The results of test loss.
> A6: We have reported test accuracy in Table 1 and test loss in Fig. 2 for image classification on CIFAR, and validation perplexity in Fig. 3 and test perplexity in Table 2 for language model on Penn TreeBank.
>
> Q7. Tests of small batch sizes.
> A7: In our experiments on CIFAR10/CIFAR100, the batch size is set as 128 as commonly suggested. In our experiment on LSTM on Penn TreeBank, the batch size is set as 20. The results on Table 1, Figure 2, Figure 3, and Table 2 demonstrate the generalization performance of our method.
>
> Q8. The performance of SAM-VR.
> A8: AdaSAM-VR is a specific instance of SAM-VR with $\delta_k$ chosen as Eq. (13). So, we reported the performance of AdaSAM-VR in Fig. 1(c). Other choices of $\delta_k$ in SAM can be found in Sec. D.3 in the supplemental material.

---

> > ### Comment · Reviewer_wnT6 · 2021-08-30
> > **Response**
> >
> > Thank you for your response. I have read the rebuttal. Please note that SGD with large minibatch also does not require the number of iterations $N$ to set the learning rate, and can still get $O(1/\epsilon^2)$ complexity. So there is still no theoretical advantage when compared with SGD. However, this method requires more memory, which may not be good in practice since GPU memory is expensive. For these reasons, I decide not to change my score.

---

> > > ### Author Response · Authors · 2021-08-31
> > > **Thanks for your response**
> > >
> > > Thanks for your response. We would like to give some more detailed comparisons between SGD and our method.
> > >
> > > 1. Theoretical perspective
> > >
> > > a. For the general nonconvex stochastic optimization, the convergence rate of AdaSAM is the same as SGD as the complexity $1/\epsilon^2$ is not improvable in general.
> > > b. For the deterministic quadratic (positive definite) minimization, the gradient descent method only has linear convergence. In this case, we prove that SAM is essentially equivalent to GMRES (see the proof in Section A.2 in the supplementary materials), and the superlinear convergence of GMRES has been proved in [H.A. Van der Vorst and C. Vuik, The superlinear convergence behaviour of GMRES, Journal of Computational and Applied Mathematics 48(1993), 327-341].
> > > c. For the general nonlinear programming, SAM can also have superlinear convergence rate in some cases as it is closely related to the multi-secant quasi-Newton method [13], as also pointed out in Remark 1 in our paper.
> > > d. Anderson mixing (AM) was originally proposed in 1965 and its convergence analysis has not been systematically studied until recent years. Since the original AM does not converge in general, its theoretical analysis can be technical and it is not an easy task to develop a modified version with competitive performance. As far as we know, the stochastic version of AM and the corresponding complete convergence analysis do not exist in the literature.
> > >
> > > 2. Practical perspective
> > >
> > > a. Memory usage during training process
> > > Since the gradient evaluation can be very expensive, in AdaSAM, we reuse the historical gradients in order to capture the curvature information to accelerate convergence. In Figure 22, we plot the performance with different number of historical gradients. It can be seen that $m=5$ or 10 often works well in training neural networks. Moreover, we have used the command “nvidia-smi” to monitor the memory usage during the training process, and found that the additional memory cost of AdaSAM is marginal since the mini-batch data, the computational graph and other data structures related to the model can dominate the memory footprint. The following table lists the GPU memory usage ratio of AdaSAM to SGD when training ResNet20 and ResNet56 on CIFAR10 using different batch size (128, 256, 512 and 1024):
> > >
> > > model  | 128 | 256 | 512 | 1024
> > > :-: | :-: | :-: | :-: | :-:
> > > ResNet20  | 1.14x | 1.13x | 1.07x | 1.05x
> > > ResNet56  | 1.21x | 1.14x | 1.08x | 1.04x
> > >
> > >
> > > b. Computational cost during training process
> > > Compared to SGD, the additional computational cost of AdaSAM can be reduced to $O(md)+O(m^3)$, where $m<<d$ is the historical length. In practice, this additional cost is minor since the function and gradient evaluations often dominate the computation and reduction in training epochs can largely save computational time. Moreover, it is worth mentioning that the additional computation of AdaSAM is the dense matrix computation, which can be further accelerated by the existing parallel computing toolbox. The following table lists the ratio (running time per epoch ) of AdaSAM to SGD when training ResNet20 and ResNet56 on CIFAR10 using different batch size (128, 256, 512 and 1024):
> > >
> > > model  | 128 | 256 | 512 | 1024
> > > :-: | :-: | :-: | :-: | :-:
> > > ResNet20  | 1.34x | 1.30x | 1.18x | 1.07x
> > > ResNet56  | 1.35x | 1.30x | 1.15x | 1.07x
> > >
> > > More details can be seen in Figure 23. The computational time can be further reduced by reusing matrix multiplications as explained in the previous response.
> > >
> > > c. Total training time
> > > It is usual that the AdaSAM converges faster than SGD in terms of the number of epochs. Thus, the total training time of AdaSAM can be less than that of SGD. In the following table, we set the accuracy of the fine-tuned SGD as the baseline, and list the ratio of total running time of AdaSAM to SGD in training several neural networks on CIFAR10 and CIFAR100, where the AdaSAM were terminated when achieving comparable or better test accuracy than SGD.
> > >
> > > model | CIFAR10/ResNet20 | CIFAR10/ResNet56   | CIFAR100/ResNeXt50 | CIFAR100/DenseNet121
> > > :-: | :-: | :-: | :-: | :-:
> > > Accuracy difference (AdaSAM - SGD)    | -0.02% | 0.00%   | -0.04% | +0.35%
> > > Time ratio (AdaSAM/SGD) | 0.83      | 0.84      |   0.83    | 0.73
> > >
> > > It is shown that with the similar accuracy as SGD, AdaSAM can save more than 15% training time.
> > >
> > > d. Test performance
> > > When we run our method within the same number of epochs as SGD, it is shown in Table 1 and Table 2 in our paper that our method achieves better generalization performance than SGD in both image classification and language modeling. More specifically, in average, the test accuracy is AdaSAM - SGD$\geq $ 0.3% on CIFAR10 and AdaSAM - SGD > 0.8% on CIFAR100; for training 1-layer, 2-layer and 3-layer LSTM on Penn TreeBank, the test perplexity of our method is 5.87, 3.94 and 2.09 lower than SGD, respectively.
> > >
> > > Based on the above analysis, we summarize the comparisons of our method and SGD in the following table:
> > >
> > > ( '+' ('-') means that SAM is better (worse) than SGD, and '=' means that SAM is similar to SGD.)
> > >
> > > criterion | general | specific | memory | time/epochs | epochs |  training time | generalization
> > > :-: | :-: | :-: | :-: | :-: | :-: | :-: | :-:
> > > SAM vs. SGD | = | + | - | - | + | + | +
> > >
> > > Here, the table includes 7 types of criterion. “general” denotes general nonconvex stochastic optimization; “specific” denotes some specific problems such as deterministic quadratic optimization; “memory” is the memory usage; “time/epochs” is the running time per epoch; “epochs” is the training epochs to achieve comparable test performance as SGD and “training time” is the corresponding running time; “generalization” denotes the test performance when training with the same number of epochs as SGD.
> > >
> > > Therefore, it may be a trade-off for the practitioners to decide whether to use more computational and memory resource with our method to obtain better solutions or faster convergence.
> > >
> > > We hope the above explanations can help you better evaluate this work.

---

### Official Review · Reviewer_bi9T · 2021-07-16

**Rating:** 7
**Confidence:** 4

**Summary:**

The paper proposes a variant of Anderson mixing applied to a stochastic optimization problem. Convergence analysis is carried out and experiments show good results.

**Limitations And Societal Impact:**

See above

**Main Review:**

The proposed method relies on damping and regularization. These techniques are very common in Anderson mixing (AM) methods due to the potential divergence of AM and the numerical instability when solving its subproblems.

In this paper, the positive definiteness of a matrix, acting like an approximate Hessian, is checked at every step. This allows the author(s) to obtain convergence guarantees, which are otherwise impossible for AM. The damping condition enforces the method to stay close to a stochastic descent method; when alpha_k is small (to satisfy condition (15)), we essentially take an SGD step.

My concerns are that:

(i) since the stepsize beta is very small and is often reduced further as the algorithm proceeds, condition (15) enforces that alpha should tend to zero too. So, effectively we are just running SGD most of the time. It would be interesting to profile and see how often the Anderson step are performed compared to SGD.

(ii) the rate does not show the benefit over SGD type methods that require much less memory footprint. I am not concerning the extra computations as they are just some matrix-matrix multiplications and GPUs are super efficient at this. However, the memory footprint is extremely important when it comes to training large models on GPUs. GPU's memory is very small compared to CPUs, and a single model would contain millions if not billions of parameters. The ADM method stores two extra matrices that would need 2*m times the memory required to store one model, which would be otherwise used to increase the mini-batch size and hence reduce the runtime.

Some other comments:

In Algorithm 1, it is stated that beta_k = alpha_k = 1, but the conditions on Theorem 1, 2, 3 require beta_k -> 0. I believe it shoud be beta_0 = 1??

Theorem 2 needs smoothness and boundedness of f + bounded variance and second moment of the gradients, which seems not very standard.

**Time Spent Reviewing:**

5

---

> ### Author Response · Authors · 2021-08-10
> **Response to Reviewer bi9T**
>
> Thank you for your review. We hope our following answers can address your concerns.
>
> Q1: How often the Anderson step is performed compared with SGD?
> A1: The training profile of $\alpha_k$ is shown in Figure 11 (c) and (f) (MNIST) and Figure 12 (f) (CIFAR). When $\alpha_k=0$, SAM is reduced to SGD. From the above results, it can be seen that Anderson step dominates a large portion of the training process, while the SGD-like step mainly occurs at the end of the training process.
>
> Q2: The memory requirement of our method.
> A2: (1) Although compared with SGD, our method needs additional cost of computation and memory, the advantages of (preconditioned) SAM in terms of epochs (Figure 1, 3 in the main article and Figure 4, 8 in the supplemental material) and generalization error (Table 1 and Table 2) may make it worthy. In CIFAR10/ResNet 20, training by AdaSAM with 100 epochs (total time is 1188±2s) can achieve 92.01±0.23% accuracy while SGD needs 160 epochs (total time is 1437±15s) to achieve 92.03±0.16% accuracy; in CIFAR100/DenseNet121, training by AdaSAM with 80 epochs (total time is 8776±47s) can achieve 78.84±0.22% accuracy while SGD needs 160 epochs (total time is 12074±112s) to achieve 78.49±0.12% accuracy. Thus, the AdaSAM obtains comparable accuracy with SGD, and saving more than 15% training time.  (2) Also, since the historical length $m$ is usually small in our algorithm ($m=10<<d$ on CIFAR and Penn TreeBank experiments), the additional memory requirement $2md$ of SAM is marginal: In the test on CIFAR10/ResNet20 and CIFAR10/ResNet56, our memory requirements are only 1.14x and 1.21x as those of SGDM. From Figure 22 in the supplemental material, it shows that even $m=5$ can achieve comparable results, which can further reduce the memory requirement.
>
> For extremely large model training, distributed parallel computing is necessary, which facilitates larger memory. In this case, compared with the additional cost incurred by AdaSAM, the heavy cost of data access, data communication and synchronization of all-works often stands out as the more critical issue for distributed SGD. We leave the design of distributed AdaSAM as future work.
>
> For your other comments:
> Q3:  $\beta_k = \alpha_k = 1$ in the first line of Algorithm 1.
> A3: Thanks for your pointing out this issue. It is $\alpha_0=1, \beta_0=1$ in Algorithm 1. We will clarify it in revision.
>
> Q4: Are the assumptions of Theorem 2 standard?
> A4: Theorem 2 is a stronger result of Theorem 1 as Theorem 2 requires the boundedness of the gradient estimation. The assumption that $f$ is bounded from below is a standard assumption in nonconvex optimization [17]. The smoothness of $f$ is necessary for a gradient method. The boundedness of the noisy gradient is a common assumption in Adam and the stochastic quasi-Newton method proposed in [8].

---

> > ### Comment · Reviewer_bi9T · 2021-08-30
> > **Thank you for the response**
> >
> > Thank you for the response.
> >
> > An extra question: how did you estimate the memory requirement? Since the $2*m$ extra #parameters-dimensional vectors can be quite large, I am quite surprised that the increase in the memory footprint is only 1.14x or 1.2x compared to SGM.
> >
> > While I agree that the theory of Anderson mixing in the paper does not show any better (worst-case) rate than other gradient-based schemes, Anderson mixing works nicely and consistently in practice. It seems that there have been very few experimental works on AM in the context of training neural nets. Therefore, I believe that this work can be a nice contribution from the practical side. I raised my score to 7.

---

> > > ### Author Response · Authors · 2021-08-31
> > > **Thank you for your support**
> > >
> > > Thanks a lot for your support.
> > >
> > > Q: Memory footprint during training.
> > > A: We monitored GPU memory usage by using the command “nvidia-smi”, which displays the memory footprint for each training process. Besides the model parameters, the memory footprint also includes the mini-batch data, computational graph for automatic differentiation and other data structures used in the model. It is observed that the model parameters may occupy a small portion of the total memory. Moreover, we found that the percentage of the additional memory footprint of our method AdaSAM is even smaller when increasing the mini-batch size. For example, in the tests on CIFAR10/ResNet20 and CIFAR10/ResNet56, when setting mini-batch size to be 1024, the total memory usage of AdaSAM is only 5% larger than SGDM.

---

### Official Review · Reviewer_v7Hn · 2021-07-16

**Rating:** 7
**Confidence:** 3

**Summary:**

The paper proposes a stochastic Anderson mixing (SAM) scheme and its variance reduction version to solve nonconvex stochastic optimization problems. Extensive experiments are conducted to verify the efficacy of the proposed methods.

**Limitations And Societal Impact:**

N.A

**Main Review:**

The paper provides promising results and the idea of using damped projection combined with adaptive regularization is new. I'm impressed with the experiments.

1. Provide the (approximate) formula of $M_f$ in (20). This constant is important to visualize the rate of $1/N \sum E || \nabla f(x_k) ||^2$, see (23).

2. How is the batch size $n_k$ in Theorem 4 estimated (see line 205) when $\sigma$ in Assumption 2 is not easy to determine?

3. Is the batch size in the experiments chosen following the theory?

**Time Spent Reviewing:**

10

---

> ### Author Response · Authors · 2021-08-10
> **Response to Reviewer v7Hn**
>
> Thank you for your review. We hope our following answers can address your concerns.
>
> Q1: Provide the (approximate) formula of $M_f$ in Eq. (20).
> A1: From the proof of Theorem 1 (Lines 578-586 in the appendix in the supplemental material), a bound of $M_f$ is $ \gamma_0 = f(x_0)+(L+\mu^{-1})(1+C^{-1})\frac{\sigma^2}{n}\sum_{i=0}^{\infty}\beta_i^2 $.
>
> Q2: How is the batch size $n_k$ in Theorem 4 (Line 205) estimated?
> A2: In practice, choosing the theoretical batch size $n_k$ is difficult since $\sigma$ and $L$ are unknown. A rough estimation of $n_k$ is $ \Omega(\sqrt{\bar{N}})$, where $\bar{N}$ is the total number of $\mathcal{SFO}$-calls needed to calculate the stochastic gradients. For example, running for 100 epochs on a dataset containing 10K data samples needs $\bar{N} = 100\times (10\times 10^3) = 10^6$ $\mathcal{SFO}$-calls. Then a rough estimation of $n_k$ given by $\sqrt{\bar{N}}$ is $ \sqrt{10^6} = 1000$. After that, different values of batch size can be tried based on this rough estimation.
>
> Q3: Is the batch size in the experiments chosen following the theory?
> A3: In our experiment on MNIST, we followed the theory and chose relatively large batch size (2K, 3K, 6K, etc.). In the experiments on CIFAR and Penn TreeBank, for fair comparison, we followed the standard setting of batch size in the literature (e.g., [21] and [65]): the batch size is 128 on CIFAR10/CIFAR100, and 20 on Penn TreeBank.

---

### Author Response · Authors · 2021-08-28
**Author Response**

Dear reviewers,

Thank you very much for your efforts in reviewing this paper and providing constructive suggestions.
We tried our best to address all the proposed concerns and problems. Are there any unclear explanations that we could further clarify or expand upon? We are happy to address them as well.

Yours faithfully,

The Authors

---

### Decision · Program_Chairs · 2021-09-28

**Decision:**

Accept (Poster)

**Comment:**

The direction is novel and interesting, the analysis of a stochastic version of AM could be independently interesting. Even though the practical aspects are not the strongest, I recommend acceptance; in my opinion the results are of interest to the community.

The authors did generally a good job responding to reviewers questions resulting to one reviewer raising their score.

There are legitimate arguments by one reviewer saying that there are no theoretical benefits of SAM over SGD (both achieve theoretical optimal, with SGD being simpler, less memory and computation-intensive). And I encourage the authors to include relevant discussion.


**Consistency Experiment:**

NeurIPS has a long history of experimentation. In 2014, NeurIPS ran an experiment in which 10% of submissions were reviewed by two independent committees to quantify the randomness in the review process. This year, we repeated a variant of this experiment to see how the quality of the review process has changed over time.  This paper was part of the experiment and was therefore assigned to two committees (consisting of reviewers, an Area Chair, and a Senior Area Chair) that reached independent decisions.  If both committees made the same recommendation, this recommendation was followed. If a single committee recommended acceptance, the paper was accepted (with the exception of a few cases in which the other committee identified what we considered a fatal flaw, e.g., an error in a key result).

Both committees reached the same decision: **Accept (Poster)**

The other committee assigned to the paper recommended **Accept (Poster)**.  You can find the other set of reviews, along with any follow up discussion with the authors here:
https://openreview.net/forum?id=hx2Ckkzdf53